# Hepatic stellate cells control liver zonation, size and functions via R-spondin 3

Atsushi Sugimoto[1,2,26], Yoshinobu Saito[1,2,3,26], Guanxiong Wang[4,5,26], Qiuyan Sun[1,2], Chuan Yin[1,2], Ki Hong Lee[4,5], Yana Geng[1,2], Presha Rajbhandari[6], Celine Hernandez[1,2], Marcella Steffani[1,2], Jingran Qie[1,2], Thomas Savage[7], Dhruv M. Goyal[1,2], Kevin C. Ray[8], Taruna V. Neelakantan[6], Deqi Yin[1,2], Johannes Melms[1], Brandon M. Lehrich[9], Tyler M. Yasaka[9], Silvia Liu[9], Michael Oertel[9], Tian Lan[10], Adrien Guillot[10], Moritz Peiseler[10], Aveline Filliol[1,2], Hiroaki Kanzaki[11], Naoto Fujiwara[11], Samhita Ravi[12], Benjamin Izar[1,13,25], Mario Brosch[14], Jochen Hampe[14], Helen Remotti[2,15], Josepmaria Argemi[16,17], Zhaoli Sun[18], Timothy J. Kendall[19], Yujin Hoshida[11], Frank Tacke[10], Jonathan A. Fallowfield[19], Storm K. Blockley-Powell[1,2], Rebecca A. Haeusler[1,2], Jonathan B. Steinman[20], Utpal B. Pajvani[1,2,21], Satdarshan P. Monga[9], Ramon Bataller[22], Mojgan Masoodi[23], Nicholas Arpaia[2,13,25], Youngmin A. Lee[8], Brent R. Stockwell[2,6,25], Hellmut G. Augustin[4,5 ✉] & Robert F. Schwabe[1,2,10,21,24,25 ✉]

Hepatic stellate cells (HSCs) have a central pathogenetic role in the development of liver fibrosis. However, their fibrosis-independent and homeostatic functions remain poorly understood[1–5]. Here we demonstrate that genetic depletion of HSCs changes WNT activity and zonation of hepatocytes, leading to marked alterations in liver regeneration, cytochrome P450 metabolism and injury. We identify R-spondin 3 (RSPO3), an HSC-enriched modulator of WNT signalling, as responsible for these hepatocyte-regulatory effects of HSCs. HSC-selective deletion of *Rspo3* phenocopies the effects of HSC depletion on hepatocyte gene expression, zonation, liver size, regeneration and cytochrome P450-mediated detoxification, and exacerbates alcohol-associated and metabolic dysfunction-associated steatotic liver disease. *RSPO3* expression decreases with HSC activation and is inversely associated with outcomes in patients with alcohol-associated and metabolic dysfunction-associated steatotic liver disease. These protective and hepatocyte-regulating functions of HSCs via RSPO3 resemble the R-spondin-expressing stromal niche in other organs and should be integrated into current therapeutic concepts.

The liver functions as a central hub for carbohydrate, glucose and protein metabolism and the detoxification of endogenous and exogenous substances[6]. As many of its functions are essential for life, the liver is endowed with a considerable capacity for regeneration and repair[2,7]. The liver's metabolic functions are carried out by hepatocytes in a zonation-dependent manner[8,9]. Hepatocytes reside within a complex niche that supports their metabolic functions: liver sinusoidal endothelial cells (ECs) promote the metabolic influx and efflux from or to the circulation[10]. Cholangiocytes enable the transport of bile acids and metabolites, excreted by hepatocytes, towards the intestine[11]. Kupffer cells form a protective firewall against gut-derived bacteria[12]. HSCs are best known for storing retinoids in health and mediating fibrogenesis after injury[1–5]. Moreover, their anatomical position, long protrusions[13] and sizeable ligand–receptor repertoire[14] render HSCs prone to interactions with other cells, including hepatocytes. However, the functions of HSCs within the hepatic niche remain poorly

[1]Department of Medicine, Columbia University, New York, NY, USA. [2]Columbia University Digestive and Liver Disease Research Center, New York, NY, USA. [3]Department of Gastroenterology and Hepatology, Osaka University Graduate School of Medicine, Osaka, Japan. [4]Division of Vascular Oncology and Metastasis Research, German Cancer Research Center, Heidelberg, Germany. [5]European Center for Angioscience (ECAS), Medical Faculty Mannheim, Heidelberg University, Mannheim, Germany. [6]Department of Biological Sciences and Department of Chemistry, Columbia University, New York, NY, USA. [7]Department of Microbiology & Immunology, Columbia University, New York, NY, USA. [8]Department of Surgery, Vanderbilt University Medical Center, Nashville, TN, USA. [9]Department of Pharmacology and Chemical Biology, Pittsburgh Liver Research Center, and Organ Pathobiology and Therapeutics Institute, University of Pittsburgh School of Medicine, Pittsburgh, PA, USA. [10]Department of Hepatology & Gastroenterology, Charité—Universitätsmedizin Berlin, Berlin, Germany. [11]Liver Tumour Translational Research Program, Harold C. Simmons Comprehensive Cancer Center, Division of Digestive and Liver Diseases, University of Texas Southwestern Medical Center, Dallas, TX, USA. [12]Division of Gastroenterology, Hepatology and Nutrition, Department of Medicine, University of Pittsburgh School of Medicine, Pittsburgh, PA, USA. [13]Columbia Center for Translational Immunology, Department of Medicine, Columbia University, New York, NY, USA. [14]Department of Internal Medicine I, University Hospital and Faculty of Medicine, Technische Universität Dresden, Dresden, Germany. [15]Department of Pathology and Cell Biology, Columbia University Irving Medical Center, New York, NY, USA. [16]Liver Unit and RNA Biology and Therapies Program, Cima Universidad de Navarra, Cancer Center Clínica Universidad de Navarra (CCUN), Pamplona, Spain. [17]Centro de Investigación Biomédica en Red de Enfermedades Hepáticas y Digestivas, Instituto de Salud Carlos III, Madrid, Spain. [18]Department of Surgery, Johns Hopkins University School of Medicine, Baltimore, MD, USA. [19]Institute for Regeneration and Repair, University of Edinburgh, Edinburgh, UK. [20]Department of Pediatrics, Columbia University, New York, NY, USA. [21]Institute of Human Nutrition, New York, NY, USA. [22]Liver Unit, Institut d'Investigacions Biomèdiques August Pi i Sunyer (IDIBAPS), Hospital Clinic, Barcelona, Spain. [23]Institute of Clinical Chemistry, Inselspital, Bern University Hospital, Bern, Switzerland. [24]Burch-Lodge Center for Human Longevity, Columbia University, New York, NY, USA. [25]Present address: Herbert Irving Comprehensive Cancer Center, New York, NY, USA. [26]These authors contributed equally: Atsushi Sugimoto, Yoshinobu Saito, Guanxiong Wang. ✉e-mail: augustin@angioscience.de; rfs2102@cumc.columbia.edu

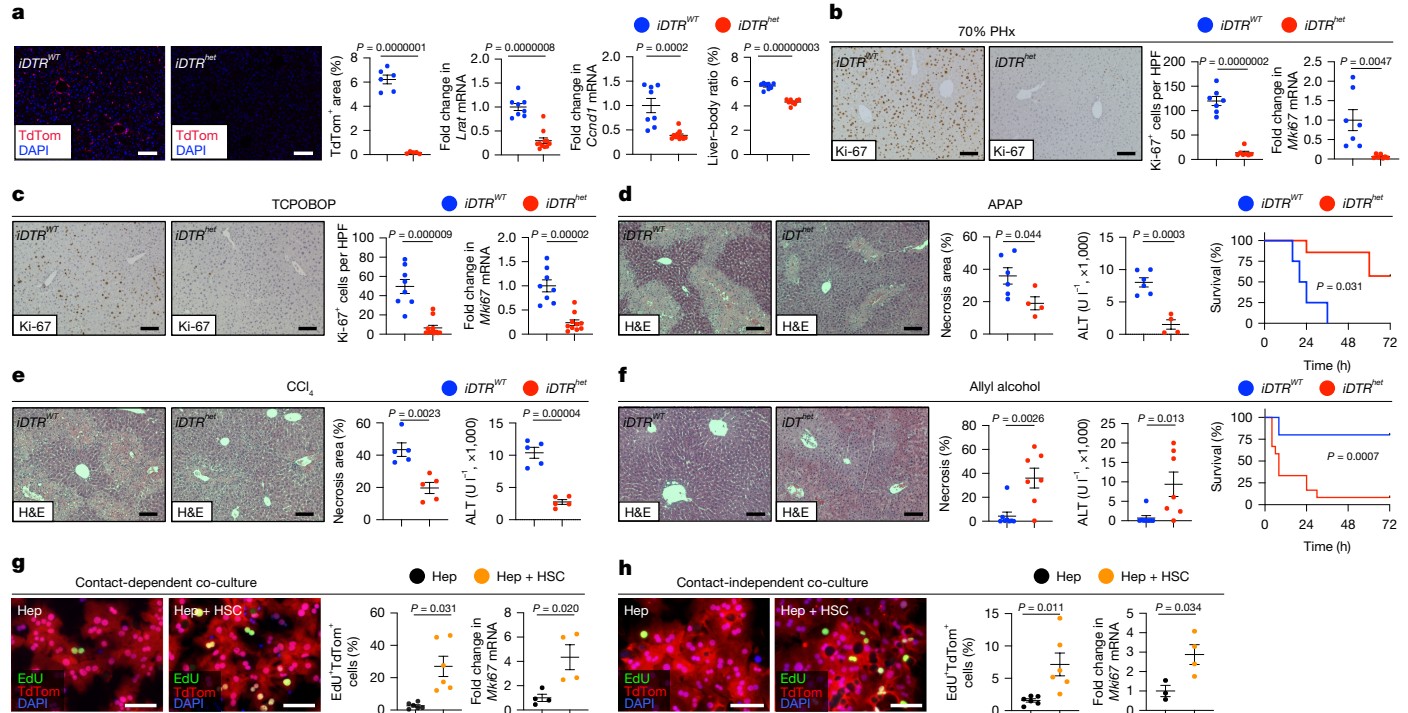

**Fig. 1 | HSCs regulate liver regeneration and injury. a**, *Lrat-cre*[+]TdTom[+] mice expressing iDTR (*iDTR*[het]) or not (*iDTR*[WT]) were injected with diphtheria toxin (DT). The TdTom[+] area (*n* = 6 (*iDTR*[WT]) and *n* = 5 (*iDTR*[het])), *Lrat* and *Ccnd1* mRNA (by qPCR, *n* = 8 (*iDTR*[WT]) and *n* = 11 (*iDTR*[het])) and the liver–body weight ratio (*n* = 8 (*iDTR*[WT]) and *n* = 8 (*iDTR*[het])) were determined 7 days later. **b,c**, *iDTR*[WT] and *iDTR*[het] mice were treated with DT and, 1 week later, were subjected to 70% PHx (*n* = 7 per group) (**b**) or treatment with constitutive androstane receptor agonist (**c**), followed by Ki-67 IHC and quantification per high-power field (HPF) (*n* = 8 (*iDTR*[WT]) and *n* = 11 (*iDTR*[het])) as well as qPCR analysis of *Mki67* (*n* = 8 (*iDTR*[WT]) and *n* = 10 (*iDTR*[het])). **d**, *iDTR*[WT] and *iDTR*[het] mice were treated with DT. Then, 1 week later, the mice were subjected to treatment with a sublethal dose of APAP to determine the serum ALT and necrosis area in haematoxylin and eosin (H&E) sections (*n* = 6 (*iDTR*[WT]) and *n* = 4 (*iDTR*[het])), or with a lethal APAP dose to determine survival (*n* = 4 (*iDTR*[WT]) and *n* = 7 (*iDTR*[het])). **e,f**, *iDTR*[WT] and *iDTR*[het]

mice (*n* = 5 per group) were treated with DT and 1 week later were then treated with CCl₄ (**e**; 0.5 mg per kg, *n* = 5 per group) or allyl alcohol (**f**; 60 mg per kg) to determine the serum ALT and necrosis area in H&E sections (*n* = 8 (*iDTR*[WT]) and *n* = 7 (*iDTR*[het])) or a lethal dose of allyl alcohol (**f**; 75 mg per kg; *n* = 10 (*iDTR*[WT]) and *n* = 12 (*iDTR*[het])) to determine survival. **g,h**, Primary mouse hepatocytes (Hep) were co-cultured with or without primary mouse HSCs in a contact-dependent (**g**; EdU, *n* = 6 per group; *Mki67* mRNA, *n* = 4 per group) or contact-independent (**h**; EdU, *n* = 6 per group; qPCR, *n* = 3 (hepatocytes), *n* = 4 (hepatocytes + HSCs)) manner to determine proliferation based on EdU staining and qPCR analysis of *Mki67* mRNA. Data are mean ± s.e.m. For **a–h**, each dot represents one biological replicate. Scale bars, 100 μm (**a–h**). *P* values were calculated using unpaired two-tailed *t*-tests (**a–c, e, g** and **h**, and **d** and **f** (middle and left)) or log-rank test (**d** and **f** (right)).

understood[5]. HSC depletion with lower potency and specificity, using gliotoxin or GFAP-TK, has yielded inconsistent effects on liver regeneration and injury[15–18]. Recently, the more specific and efficient just eGFP death-inducing (JEDI) depletion method revealed a role for HSCs in homeostatic hepatocyte proliferation[19].

Here we combined genetic HSC depletion with conditional knockout of candidate genes to analyse fibrosis-independent functions of HSCs. We identified hepatocyte-regulatory functions of HSCs, controlling hepatic zonation, metabolism, injury and regeneration via RSPO3 and the subsequent activation of WNT signalling, a master regulator of liver zonation and function[20]. Together, our findings provide evidence for a mesenchymal liver niche that controls epithelial cell functions in health and disease, akin to the R-spondin–LGR5-mediated mesenchymal–epithelial cross-talk in the gastrointestinal tract[21,22].

## Altered regeneration and injury in HSC-depleted livers

To understand their functions in the healthy liver, we depleted HSCs through *Lrat-cre* and Cre-inducible diphtheria toxin receptor (iDTR)[14]. Low-dose diphtheria toxin efficiently reduced TdTom[+] HSCs and *Lrat* and *Colec11* mRNA without triggering inflammation or a decrease in *Lrat-cre*-labelled TdTom[+] cells and *Lrat* mRNA in other organs 7 days later (Fig. 1a and Extended Data Fig. 1a–c). Flow cytometry analysis of HSC-depleted livers did not reveal immune cell alterations except for an increase in neutrophils, probably a consequence of HSC killing,

and a decrease in dendritic cells (Supplementary Information 1a). As reported previously[19], HSC-depleted livers were smaller and contained smaller hepatocytes and less *Ccnd1* mRNA, a potent driver of hepatocyte proliferation[23] (Fig. 1a and Extended Data Fig. 1d). We therefore investigated the role of HSCs in liver regeneration, which is essential for restoring critical hepatic functions after the loss of functional liver mass[7]. After 70% partial hepatectomy (PHx), we observed substantially impaired regeneration with significant decreases in the liver–body weight ratio, Ki-67[+] and cyclin D1[+] hepatocytes and *Mki67* and *Ccnd1* mRNA in HSC-depleted mice (Fig. 1b and Extended Data Fig. 1e). Like other mice with impaired liver regeneration[24,25], HSC-depleted livers exhibited compensatory hyperproliferation at later stages, while HSC depletion persisted (Extended Data Fig. 1f). A similar but longer-lasting reduction in the liver–body weight ratio and the number of Ki-67[+] and cyclin D1[+] hepatocytes was observed in HSC-depleted mice after inducing liver proliferation by the constitutive androstane receptor agonist 1,4-bis[2-(3,5-dichloropyridyloxy)]benzene (TCPOBOP) (Fig. 1c and Extended Data Fig. 1g,h). Depleting HSCs 1 h after 70% PHx also suppressed liver regeneration (Extended Data Fig. 1i). When injected with hepatotoxin CCl₄ or acetaminophen (APAP), the leading cause for acute liver failure in patients, HSC-depleted mice displayed a nearly 80% reduction in liver injury, as determined by serum ALT and the necrosis area, as well as a significant increase in survival (Fig. 1d,e). In contrast to APAP and CCl₄, which cause pericentral liver injury, treatment with a predominantly periportal liver toxin, allyl alcohol, substantially

increased liver injury and mortality in HSC-depleted mice (Fig. 1f). Although the smaller size of HSC-depleted livers could affect injury responses, the decreased injury in some and increased injury in other models makes this unlikely. In summary, our data establish HSCs as potent regulators of liver regeneration and injury, two fundamental processes determining liver disease development and outcomes, possibly in a zone-dependent manner.

To dissect the underlying molecular mechanisms, we first determined whether HSCs may exert direct effects on hepatocytes. While hepatocyte monocultures exhibited the expected low proliferation rate, both direct and contact-independent co-culture with HSCs significantly increased hepatocyte proliferation (Fig. 1g,h), suggesting that a soluble factor is responsible. Hepatocyte growth factor (HGF) is a potent hepatomitogen that is enriched in HSCs[7,14]. However, mice lacking *Hgf* in HSCs (*Hgf^ΔHSC*)[14] did not display impaired liver regeneration and showed increased CCl₄-induced liver injury[14], in contrast to the decreased liver injury in HSC-depleted mice. Type I collagen is another HSC mediator regulating hepatocyte proliferation during hepatocarcinogenesis[14]. However, liver- or HSC-specific deletion of *Col1a1* or receptors with critical roles in HSC activation, such as *Tgfbr1* and *Pdgfrb*, did not phenocopy the changes in proliferation after 70% PHx (Extended Data Fig. 2a–g) or CCl₄-induced liver injury[14] seen in HSC-depleted mice. Together, these findings suggested that an HSC-secreted factor mediates the effects of HSCs on hepatocyte proliferation and injury.

## HSCs control hepatocyte functions and zonation

To gain insights into the mechanisms of HSC–hepatocyte cross-talk, we queried transcriptomic data in HSC-depleted livers for differential gene expression and pathway activation. To focus on the effects of HSC depletion on the liver rather than the expected alterations in HSC-expressed genes (Extended Data Fig. 3a,b and Supplementary Table 1), we removed HSC-enriched genes. Notably, the top altered pathways in HSC-depleted livers, achieved using the iDTR system or the JEDI model[19], were metabolic, including drug, cytochrome P450, linoleic acid, tyrosine, tryptophan and caffeine metabolism (Extended Data Fig. 4a and Supplementary Table 2). As these essential metabolic functions are carried out by hepatocytes, our analysis hinted towards an unrecognized role of HSCs in regulating hepatocyte and liver functions. Using RNA sequencing (RNA-seq) and quantitative PCR (qPCR), we identified a decrease in characteristic hepatocyte-enriched genes with roles in metabolism, regeneration and the development of metabolic dysfunction-associated steatotic liver disease (MASLD) and alcohol-related liver disease (ALD), such as cytochrome P450 oxidases *Cyp1a2* and *Cyp2e1*, and *Ang*, *Gulo*, *Hsd3b5*, *Avpr1a* and *Chrna4*[26–28] as well as an increase in *Cyp2f2* (Fig. 2a,b and Extended Data Fig. 4b). Single-cell RNA-seq (scRNA-seq) analysis confirmed that these genes were indeed enriched in hepatocytes (Extended Data Fig. 4c). Immunohistochemistry (IHC) analysis confirmed an approximately 70% decrease and altered pattern of CYP1A2, CYP2E1 and CYP2F2 protein expression, accompanied by decreased CYP2E1 activity (Fig. 2c and Extended Data Fig. 4d).

Further zone-specific quantification identified marked changes in liver zonation, showing a condensed pericentral-to-midzonal CYP2E1⁺CYP1A2⁺RGN⁺ zone, a concomitant expansion of the CYP2F2⁺HAL⁺ periportal zone and unaltered or minimally reduced pericentral marker glutamine synthetase (GS) and OAT in HSC-depleted mice (Fig. 2d,e and Extended Data Fig. 4e). The condensation of zone 3 and expansion of zone 1 in HSC-depleted livers were confirmed by a 100-plex spatial transcriptomic panel[29] (Fig. 2f and Extended Data Fig. 4f). Similar alterations in CYP2E1 and CYP1A2 were observed in the JEDI HSC depletion model (Extended Data Fig. 4g). Thus, in HSC-depleted mice, the hepatic midzone shifted from expressing well-established zone 3 markers to expressing zone 1 markers. The decrease in hepatocyte proliferation after 70% PHx or TCPOBOP as well as alterations in cell death after APAP, CCl₄ or allyl alcohol treatment followed similar zonal patterns, with the largest differences occurring in the midzonal areas in HSC-depleted mice (Fig. 2g,h). Together, these findings revealed a role of HSCs in regulating the metabolic zonation of the liver and zone-specific hepatocyte proliferation and cell death.

## HSCs regulate hepatic WNT activity

We next sought to identify mediators through which HSCs regulate hepatocyte proliferation, metabolism and zonation. Consistent with its master role in liver function and zonation[9,20], WNT/β-catenin emerged as a top hit in our pathway analysis in HSC-depleted livers (Extended Data Fig. 4a). Gene set enrichment analysis (GSEA) confirmed significant changes in the WNT pathway in JEDI and iDTR HSC-depleted mice (Fig. 3a and Extended Data Fig. 4h). HSC-depleted livers displayed a substantial reduction in hepatocyte-enriched WNT-target genes, including the top 15 genes decreased in *Ctnnb1^ΔHep* livers (Fig. 3a, Extended Data Fig. 4b,i and Supplementary Table 3). To understand how HSCs might regulate WNT signalling in the liver, we next analysed hepatic WNT expression using scRNA-seq. Although *Wnt4* and *Wnt5a* were enriched in HSCs (Extended Data Fig. 5a), mice with HSC-selective deletion of Wntless (*Wls^ΔHSC*), a gene required for WNT secretion, did not display the same alterations in *Cyp2e1* and *Cyp1a2*, in the liver–body weight ratio or in APAP-induced liver injury compared with HSC-depleted mice (Extended Data Fig. 5b–g). These findings align with previous studies[30] and exclude HSC-secreted WNTs mediating the alterations observed in HSC-depleted mice.

To identify candidates through which HSCs regulate WNT activity in hepatocytes, we next analysed ligand–receptor interactions using CellPhoneDB. RSPO3, a potent positive regulator of the WNT pathway[21,22], emerged among the top-ten HSC ligands mediating HSC–hepatocyte interactions through its corresponding receptors LGR4 and LGR5 (Fig. 3b and Supplementary Table 4). *Rspo3* was highly enriched in HSCs and a small population of pericentral ECs without appreciable expression in other liver cells (Fig. 3c and Extended Data Fig. 6a–c). Conversely, RSPO3 receptors *Lgr4* and *Lgr5* were enriched in hepatocytes (Extended Data Fig. 6d), as described previously[24]. Notably, *Rspo3* was expressed at higher levels compared with other secreted mediators of the WNT pathway in isolated HSCs (Extended Data Fig. 6e). Moreover, *Rspo3* was significantly reduced in iDTR and JEDI HSC-depleted livers and correlated significantly with the HSC marker *Lrat* and the WNT-target genes *Cyp1a2* and *Cyp2e1* in HSC-depleted mice (Extended Data Fig. 6f–h). RNAscope analysis demonstrated the expression of *Rspo3* within *Lrat-cre*-labelled HSCs (Fig. 3d). Enzyme-linked immunosorbent assay (ELISA) and immunoblotting confirmed RSPO3 expression in HSCs at the protein level (Fig. 3e and Extended Data Fig. 6i). Consistent with previous studies[31–33] and befitting the zonal alterations in HSC-depleted livers, we observed a pericentral-to-periportal gradient of *Rspo3* in HSCs (Fig. 3f). Notably, mice with hepatocyte-specific deletion of the RSPO3 receptors *Lgr4* alone, *Lgr4* plus *Lgr5* or their downstream target β-catenin displayed alterations in liver size, zonation and regeneration, including hyperproliferation at late timepoints, similar to HSC-depleted mice[24,25]. On the basis of these findings, we hypothesized that RSPO3 mediates the communication between HSCs and hepatocytes and that its lack in HSC-depleted livers could be the cause of altered regeneration and hepatocyte injury. Treatment with recombinant RSPO3 induced proliferation and WNT-dependent gene expression in AML12 hepatocytes, while RSPO3 neutralization significantly reduced the proliferation of primary hepatocytes in HSC–hepatocyte co-cultures (Extended Data Fig. 7a,b), suggesting direct effects on hepatocytes. This was corroborated by the restoration of liver regeneration after 70% PHx and APAP-induced liver injury after AAV8-mediated rescue of *Rspo3* expression in HSC-depleted mice (Extended Data Fig. 7c–f).

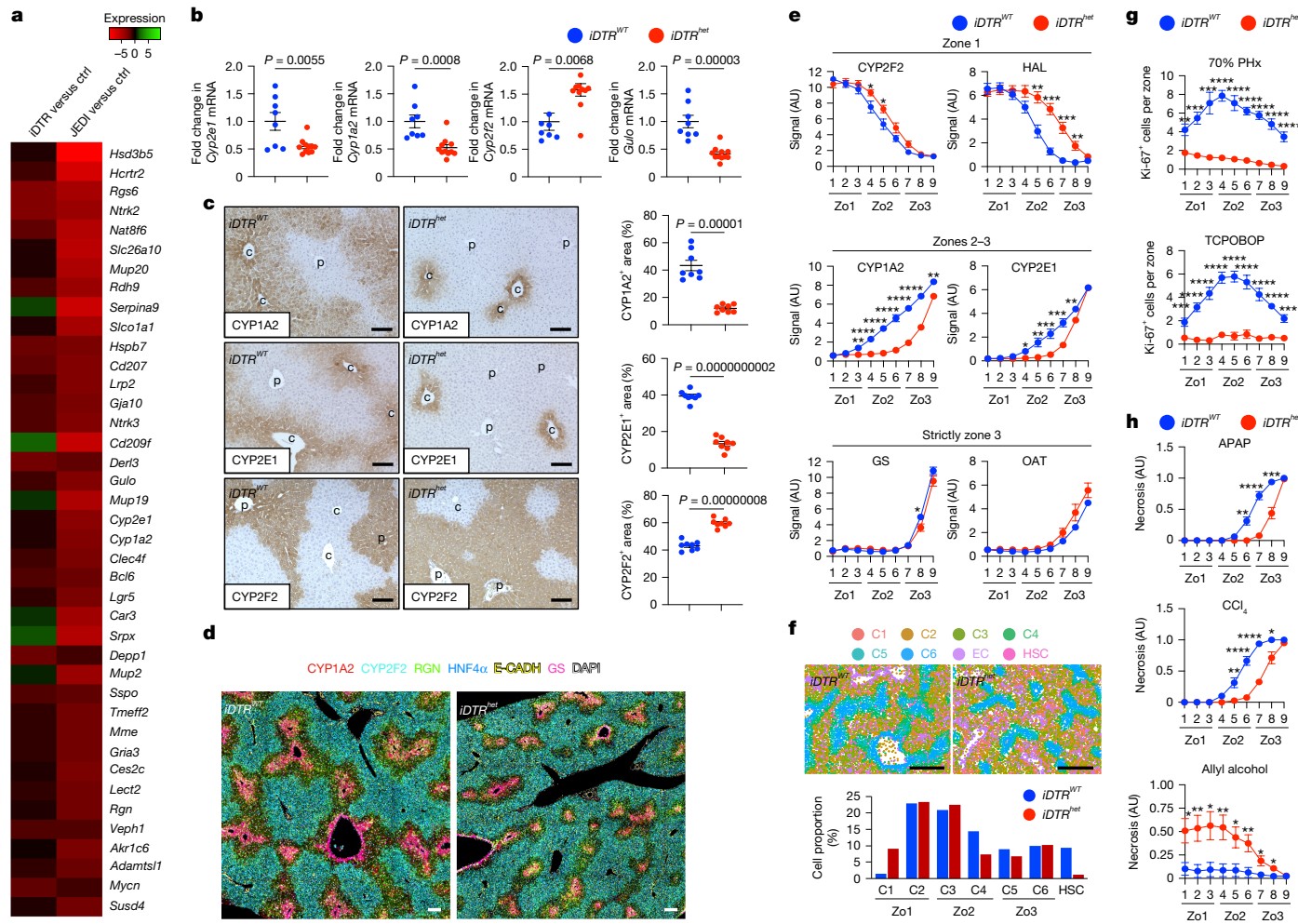

**Fig. 2 | HSCs regulate metabolic zonation and zone-specific injury and proliferation in the liver. a,b**, The top 40 genes downregulated in RNA-seq data from HSC-depleted mice versus controls (ctrl) in the iDTR × *Lrat-cre* and the JEDI models versus controls after subtraction of HSC-enriched genes (**a**), and qPCR confirmation in *iDTR^WT* (*n* = 8) and *iDTR^het* (*n* = 11) mice (**b**) of select genes in livers from the iDTR × *Lrat-cre* model. **c**, CYP2E1, CYP1A2 and CYP2F2 IHC and quantification in *iDTR^WT* (*n* = 8) and *iDTR^het* (*n* = 8) mice 7 days after treatment with diphtheria toxin. c, central vein; p, portal vein. **d**, Multiplex IHC analysis showing significantly altered expression of zonal genes in *iDTR^WT* (*n* = 5) and *iDTR^het* (*n* = 5) mice. **e**, Zonal quantification of the indicated zone 1 (Zo1), zones 2–3 and strictly zone 3 markers from IHC performed in Fig. 2c and

Extended Data Fig. 4g in *iDTR^WT* and *iDTR^het* mice (*n* = 8 per group). **f**, 100-plex spatial transcriptomics for WNT-regulatory, WNT-target and cell marker genes shows differences in zonation patterns and WNT-target genes between *iDTR^WT* (*n* = 1) versus *iDTR^het* (*n* = 1) mice. **g,h**, Zonal quantification of Ki-67+ cells after 70% PHx and TCPOBOP treatment (**g**) or of necrosis after APAP, CCl₄ or allyl alcohol treatment (**h**) in *iDTR^WT* (*n* = 5–8) and *iDTR^het* (*n* = 4–11) mice. Data are mean ± s.e.m. For **b** and **c**, each dot represents one biological replicate. Scale bars, 100 μm (**c** and **d**) and 1 mm (**f**). *P* values were calculated using unpaired two-tailed *t*-tests (**b**, **c**, **e**, **g** and **h**). **P* < 0.05, ***P* < 0.01, ****P* < 0.001, *****P* < 0.0001. AU, arbitrary units.

## HSCs control liver size and zonation through RSPO3

To further investigate the hepatocyte-regulatory roles of HSC-derived RSPO3 in vivo, we generated mice with HSC-specific deletion of *Rspo3* (*Rspo3^ΔHSC*). *Rspo3* deletion in HSCs was highly efficient and did not alter hepatic immune cell composition except for small increases in B cells and dendritic cells (Fig. 3g and Supplementary Information 1b). Importantly, *Rspo3^ΔHSC* mice phenocopied key changes of HSC-depleted mice, including a reduction in the liver–body weight ratio, suppressed WNT-target gene expression in livers and hepatocytes, and an altered zonation with a condensation of the CYP2E1+CYP1A2+RGN+ zones 2–3 and a concomitantly expanded CYP2F2+HAL+ zone 1 without alteration in GS and OAT (Fig. 3g–i, Extended Data Fig. 8a–d and Supplementary Table 5). ECs isolated from *Rspo3^ΔHSC* mice did not display downregulated *Rspo3*, *Wnt2* or *Wnt9b*, excluding indirect effects of HSC-derived RSPO3 through ECs (Extended Data Fig. 8e). Thus, *Rspo3^ΔHSC* mice reproduced key aspects of HSC-depleted, *Lgr4/Lgr5^ΔHep* and *Ctnnb1^ΔHep* mice[24,25,34–39], suggesting HSC-derived RSPO3, hepatocyte-expressed

LGR4 and its downstream target β-catenin as central mediators of the HSC–hepatocyte cross-talk that controls liver size, zonation and function.

As *Rspo3* is expressed in HSCs during development[32], we generated mice with an inducible knockout of *Rspo3* in HSCs through *Pdgfrb-creERT2* (*Rspo3^ΔHSC-ind*) mice for postnatal deletion. *Rspo3* deletion during adulthood in *Rspo3^ΔHSC-ind* mice caused a similar reduction in the liver–body weight ratio and alterations in liver zonation to in *Rspo3^ΔHSC* mice and did not change *Rspo3*, *Wnt2* or *Wnt9b* expression in ECs isolated from these mice (Extended Data Figs. 8f,g and 9a), thereby excluding developmental abnormalities as a cause for these effects. Consistent with our single-nucleus RNA-seq (snRNA-seq) data and the critical role of HSC-derived RSPO3, we did not find a reduction in *Rspo3* mRNA, an altered liver–body weight ratio or changes in zonation in mice with hepatocyte-specific deletion (through AAV8-TBG-cre) or Kupffer cell-specific deletion (through *Clec4f-cre*) of *Rspo3* (Extended Data Figs. 8h,i and 9b,c).

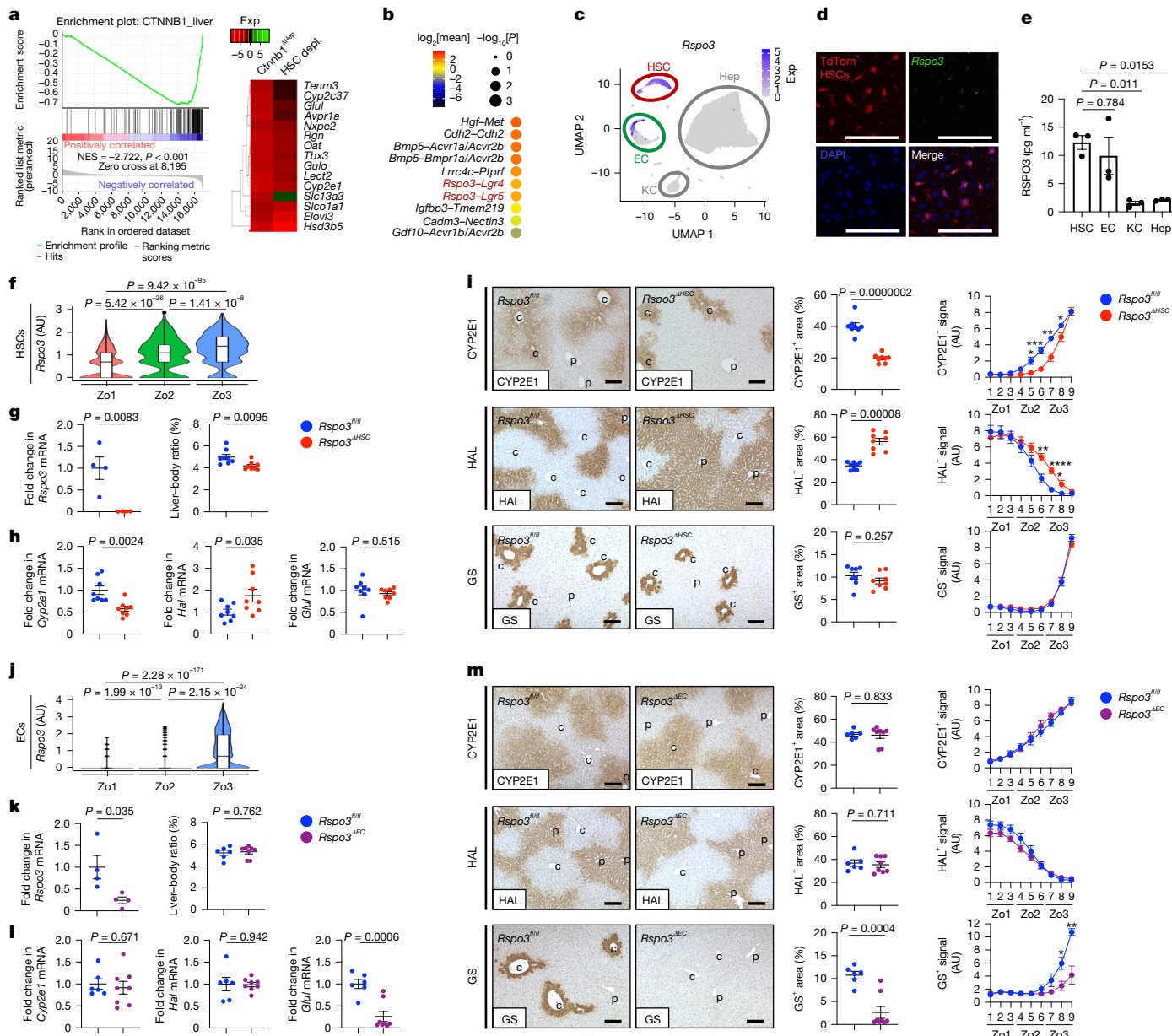

**Fig. 3 | HSC-derived RSPO3 regulates hepatocyte gene expression and liver zonation. a**, GSEA of CTNNB1-regulated genes from RNA-seq data of HSC-depleted (JEDI) versus control livers; and a heat map showing the expression (Exp) of the top 15 downregulated genes from $Ctnnb1^{\Delta Hep}$ versus $Ctnnb1^{fl/fl}$ livers in HSC-depleted versus control livers. **b**, CellPhoneDB analysis showing the top HSC–hepatocyte ligand–receptor interactions in healthy mouse liver snRNA-seq data. $n = 2$ livers. **c**, snRNA-seq analysis of $Rspo3$ expression in healthy mouse liver. $n = 2$. **d**, RNAscope analysis of $Rspo3$ colocalization with TdTom[+] HSCs in $Lrat$-cre × TdTom livers. A representative image of two technical replicates is shown. **e**, RSPO3 ELISA in the supernatants from primary mouse HSCs, ECs, Kupffer cells (KCs) and hepatocytes. $n = 3$ per group. **f**, Analysis of $Rspo3$ expression in HSCs across mouse liver zones using 100-plex spatial transcriptomics data. **g**, The liver–body weight ratio and qPCR analysis of $Rspo3$ mRNA ($n = 4$ per group) in HSCs from $Rspo3^{fl/fl}$ ($n = 8$) and $Rspo3^{\Delta HSC}$ ($n = 8$, 7 male, 1 female) mice. **h,i**, The indicated WNT-target genes determined

by qPCR (**h**) or IHC with morphometric and zone-specific quantification (**i**) in $Rspo3^{fl/fl}$ ($n = 8$) and $Rspo3^{\Delta HSC}$ ($n = 8$, 7 male, 1 female) livers. **j**, Analysis of $Rspo3$ expression in ECs across mouse liver zones using 100-plex spatial transcriptomics data. **k**, $Rspo3$ mRNA in isolated ECs ($n = 4$ per group), and the liver–body weight ratio in $Rspo3^{fl/fl}$ ($n = 6$) and $Rspo3^{\Delta EC}$ ($n = 8$) mice. **l,m**, WNT-target genes determined by qPCR (**l**), and IHC analysis with morphometric and zone-specific quantification (**m**) in $Rspo3^{fl/fl}$ ($n = 6$) and $Rspo3^{\Delta EC}$ ($n = 8$) livers. Data are mean ± s.e.m. Each dot represents one cell (**c**) or one biological replicate (**g–i** and **k–m**). For **d**, **i** and **m**, scale bars, 100 μm. For the violin plots in **f** and **j**, the box plots show the interquartile range (IQR; Q1–Q3) (box limits), the median (centre line), and the minimum (Q1 – 1.5 × IQR) and maximum (Q3 + 1.5 × IQR) values (whiskers). $P$ values were calculated using unpaired two-tailed $t$-tests (**e**, **g–i** and **k–m**) or Wilcoxon rank-sum tests (**f** and **j**). UMAP, uniform manifold approximation and projection.

## ECs control WNT^high hepatocytes through RSPO3

In addition to HSCs, pericentral ECs constitute a second liver cell population with high expression of $Rspo3$[31,40,41], confirmed by scRNA-seq and snRNA-seq, ELISA, immunoblotting and spatial transcriptomics (Fig. 3c,e,j and Extended Data Fig. 6a–c,i). In contrast to HSC-depleted

and $Rspo3^{\Delta HSC}$ mice, constitutive or inducible EC-selective deletion of $Rspo3$ ($Rspo3^{\Delta EC}$ and $Rspo3^{\Delta EC\text{-}ind}$) did not alter the liver–body weight ratio or the expression and zonation of CYP2E1, CYP1A2, RGN, CYP2F2 and HAL, despite being highly efficient (Fig. 3k–m and Extended Data Fig. 9d,e). However, RNA-seq, qPCR and IHC revealed alterations in select WNT-target genes such as $Glul$ and $Oat$, which are

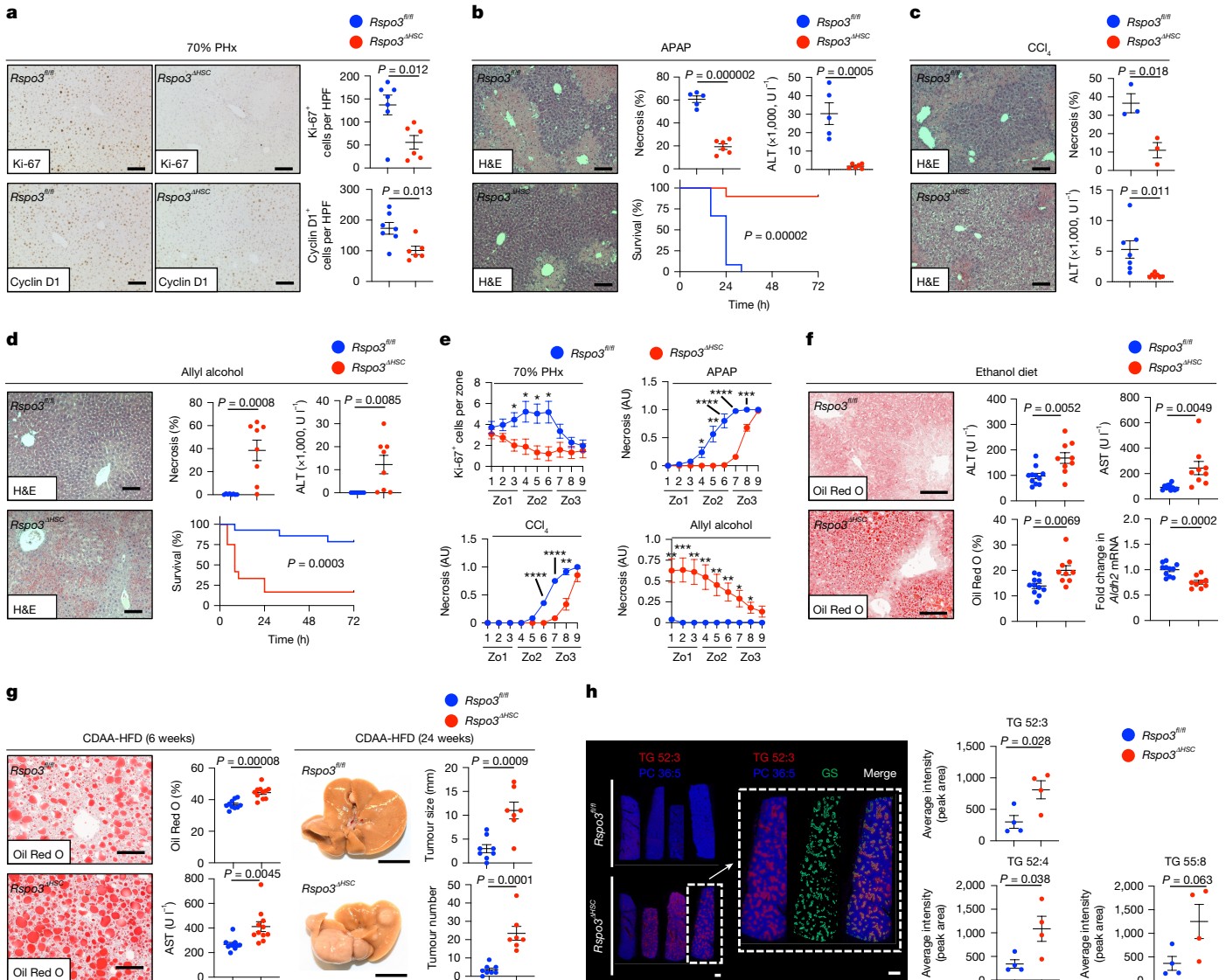

**Fig. 4 | HSC-derived RSPO3 regulates hepatocyte injury, liver regeneration and steatosis. a**, Ki-67 and cyclin D1 IHC from *Rspo3*[fl/fl] (*n* = 7) and *Rspo3*[ΔHSC] (*n* = 6) mice subjected to 70% PHx. **b**–**d**, The necrosis area (*n* = 5 (*Rspo3*[fl/fl]), *n* = 6 (*Rspo3*[ΔHSC])), ALT levels (*n* = 5 (*Rspo3*[fl/fl]), *n* = 6 (*Rspo3*[ΔHSC])) and survival (*n* = 12 (*Rspo3*[fl/fl]), *n* = 10 (*Rspo3*[ΔHSC])) in *Rspo3*[fl/fl] and *Rspo3*[ΔHSC] mice treated with APAP (**b**; 300 mg per kg or 750 mg per kg lethal dose); the necrosis area (*n* = 3 per group) and ALT levels (*n* = 7 per group) in *Rspo3*[fl/fl] and *Rspo3*[ΔHSC] mice treated with CCl₄ (**c**, 0.5 ml kg⁻¹); and the necrosis area (*n* = 8 per group), ALT levels (*n* = 8 per group) and survival (*n* = 14 (*Rspo3*[fl/fl]), *n* = 12 (*Rspo3*[ΔHSC])) in *Rspo3*[fl/fl] and *Rspo3*[ΔHSC] mice treated with allyl alcohol (**d**; 60 mg per kg or 75 mg per kg lethal dose). **e**, Zonal quantification of Ki-67⁺ cells (*n* = 7 (*Rspo3*[fl/fl]), *n* = 6 (*Rspo3*[ΔHSC])) and necrosis in APAP (*n* = 5 (*Rspo3*[fl/fl]), *n* = 6 (*Rspo3*[ΔHSC])), CCl₄ (*n* = 3 per group) and allyl alcohol (*n* = 8 per group) models in *Rspo3*[fl/fl] and *Rspo3*[ΔHSC] mice. **f**, Oil Red O staining and quantification, serum ALT and AST, and qPCR analysis of *Aldh2* mRNA in *Rspo3*[fl/fl] (*n* = 11) and *Rspo3*[ΔHSC] (*n* = 9) mice treated with the Lieber–DeCarli diet. **g**, Oil Red O staining and quantification, the serum ALT and AST (*n* = 10 (*Rspo3*[fl/fl]), *n* = 11 (*Rspo3*[ΔHSC])), representative images and the tumour number and tumour size in *Rspo3*[fl/fl] (*n* = 8) and *Rspo3*[ΔHSC] mice (*n* = 7) that were treated with CDAA-HFD diet for the indicated times. **h**, DESI–MS imaging showing triglycerides (TG; 52:3, red) and phosphatidylcholine (PC; 36:5, blue) in *Rspo3*[fl/fl] and *Rspo3*[ΔHSC] (*n* = 4 per group) mice as well as a representative for localization of TG 52:3 around pericentral zones marked by GS (green) and quantification of TG 52:3, TG 52:4 and TG 55:8 species. Data are mean ± s.e.m. Each dot represents one biological replicate (**a**–**d** and **f**–**h**). Scale bars, 100 μm (**a**–**d**, **f** and **g**), 1 cm (**h**, left), 1 mm (**h**, right). *P* values were calculated using unpaired two-tailed *t*-tests (**a**, **c** and **e**–**h**, and **b** and **d** (top)) or log-rank tests (**b** and **d** (bottom)).

characteristically expressed by the most pericentral WNT[high] hepatocytes, in *Rspo3*[ΔEC] mice (Fig. 3l,m, Extended Data Fig. 9d,e and Supplementary Table 6). Notably, expression of these genes was not altered in HSC-depleted and *Rspo3*[ΔHSC] mice, with a small pericentral rim of WNT[high] hepatocytes—expressing GS, OAT, but also CYP2E1, CYP1A2 and RGN—remaining (Fig. 3h,i and Extended Data Fig. 8b,c). Together, these findings suggest distinct zonal roles of RSPO3-expressing HSCs and ECs, with HSCs regulating WNT activity in the majority of hepatocytes and, thereby, exerting strong effects on liver size and all liver zones with the exception of the most pericentral hepatocyte layers.

## Altered regeneration and injury in *Rspo3*[ΔHSC] mice

We next tested whether *Rspo3*[ΔHSC] mice displayed similar alterations in liver regeneration and injury compared to HSC-depleted mice. After 70% PHx or treatment with TCPOBOP, *Rspo3*[ΔHSC] mice exhibited a lower liver–body weight ratio, significantly reduced Ki-67⁺ and cyclin D1⁺ hepatocytes and late-stage compensatory hyperproliferation (Fig. 4a and Extended Data Fig. 10a,b). Analysis using qPCR and IHC suggested that this effect was mediated by RSPO3-dependent regulation of the WNT target CCND1 (Fig. 4a and Extended Data Figs. 7a and 10b),

a key driver of liver regeneration[23,42]. While the effects of *Rspo3* deletion on liver regeneration were potent, they were not as pronounced as those in HSC-depleted mice. Double knockout of *Rspo3* and *Hgf* in HSCs had a stronger effect on liver regeneration after 70% PHx or TCPOBOP (Extended Data Fig. 10c), suggesting that additional HSC mediators collaborate with RSPO3 to maximally stimulate hepatocyte proliferation.

To investigate the potential role of HSC-derived RSPO3 in toxin-induced liver injury, *Rspo3^ΔHSC* mice were injected with APAP, CCl4 or allyl alcohol. Similar to HSC-depleted mice, *Rspo3^ΔHSC* mice exhibited significant reductions in serum ALT and hepatic necrosis area as well as improved survival after sublethal or lethal doses of pericentral toxins APAP or CCl4 (Fig. 4b,c). Accordingly, liver fibrosis was reduced in *Rspo3^ΔHSC* mice after chronic CCl4 treatment (Extended Data Fig. 10d). Like HSC-depleted mice, *Rspo3^ΔHSC* mice displayed increased injury and mortality after treatment with periportal toxin allyl alcohol (Fig. 4d). Corresponding to the predominantly midzonal alterations in WNT-target genes as a result of a condensed zone 3 and an expanded zone 1 (Fig. 3i and Extended Data Fig. 8c), the alterations in proliferation and liver injury in *Rspo3^ΔHSC* mice were most pronounced in the hepatic midzone (Fig. 4e and Extended Data Fig. 10b), which is pivotal for hepatocyte regeneration[42].

*Rspo3^ΔHSC* mice, subjected to the Lieber–DeCarli model of ALD, displayed increased liver steatosis and injury, alongside decreased expression of the acetaldehyde-metabolizing gene *Aldh2* (Fig. 4f and Extended Data Fig. 10e,f). Similarly, *Rspo3^ΔHSC* mice subjected to a choline-deficient amino-acid-supplemented high-fat diet (CDAA-HFD) model of MASLD displayed increased steatosis, serum ALT, fibrogenic gene expression, fibrosis and tumour formation, with TUNEL-positive cells present in zone 3 (Fig. 4g and Extended Data Fig. 10g,h). Furthermore, when aged, *Rspo3^ΔHSC* mice exhibited more pronounced changes of WNT-target genes and increased fibrogenic gene expression (Extended Data Fig. 10i and Supplementary Table 7), similar to the exacerbated phenotype seen in aged *Ctnnb1^ΔHep* mice[43]. HSCs isolated from *Rspo3^ΔHSC* mice did not display differences in *Acta2*, *Col1a1* and *Col1a2* mRNA, suggesting that RSPO3 does not affect cell-intrinsic mechanisms of HSC activation (Extended Data Fig. 10j). A similar reduction in CCl4-induced liver injury was observed in *Rspo3^ΔHSC-ind* mice (Extended Data Fig. 10k), excluding developmental effects as the cause of altered injury. Together, these data align with the critical role of the β-catenin–WNT pathway in liver injury, promoting APAP- and CCl4-induced injury[34] but protecting from ALD[36,37], MASLD[35,38,39] and ageing-induced injury[43]. By contrast, *Rspo3^ΔEC* or *Rspo3^ΔEC-ind* mice did not show alterations in CCl4-, APAP- or CDAA-HFD-induced liver injury or regeneration after 70% PHx (Extended Data Fig. 10l–o). In conjunction with our findings in HSC-depleted mice, these data suggest that HSCs, through RSPO3, regulate the majority of hepatocytes in the liver, affecting liver size, regeneration and injury. By contrast, EC-derived RSPO3 regulates only the most pericentral WNT^high hepatocytes that have a key role in the liver's glutamine synthesis and ammonia detoxification[44]. In silico metabolomic pathway analysis of transcriptomic data from *Rspo3^ΔHSC*, HSC-depleted and *Ctnnb1^ΔHep* livers revealed many shared alterations, including bile acid metabolism and xenobiotic metabolism, mitochondrial β-oxidation, carnitine shuttling and acyl-CoA hydrolysis (Supplementary Information 2 and Supplementary Table 8). Consistent with the altered zonation, increased susceptibility and possible changes in mitochondrial β-oxidation, desorption ionization (DESI) mass spectrometry (MS) imaging revealed a marked pericentral accumulation of triglycerides in *Rspo3^ΔHSC* mice (Fig. 4h and Supplementary Information 3a,b). Metabolomic analyses also showed significant increases in bile acid levels, including taurocholic acid, tauromuricholic acid and taurochenodeoxycholic acid, in *Rspo3^ΔHSC* mice (Extended Data Fig. 10p–r and Supplementary Table 9). These findings match the role of β-catenin in hepatic bile acid metabolism[35,39,45] as well as mitochondrial function and energy balance[37,38], which have been linked to

increased steatosis and liver injury in mice with hepatocellular deletion of β-catenin signalling components[35,37–39] and in non-hepatic tissues[46,47].

## Dynamic regulation of *Rspo3* in liver disease

Finally, we determined the regulation of RSPO3 and its receptors in clinically relevant contexts. *RSPO3* expression was substantially decreased in HSCs isolated from models of toxic, biliary or MASLD-related liver fibrosis, progressively declining in advanced disease stages, alongside a reduction of RSPO3 protein (Fig. 5a,b and Extended Data Fig. 11a–c). This decline suggested a maladaptive process that may contribute to the loss of hepatocyte function and alterations in liver injury during progression of chronic liver disease (CLD). Conversely, analysis of HSC deactivation during fibrosis regression revealed restored *Rspo3* mRNA levels alongside partially restored *Col1a1* and *Hgf* mRNA levels in HSCs (Fig. 5c and Extended Data Fig. 11d). TGFβ, the most potent inducer of HSC activation, but not inducers of HSC proliferation and inflammation, such as PDGF and IL-1β, substantially reduced *Rspo3* mRNA levels in primary HSCs (Fig. 5d). Notably, *Tgfb2* and *Tgfb3* displayed a periportal-to-pericentral gradient and potently suppressed *Rspo3* mRNA in primary HSCs (Extended Data Fig. 11e). Thus, a periportal suppression of *Rspo3* expression by TGFβ2 and TGFβ3 could contribute to the zonal *Rspo3* gradient in HSCs. Analysis of human snRNA-seq datasets revealed a similar enrichment of *RSPO3* in HSCs and pericentral ECs as in mice and confirmed RSPO3–LGR4 as one of top ligand–receptor pairs mediating HSC–hepatocyte interactions (Fig. 5e,f, Extended Data Fig. 11f,g and Supplementary Tables 10 and 11). *RSPO3* was more abundantly expressed in quiescent cytokine- and growth factor-expressing HSCs (cyHSCs) than in activated myofibroblastic HSCs (myHSCs) (Fig. 5g and Extended Data Fig. 11h) and was potently downregulated by TGFβ1, TGFβ2 and TGFβ3 but not PDGF in human LX-2 HSCs (Fig. 5h and Extended Data Fig. 11e). snRNA-seq analysis revealed a marked and progressive downregulation of *RSPO3* mRNA in HSCs with advancing stages fibrosis in patients with MASLD, alcohol-associated cirrhosis or alcoholic hepatitis (Fig. 5i), whereas *RSPO3* levels were low and remained largely unchanged during disease progression in ECs from the same patients (Extended Data Fig. 11i). The decrease in *RSPO3* mRNA expression with advancing liver disease was confirmed by bulk RNA-seq analysis in additional MASLD cohorts (Fig. 5j and Extended Data Fig. 12a). *RSPO3* mRNA, but not *RSPO1*, *RSPO2* and *RSPO4* mRNA, showed significant positive correlations with WNT-target gene expression in multiple cohorts of patients with MASLD (Fig. 5j and Extended Data Fig. 12b–e). Notably, correlation analyses in human snRNA-seq data directly linked *RSPO3* expression in HSCs to WNT-target genes *CYP1A2* and *CYP2E1* in hepatocytes (Fig. 5k). The reduction of *RSPO3* expression and its correlation with WNT-target genes was confirmed by bulk RNA-seq in patients with ALD (Fig. 5l and Extended Data Fig. 12f,g). Consistent with our functional data in mouse models of MASLD and ALD, we observed that *RSPO3* conferred protection in patients, revealing an association of high *RSPO3* expression with reduced mortality, HCC and liver-related events in patients with MASLD, as well as a trend towards lower mortality in patients with alcohol-associated hepatitis (Fig. 5l,m and Extended Data Fig. 12h,i). Together, these findings suggest that the dynamic regulation of RSPO3 in HSCs during CLD progression and regression may affect hepatocyte functions and disease outcomes.

## Discussion

With fibrosis representing the primary determinant of CLD outcomes[48], the field has focused on the paradigm of HSCs as the liver's primary fibrogenic cell population[1,3–5]. The current study amends this fibrocentric concept by demonstrating fibrosis-independent roles for HSCs through RSPO3, regulating critical parameters that affect CLD progression and outcomes, including metabolic hepatocyte functions, liver

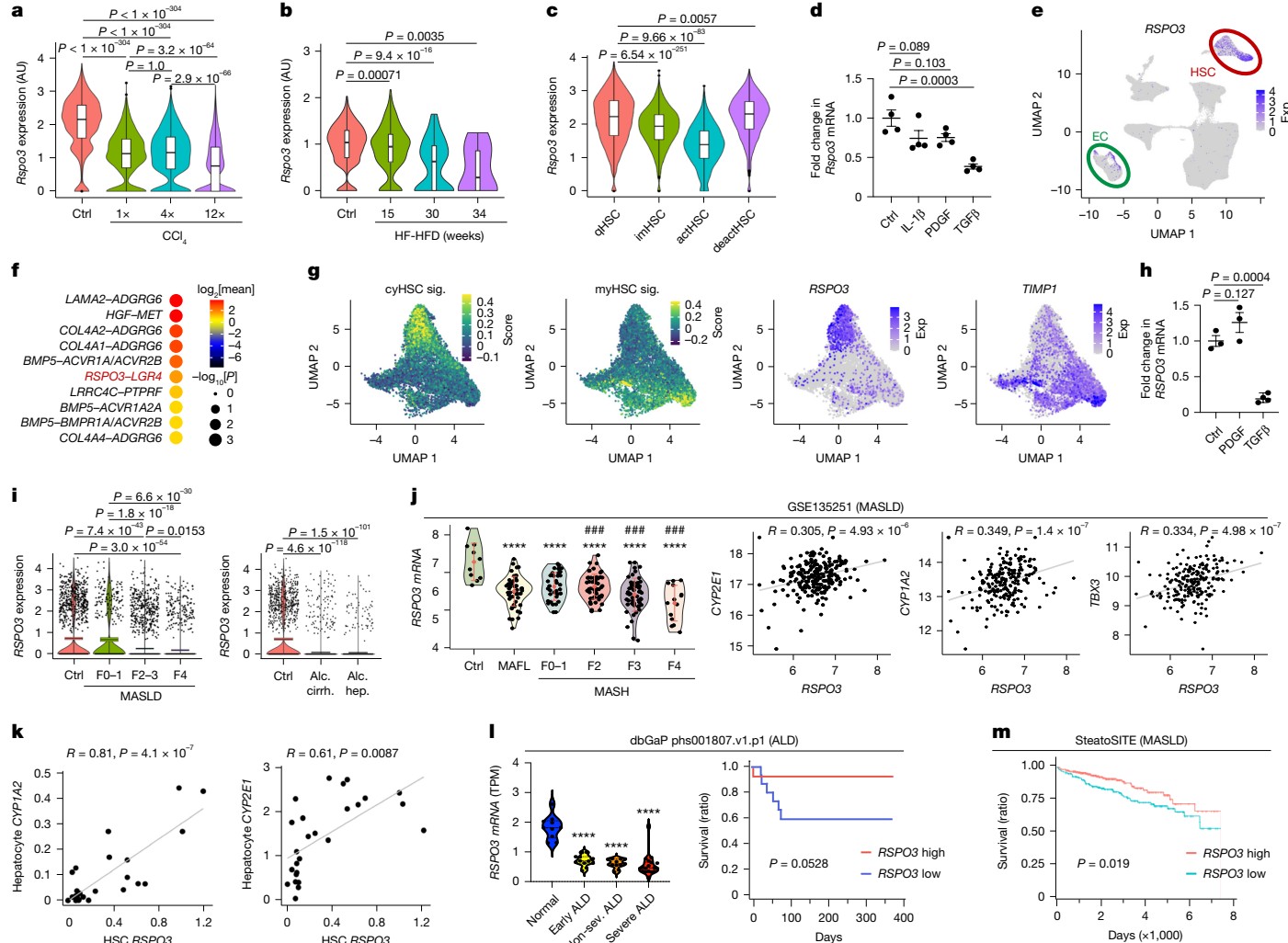

**Fig. 5 | Dynamic regulation of RSPO3 in liver disease. a,b**, scRNA-seq analysis of *Rspo3* mRNA of HSCs from CCl$_4$-treated (**a**) or high-fat high-fructose diet (HF-HFD)-treated (**b**) mice. **c**, scRNA-seq analysis of *Rspo3* mRNA from quiescent (qHSC), intermediate-active (imHSC), activated (actHSC) and deactivated (deactHSC) mouse HSCs (from ref. 61). **d**, qPCR analysis of *Rspo3* in quiescent mouse HSCs treated with the indicated cytokines. $n = 4$ per group. **e**, snRNA-seq analysis of *RSPO3* mRNA expression (Exp) in human liver. $n = 6$. **f**, CellPhoneDB analysis showing the top HSC–hepatocyte interactions in snRNA-seq data from human livers. **g**, snRNA-seq analysis of *RSPO3* mRNA in human cyHSCs and myHSCs. **h**, qPCR analysis of *RSPO3* in PDGF- and TGFβ-treated LX-2 human HSCs. $n = 3$ per group (control and PDGF) and $n = 4$ (TGFβ). **i**, snRNA-seq analysis of *RSPO3* mRNA in HSCs from healthy control individuals (Ctrl) and patients with MASLD, alcoholic cirrhosis (Alc. cirrh.) or alcoholic hepatitis (Alc. hep.). Data are mean ± 95% confidence intervals. **j**, *RSPO3* mRNA in different stages of MASLD and correlation with the WNT-target genes *CYP2E1*, *CYP1A2* and *TBX3* in the GSE135251 cohort. **k**, The correlation between HSC *RSPO3* and hepatocyte

*CYP1A2* and *CYP2E1* expression in snRNA-seq data of healthy individuals and patients with MASLD and ALD ($n = 25$). **l**, *RSPO3* mRNA in different stages of ALD and survival stratified by *RSPO3* expression in the dbGaP phs001807.v1.p1 ALD cohort. Non.-sev., non-severe; TPM, transcripts per million. **m**, Survival by *RSPO3* expression in the SteatoSITE MASLD cohort. Data are mean ± s.e.m. Each dot represents one cell (**a–c**) or one biological replicate (**d** and **h**). ****$P < 0.0001$ versus control or normal; ###$P < 0.001$ versus F0–1. In the violin plots, the box plots show the IQR (box limits), the median (centre line), the minimum (Q1 − 1.5 × IQR) and maximum (Q3 + 1.5 × IQR) values (whiskers), and outliers (individual dots). $P$ values were calculated using two-way analysis of variance (ANOVA) with Tukey's multiple-comparison test (**a–c**, **i** and **l** (left)), one-way ANOVA with Dunnett's multiple-comparison test (**d** and **h**), Wilcoxon ranksum tests (**j**) or log-rank tests (**l** (right) and **m**). Correlations were evaluated by the Pearson correlation coefficient (**j** and **k**). MASH, metabolic dysfunction-associated steatohepatitis.

regeneration and hepatocyte death[2,49,50]. These hepatocyte-regulatory roles of HSC-derived RSPO3 are highlighted by our functional studies in *Rspo3*$^{ΔHSC}$ mice and the association of *RSPO3* expression with patient outcomes. However, further studies are needed to confirm altered RSPO3 protein levels in patients and their associations with outcomes. Our data suggest that the positive regulation of hepatocyte function by RSPO3 from quiescent HSCs is gradually lost during CLD progression, accompanied by a shift towards a disease-promoting activated HSC state. The critical impact of HSC-derived RSPO3 on liver size, zonation, metabolism, regeneration and injury matches findings in mice with hepatocyte-specific deletion of β-catenin or LGR4/5[24,25,34–39], rendering

RSPO3–LGR4-mediated HSC–hepatocyte communication a gatekeeper of liver function in health and disease.

Evolutionarily, the liver is geared for rapid regeneration and restoration of function after toxic injury to ensure organismal survival. The seemingly opposing functions of RSPO3 in acute toxic liver and chronic metabolic injury align with the functions of its downstream target β-catenin in hepatocyte injury[34–39]. Notably, the functions of hepatic cytochrome P450 oxidases are generally beneficial, protecting other organs and thereby promoting organismal health. Massive cell death after exposure to P450-activated hepatotoxins such as APAP or CCl$_4$ probably represents a new evolutionary development as these

are man-made toxins that do not exist in nature. Owing to the lack of evolutionary pressure, the liver may only be geared for eliminating smaller amounts and less toxic P450-metabolized natural substances. Deletion of *Rspo3* in HSCs or inhibition of RSPO3 by blocking antibodies[51] inhibited CCl₄-induced fibrosis in mice, but P450-activated toxins like APAP are not considered clinically relevant causes of CLD and liver fibrosis as they usually trigger only acute liver disease. By contrast, MASLD and ALD, against which RSPO3, LGR4/5 and their downstream target β-catenin protect[35,39], represent the commonest forms of CLD and liver fibrosis in patients. Thus, the overall functions of RSPO3 in the liver seem to be protective. The reduction in hepatic RSPO3 expression appears to be part of a maladaptive process linked to the prolonged HSC activation in CLD[5], contributing to increased metabolic injury, reduced regeneration and, thereby, adverse outcomes. Conversely, restored *RSPO3* expression during HSC deactivation[52,53] or by therapeutic interventions might help to regain hepatocyte functions. Our findings underscore the complex multicellular nature of interactions that maintain liver homeostasis and hepatocyte functions and suggest that shifts in HSC-expressed RSPO3 contribute to a gradual switch from liver homeostasis with intact regeneration and metabolism to fibrosis with impaired regeneration and altered metabolism. Accordingly, liver zonation is perturbed in CLD, including MASLD[54,55]. Moreover, many hepatocyte functions and characteristic features like their predominant proliferation in zone 2 may not be intrinsic but regulated by the niche surrounding hepatocytes, with HSCs as key actors in the niche. Our data amend the purely angiocrine model of liver zonation[29,41,56,57] and suggest that liver zonation and functions are regulated by a 'ménage à trois' of WNT-secreting ECs, RSPO3-secreting HSCs and hepatocytes expressing the R-spondin receptors LGR4 and LGR5[24]. The strictly pericentrally expressed EC-derived RSPO3 selectively regulates WNT^high hepatocyte layers, which control glutamine metabolism and thereby prevent ammonia toxicity and encephalopathy[44] but not liver size, regeneration, toxic or metabolic injury. Previous studies, using global knockout or overexpression, have found that RSPO3 is an important regulator of liver zonation and hepatocyte proliferation, but did not dissect its cell-specific functions and its key role in the HSC–hepatocyte cross-talk[24,41,55]. These cell-specific expression patterns of *Rspo3* underscore the importance of spatial organization in liver. It is likely that the pericentral to periportal gradient of *Rspo3* expression in HSCs is not only explained by zonal *Tgfb2* and *Tgfb3* expression but also by the liver's well-characterized oxygen and nutrient gradients[9,58]. While HSCs interact with other cell populations, such as Kupffer cells and ECs, through additional mediators including GDF2 and BMP10[59], our findings in HSC-depleted and *Rspo3*^ΔHSC mice underscore the central role of HSCs interacting with and regulating hepatocytes. Moreover, HSC mediators besides RSPO3, such as HGF, but possibly also neurotrophin-3[19] or epiregulin[60] may be involved in additional aspects of HSC–hepatocyte cross-talk and—together with ECs[29,59]—collaboratively control hepatocyte health and functions.

Owing to their key role in fibrosis and CLD outcomes, HSCs are considered to be a potential therapeutic target[5]. The positive effects of HSCs on hepatocyte metabolism, regeneration, MASLD and ALD through RSPO3 suggest that reverting HSCs to quiescence or increasing RSPO3 levels may represent a more potent therapeutic approach for most liver diseases compared with inhibiting or killing activated HSCs. Integrating the dichotomous roles of HSCs in hepatocyte protection and fibrogenesis into current therapeutic concepts for CLD may allow simultaneous inhibition of fibrogenesis and restoration of liver function.

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

## Methods

### Human specimens

Human liver tissues for the single-nucleus sequencing were obtained from different sources as shown in Supplementary Table 10. Liver samples were collected under IRB-approved protocols at the University of Pittsburgh (IRB protocol 19120198), Johns Hopkins University School of Medicine (IRB protocol 00107893), the University of Kiel, Germany (Ethikkommission der Medizinischen Fakultät der Universität Kiel, D425/07, A111/99) or obtained from LifeNet Health, an organ procurement organization (operating under the Anatomical Gift Act). All patients provided written informed consent. Liver samples for alcoholic hepatitis, alcohol-associated cirrhosis and MASLD-associated cirrhosis were obtained from liver explants in patients undergoing liver transplant. All other liver samples from living donors were obtained intraoperatively in patients in whom an intraoperative liver biopsy was indicated on clinical grounds such as during scheduled liver resection, exclusion of liver malignancy during major oncologic surgery or assessment of liver histology during bariatric surgery. The samples were frozen immediately in liquid nitrogen ensuring an ex vivo time of less than 40 s in all cases. Patients with evidence of viral hepatitis or haemochromatosis were excluded in all groups, and patients with alcohol consumption >20 g per day (for women) and >30 g per day (for men) were excluded in the normal and MASLD groups. Liver sections stained by H&E, Sirius Red and/or Trichrome were reviewed by a board-certified pathologist.

### Animal studies and strains

All animal procedures were performed with approval by the Columbia University Institutional Animal Care and Use Committee (protocols, AC-AABQ5565, AC-AABQ5566), the local institutional or the Vanderbilt University Institutional Animal Care and Use Committee (protocol M2000054-01) and in accordance with the Guide for the Care and Use of Laboratory Animals; or with approval from the governmental animal care and use committees Karlsruhe, Germany (in accordance with German national guidelines on animal welfare and the regulations of the regional council Karlsruhe under permit number G-251/20). Randomization and blinding were done as described in the Reporting Summary. Mice were housed in the Irving Cancer Research Center at Columbia University and in the Central Animal Facility of the German Cancer Research Center (DKFZ) Heidelberg. They were fed with a standard mouse diet (ad libitum water and food access) with a constant temperature of 21–24 °C, 45–65% humidity and under a 12 h–12 h light–dark cycle. *Lrat-cre* mice have been described previously[4]. *Clec4f-cre* mice (Jax, 033296). *Rspo3*-floxed mice (Jax, 027313; this strain was used for crosses with *Lrat-cre* or *Lyve1-cre*), *Pdgfbr*-floxed mice (Jax, 010977), *Rosa26-lox*-stop-*lox*-tdTomato (TdTom) mice (Jax, 007908), *Rosa26-lox*-stop-*lox*-HBEGF (iDTR) mice (Jax, 007900), *Wls*-floxed mice (Jax, 012888), *Lyve1-cre* (Jax, 012601) and *Pdgfrb-P2A-creERT2* mice (Jax, 030201) were obtained from the Jackson Laboratory. *Cdh5-creERT2* mice[62] as well as *Mx1-cre* mice, *Col1a1*-floxed mice, *Hgf*-floxed mice and *Tgfbr1*-floxed mice[63] have been described previously. *Rspo3*-floxed mice (this strain was used for crosses with *Pdgfrb-P2A-creERT2* mice or *Cdh5-creERT2*) have been described previously[64]. JEDI mice have been described previously[19]. All mouse strains, except for *Wls*-floxed mice (backcrossed 2× to C57BL/6) were backcrossed with C57BL/6J mice more than five times. Male mice were used between 7 and 42 weeks of age with the exception of few experiments that included female mice, as detailed in each figure legend. To induce HSC- and EC-specific deletion of *Rspo3*, mice were treated with 2 mg tamoxifen, dissolved in 100 µl corn oil, through oral gavage for 5 consecutive days at 8–12 weeks of age. Mice were kept for a 1-week washout period before experimental use. For the deletion of *Rspo3* in hepatocytes, *Rspo3*-floxed mice were intravenously administrated with 10[11] genomic copies per mouse of AAV8-TBG-cre (Addgene, 107787-AAV8) or AAV8-TBG-Null (Addgene, 105536-AAV8) diluted in saline solution. For the deletion of *Col1a1* in the liver, *Mx1-cre*[neg]*Col1a1*-floxed and *Mx1-cre*[pos]*Col1a1*-floxed mice received polyI:C (GE Healthcare, ww27-432-01, intraperitoneal, 10 µg g[−1]) three times every 3 days. Samples sizes for animal experiments differed between models and were based on estimates on the variability of specific models.

### HSC depletion, liver regeneration and liver injury models

To deplete HSCs, *Lrat-cre*[+]TdTom[+]DTR[+] mice or *Lrat-cre*[+]TdTom[+]DTR[−] littermates were injected with 0.25–2.0 ng kg[−1] diphtheria toxin (DT; Sigma-Aldrich, D0564) as previously described[14] and were used for experiments 7 days after DT injection unless otherwise specified. For some experiments, livers from mice with HSCs depleted using the JEDI model, described previously[19], were used for analysis. For the ethanol-induced liver injury model, mice were treated with Lieber–DeCarli '82 ethanol-containing and control diets (BioServ, F1258, F1259). The ethanol diet was introduced gradually over 5 days, for a final concentration of 5%, and maintained for the indicated duration according to manufacturer's protocol. To model metabolic dysfunction-associated steatotic liver injury, mice were fed a CDAA-HFD diet (Research Diet, A06071302) for 6 weeks. To induce MASLD-associated liver cancer, mice were fed the CDAA-HFD for 24 weeks. Tumours did not exceed the size limit of 20 mm in our IRB protocol. To model toxic liver injury, mice were treated with either CCl$_4$ or APAP. CCl$_4$ (Sigma-Aldrich, 319961) was dissolved in corn oil at ratio of 3:1 and injected either intraperitoneally at 0.5 ml kg[−1] or given by gavage at 1.6 g kg[−1]. To induce severe liver fibrosis, CCl$_4$ was dissolved in corn oil (400 µg µl[−1]) and delivered through oral gavage at a concentration of 1.6 g kg[−1] body weight. Mice were treated twice per week for up to 20 weeks. APAP (Sigma-Aldrich, A5000) was dissolved in warm 0.9% NaCl and injected intraperitoneally into mice after overnight starvation at 300 mg kg[−1] (sublethal dose) to assess liver injury or at 750 mg kg[−1] (lethal dose) to determine survival. Allyl alcohol (AlOH, Sigma-Aldrich, 240532) was dissolved in 0.9% NaCl and injected intraperitoneally into mice at 60 mg kg[−1] (sublethal dose) to assess liver injury or at 75 mg kg[−1] (lethal dose) to determine survival. To induce liver regeneration, mice were either subjected to 70% PHx or treatment with constitutive androstane receptor ligand, 1,4-bis(2-(3, 5-dichloropyridyloxy))benzene (TCPOBOP). 70% PHx was performed according to a previously published protocol[65] under isoflurane and buprenorphine anaesthesia, and mice were euthanized 48 h later unless indicated otherwise. TCPOBOP (Sigma-Aldrich, T1443) was injected intraperitoneally at a dose of 3 mg kg[−1] and the mice were euthanized 48 h later. For *Rspo3* rescue experiments, HSC-depleted or control mice were intravenously injected with 0.5 × 10[11] genomic copies per mouse of AAV8-CMV-Rspo3 (Vector Biolabs, AAV-271188) or AAV-8-CMV-EGFP (Addgene, 105530-AAV8).

### Cell isolation and cell culture

Primary mouse hepatocytes were isolated as described previously[14] using 15–30-week-old LSL-TdTom[+] C57BL/6 mice that had been injected with 1 × 10[11] GC AAV8-TBG.PI.Cre.rBG (Addgene, 107787) 7 days before isolation. Primary hepatocytes were plated and cultured in serum-free William's E medium (Gibco, 12551-032) supplemented with hepatocyte supplement (Gibco, A13448), 10 µM dexamethasone (Gibco, A13449), 10% FBS, gentamicin and antibiotic–antimycotic (Gibco, 150062) as described previously[14,66]. Primary HSCs were isolated from 9–10 months old male BALB/c mice as described previously[67]. HSCs were seeded in 12-well plates and incubated in Dulbecco's modified Eagle's medium (DMEM, Gibco, 11965092) supplemented with 10% FBS, gentamicin and antibiotics. After 4 h, the medium was changed to a medium containing 0.5% FBS, followed by treated with TGFβ1 (2.5 ng ml[−1], R&D systems, 240B), PDGF-BB (20 ng ml[−1], R&D systems, 220BB) or IL-1β (5 ng ml[−1] R&D systems, 401ML). After 24 h, cells were collected and processed for RT–qPCR. For HSC–hepatocyte co-culture, HSCs were added either directly to the tissue culture well before hepatocyte

plating for contact-dependent co-culture, or into Transwell inserts (Corning, 353180) for contact-independent coculture, both at a hepatocyte:HSC ratio of 5:1. For some experiments, RSPO3 neutralizing antibody (ProteoGenix, PX-TA1446) or isotype control antibody (ProteoGenix, PTX17885) was added to the medium at a concentration of 100 nM. After 24 h of culture, EdU was added to the medium for another 24 h, followed by EdU staining according to the manufacturer's instructions (Thermo Fisher Scientific, C10637). Images were captured using an Olympus IX71S1F-3 microscope and analysed using ImageJ software. Cells co-cultured in the absence of EdU were also collected for RT−qPCR analysis. The human HSC line LX-2[68] was serum-starved overnight and treated with TGFβ1 (2.5 ng ml$^{-1}$, R&D systems, 240B), TGFβ2 (2.5 ng ml$^{-1}$, R&D systems, 302-B2-002), TGFβ3 (2.5 ng ml$^{-1}$, R&D systems, 8420-B3-005), PDGF-BB (20 ng ml$^{-1}$, R&D systems, 220BB) or IL-1β (5 ng ml$^{-1}$ R&D systems, 401ML) for evaluation by RT−qPCR. The mouse hepatocyte line AML12, obtained from the American Type Culture Collection, was cultured in DMEM (Thermo Fisher Scientific, 11965118) with 10% (v/v) antibiotic−antimycotic (Gibco, 150062) and 10% (v/v) fetal bovine serum (FBS; GeminBio, 900-108) at 37 °C under 5% $CO_2$. Cell lines were regularly screened for mycoplasma contamination. Recombinant mouse RSPO3 (R&D 3500-RS) was added to the culture medium for 24–48 h at the indicated concentration. Proliferation was determined by WST-1 (Roche, 11644807001) and WNT-dependent gene expression was determined using RT−qPCR. Liver ECs were isolated from mice using a protocol similar to the HSC isolation described above but using liver perfusion medium (Gibco) and liver digestion medium (Gibco) supplemented with 20 μg ml$^{-1}$ Liberase (Roche) at 3 ml min$^{-1}$ for 5 min and purification of EC from the non-parenchymal cell fraction using mouse CD146 MicroBeads (Miltenyi Biotec, 130-092-007) and LS columns. Kupffer cells were purified from the non-parenchymal cell fraction after liver perfusion as described above, using magnetic mouse F4/80 MicroBeads (Miltenyi Biotec, 130-110-443) and LS columns (Miltenyi Biotec) according to the manufacturer's instructions.

## IHC, immunofluorescence and histological determination of liver fibrosis and steatosis

Liver samples were fixed with 10% formalin for paraffin-embedded blocks or with 4% paraformaldehyde for frozen blocks. Liver sections were stained with antibodies against Ki-67 (Abcam, ab16667), cyclin D1 (Abcam, ab134175), CYP1A2 (Santa Cruz, sc-53241), CYP2E1 (Abcam, ab28146), RGN (Thermo Fisher Scientific, PA5-56057), GS (Abcam, ab176562), OAT (antibodies.com, A15120), CYP2F2 (Santa Cruz, sc-374540), HAL (Sigma-Aldrich, HPA038547), E-cadherin (Cell Signaling, 3195), HNF4α (Thermo Fisher Scientific, MAI-199) or Na−K ATPase (Abcam, ab7671). Positive areas for Ki-67, cyclin D1, CYP1A2, CYP2E1, RGN, GS, OAT, CYP2F2 and HAL were analysed using ImageJ. Multiplex immunostaining was performed as previously described on 2-μm-thick formalin-fixed paraffin-embedded mouse liver sections[69]. The antibody elution buffer was prepared by mixing 675 μl distilled water, 125 μl 0.5 M Tris-HCl pH 6.8, 200 μl 10% (w/v) sodium dodecyl sulfate, and 8 μl 2-mercaptoethanol. The acquired images were processed and analysed using FIJI (v.2.14.0)[70], ilastik (v.1.3.3post3)[71] and CellProfiler (v.4.2.1)[72] as described previously[69]. Zone-specific hepatocyte marker expression was quantified using the FIJI profile function on a portal−central vein axis, dividing the axis into nine equal sectors. Quantification was performed in four representative areas of interest for each sample. The same procedure was used to determine zonal Ki-67+ cells in Ki-67-stained liver sections, but quantifying positive cells within each sector using ImageJ. Zonal necrosis was evaluated using the above-described division of the portal−central axis into nine sectors, followed by sector-specific determination of the necrotic area assigning a percentage of necrosis (0%, 25%, 50%, 75% and 100%) to each area, based on H&E images. For determination of liver fibrosis, paraffin liver sections were stained with Picrosirius Red solution as previously

described[73]. Frozen liver sections of 8 μm were stained in Oil Red O (Sigma-Aldrich, O9755) for 10 min. After being washed in distilled water, the sections were counterstained with Mayer's haematoxylin for 3 min and mounted in aqueous mounting. All pictures were captured on an Olympus IX 71S1F-3 microscope coupled to a QImaging Retiga camera using QCapture Suite Plus (v.3.1.3.10) and the images were analysed using Adobe Photoshop or ImageJ software. For some analyses, images were scanned on a Leica SCN400 slide scanner with a Scanner Console (v.102.0.7.5) and quantified using ImageJ.

## Determination of liver injury

Liver injury was assessed by determination of serum ALT and serum AST activity. For this, samples were measured either at the Columbia University Institute of Comparative Medicine laboratory or at the analysis centre at the University Clinic of Heidelberg. Samples were diluted with 0.9% NaCl or PBS as needed. For some experiments, the necrotic areas were determined in H&E liver sections and quantified by ImageJ software.

## Immunoblotting

Proteins were extracted from liver tissue using RIPA buffer containing anti-protease (Complete, Roche) and anti-phosphatase (PhosSTOP, Roche). After adding Laemmli buffer, sonication and boiling at 95 °C, samples were loaded and run on SDS−PAGE gels and transferred onto nitrocellulose membranes (Sigma-Aldrich) using a semi-dry blotting system (Bio-Rad). The following antibodies were used: anti-ALDH2 (Proteintech, 15310-1-AP; 1:10,000), anti-RSPO3 (Proteintech, 17193-1-AP; 1:2,000), anti-GAPDH (Sigma-Aldrich, G9295; 1:75,000), anti-β-actin (Sigma-Aldrich, A3854, 1:10,000), and HRP anti-rabbit (Santa Cruz, sc-2004; 1:2,000). Blots were visualized using ultrasensitive enhanced chemiluminescent substrate (Thermo Fisher Scientific, 34094) on a FluorChem M System instrument (ProteinSimple) and quantified using FIJI.

## ELISA

RSPO3 protein concentrations were quantified in homogenized mouse liver tissues using the DuoSet ELISA kit (R&D Systems). For this, lysates from snap-frozen liver samples were collected from the supernatant of homogenized tissue followed by centrifugation at 1,000$g$ for 20 min and adjusted to 100 mg ml$^{-1}$. RSPO3 protein was also determined in the supernatants from cultured primary mouse liver cells using a sandwich ELISA kit (LSBio) according to the manufacturer's protocol. For this, supernatant samples were collected 24 h after 0.1 million cells were seeded in a 12-well plate with DMEM supplemented with 10% FCS, 1% penicillin−streptomycin and 50 mg ml$^{-1}$ gentamycin, and centrifuged at 1,000$g$ for 20 min. ELISAs were read on the iMark Microplate Reader (Bio-Rad).

## Flow cytometry analysis of immune cells

Flow cytometry analysis of the lymphocytic and myeloid liver cell population was performed as previously described[14,63]. Briefly, liver tissues were mechanically homogenized followed by an enzymatic digestion with 1 mg ml$^{-1}$ of collagenase A (Roche, 10103578001) and 0.5 μg ml$^{-1}$ DNase I (Roche, 10104159001) in isolation buffer (RPMI 1640, 5% FBS, 1% L-glutamine, 1% penicillin−streptomycin and 10 mM HEPES) for 45 min at 150 rpm at 37 °C. Cells were filtered through a 100-μm cell strainer, washed and separated in two parts to analyse the myeloid and the lymphocytes cell subsets. For the latter, cells were loaded onto a Percoll gradient (67% overlay with 40%) followed by red blood cell lysis using ammonium-chloride-potassium buffer and stained. Cells were incubated with Ghost dye red 780 (Tonbo Biosciences) to exclude dead cells and anti-CD16/32 (Tonbo, 2.4G2, 1:200) before staining. The following extracellular antibodies were included: anti-CD45 (BD and BioLegend, 30-F11, 1:400), anti-CD19 (Tonbo, 1D3, 1:200), anti-CD3e (Tonbo, 145-2C11, 1:400), anti-CD4 (BD, RM4-5, 1:400), anti-CD8a

(Tonbo, 53-6.7, 1:400), anti-NK1.1 (BD, PK136, 1:300), anti-CD11b (BD, M1/70, 1:500), anti-CD11c (BD, HL3, 1:200), anti-F4/80 (Tonbo, BM8.1, 1:500), anti-Ly6C (BioLegend, HK1.4, 1:500), anti-Ly6G (BioLegend, 1A8, 1:500), anti-B220 (BD, RA3-6B2, 1:200), anti-CD44 (BioLegend, IM7, 1:200), anti-CD64 (BioLegend, X54-5/7.1, 1:200), anti-CD80 (Tonbo, 16-10A1, 1:200), anti-CD86 (BD, GL1, 1:200), anti-VSIG4 (eBioscience, NLA14, 1:200) and anti-MHCII (Tonbo, M5/114.15.2, 1:400). The following intracellular antibodies were included: anti-CD3e (BD, 145-2C11, 1:400), anti-TCRβ (BD, H57-597, 1:300), anti-FOXP3 (eBioscience, FJK-16s, 1:300), anti-Ki-67 (Thermo Fisher Scientific, SolA15, 1:200) and anti-granzyme-B (BioLegend, QA16A02, 1:200). Cells were fixed using the FOXP3/transcription factor staining buffer set (Tonbo) according to the manufacturer's protocol. The samples were analysed using the BD LSR Fortessa cell analyser. Flow cytometry analysis was performed using FlowJo (v.10.10.0).

## CYP2E1 activity assay

CYP2E1 activity was analysed in liver microsomes as previously described[74] with minor modifications. For microsome preparation, liver tissue was dounce homogenized in 50 mM Tris, 150 mM KCl, 2 mM EDTA buffer containing PhosSTOP (Roche, 59124500) and cOmplete Mini (Roche, 57350900). The liver homogenate was centrifuged at 6,000$g$ for 5 min, followed by a second centrifugation of the supernatant centrifuge at 12,000$g$ for 10 min. CaCl$_2$ was added to the supernatant to a final concentration of 8 mM, followed by centrifugation at 211,000$g$ for 20 min. After removal of the supernatant, the pellet was resuspended in KCl-Tris-EDTA buffer, and again centrifuged at 211,000$g$ for 20 min. The pellet was resuspended in 0.1 ml containing 100 mM KPi, pH 7.2, 0.2 mM PNP at 37 °C for determination of CYP2E1 using the PNP method as described previously[74], using a Varioskan LUX spectrophotometer (Thermo Fisher Scientific) spectrophotometer at 510 nm.

## RNA isolation and RT–qPCR

Liver tissue was homogenized in a Tissuelyser (Qiagen) in Trizol and purified using chloroform, followed by isolation of total RNA using RNA isolation kits (Qiagen, Roche or Sigma-Aldrich). Total RNA from cells was isolated directly using RNA isolation kits. After quantification using a Nanodrop ND-1000 spectrophotometer, RNA was reverse-transcribed using TaqMan reverse transcription reagents (Applied Biosystems, 4368813). qPCR was run on an Applied Biosystems QuantStudio 5 Real-Time PCR system (Applied Biosystems) using PerfeCTa FastMix II buffer (Quantabio, 95120) and the following probes (Thermo Fisher Scientific): 18S (Hs99999901_s1), *Acta2* (Mm001546133_m1), *Aldh2* (Mm0047763_m1), *Ang* (Mm00833184_S1), *Avpr1a* (Mm00444092_m1), *Axin2* (Mm00443610_m1), *Ccl2* (Mm00441242_m1), *Ccl3* (Mm00441259_g1), *Ccl4* (Mm00443111_m1), *Ccl5* (Mm01302427_m1), *Ccnd1* (Mm00432360_m1), *Chrna4* (Mm00516561_m1), *Col1a1* (Mm00801666_g1), *Col1a2* (Mm00483888_m1), *Colec11* (Mm01289834_m1), *Cyp1a2* (Mm00487224_m1), *Cyp2e1* (Mm00487224_m1), *Cyp2f2* (Mm00484087_m1), *Emr1* (Mm00802530_m1), *Glul* (Mm00725701_s1), *Gulo* (Mm00626646_m1), *Hal* (Mm00456709_m1), *Hand2* (Mm00439247_m1), *Hsd3b5* (Mm00657677_mH), *Il1a* (Mm00439620_m1), *Il1b* (Mm00434228_m1), *Lect2* (Mm00521920_m1), *Lox* (Mm00495386_m1), *Mki67* (Mm01278617_m1), *Oat* (Mm00497544_m1), *Pdgfrb* (Mm00435546_m1), *Rgn* (Mm00485711_m1), *Rspo3* (Mm00661105_m1, Mm01188251_m1), *Slco1b2* (Mm00451510_m1), *Tgfbr1* (Mm03024015_m1), *Timp1* (Mm00441818_m1), *Tnf* (Mm00443258_m1), *Wls* (Mm00509695_m1), *Wnt2* (Mm00470018_m1, Mm00437330_m1), *Wnt9b* (Mm00457102_m1) and *RSPO3* (Hs00262176_m1).

## RNA scope and spatial transcriptomics

Target RNA was detected using the RNAscope Multiplex Fluorescent Reagent Kit v2 (Advanced Cell Diagnostics (ACD), 323110) on 10 μm frozen mouse liver sections from *Lrat-cre*⁺LSL-TdTom⁺ mice using RNAscope Mm-*Rspo3* probe (ACD, 402011) and Opal 520 Reagent (Akoya Biosciences, FP1487001KT). Anti-RFP (Rockland, 600-401-379) was used to detect TdTomato as described previously[14]. Confocal microscopy was performed using an AXR confocal scanner mounted on a Ti2 microscope stand (Nikon Instruments) using a ×20/0.75 Plan-Apo VC objective lens or a ×60/1.49 NA Apo-TIRF oil-immersion objective lens in the DLDRC imaging core. 100-plex spatial transcriptomics (the 100 gene panel is shown in Supplementary Table 12) focusing on WNT pathway and zonation genes was done on the Resolve platform. Probe design as well as tissue sectioning, processing, probe design and hybridization, slide imaging, spot segmentation and data preprocessing were performed as previously described[29]. Single-cell spatial transcriptomic analysis was performed by quantifying gene counts per cell using cell segmentation using QuPath software[75]. The libraries from each condition (*iDTR^WT*, *iDTR^het*, *Rspo3^fl/fl*, *Rspo3^ΔHSC*) were integrated together using the R package Seurat[76]. Analysed cells included those filtered for greater than or equal to 10 gene counts per cell. After preprocessing, unbiased clustering on all 100 genes was performed using the dimensionality reduction method of principal component analysis (PCA) and uniform manifold approximation and projection (UMAP)[77]. Clusters corresponding to different zones were identified and annotated on the UMAP based on expression of different landmark genes as previously done[29]. Moreover, HSC- and EC-specific clusters were identified and annotated on the UMAP based on expression of *Lrat* and *Pecam1*, respectively. Furthermore, using the Seurat package, feature plots and violin plots were used to visualize gene expression at cluster and cell level, respectively. Lastly, once clusters were annotated, cells corresponding to specific clusters based on gene expression were mapped back onto the virtual slide to visualize spatial location of specific clusters. Expression of individual landmark genes for pericentral, midzonal and periportal zones were used to verify accurate spatial localization of clusters. For zone- and cell-specific analysis of *Rspo3* expression, the overall expression of established landmark genes was first used to define zones, followed by assignment of cells to one of three zones based on their localization within these regions. This was done by plotting cells positive for *Cyp2f2* on one image using ggplot2 in R, and cells positive for *Cyp2e1* on another ggplot2 image. These images were processed in MATLAB using the Image Processing Toolbox to smooth and fill in distinct regions as either zone 1 or zone 3, yielding two matrices delineating zone 1 and zone 3 regions. Cells were classified as zone 1 if they were located in the zone 1 region only, zone 3 if they were located in the zone 3 region only, and zone 2 if they were located in overlapping zone 1 and 3 regions. Gene expression levels, including *Rspo3*, were evaluated across both zone and cluster using the VlnPlot function in Seurat.

## Microarray and bulk RNA-seq, heat maps and pathway analysis

RNA-seq analyses were performed on high-quality total RNA samples, with RNA integrity numbers of >8 (determined using a Bioanalyzer 2100, Agilent Technologies). Bulk RNA-seq data were processed by the Columbia Genome Center. For each sample, a minimum of 20 million 100 bp single-end reads were sequenced on the Illumina NovaSeq 6000 system. RTA (Illumina) was used for base calling and bcl2fastq2 (v.2.19 and v.2.20) was used for converting BCL to fastq format, coupled with adaptor trimming. A pseudoalignment to a kallisto index was created from transcriptomes (human, GRCh38; mouse, GRCm38) using kallisto (v.0.44.0). To explore similarities and dissimilarities between samples, count data were normalized using the variance stabilizing transformation function from the DESeq2 package. Microarray analysis for comparison between *Ctnnb1^ΔHep* and *Ctnnb1^fl/fl* livers was conducted using a published dataset (GSE68779). Heat maps were generated using the Heatmapper tool[78]. KEGG pathway analysis in HSC-depleted liver was performed using enrichr using the 100 significant ($P < 0.05$) genes with the highest combined log-transformed fold change in HSC iDTR- and JEDI-depleted livers compared with their respective controls (GSE211370)[79].

## GSEA

GSEA was performed using GSEA v.4.3.2 software (https://www.gsea-msigdb.org/gsea/downloads.jsp). Analysis was performed from pre-ranked genes from DESeq2 (v.1.42.0) analysis of bulk RNA-seq data using the GSEAPreranked function with 1,000 permutations on the Hallmark collection from the Molecular Signature Database (MSigDB) or by curating the CTNNB1_Liver gene set from genes with a log-transformed fold change of >1 in the microarray analysis from *Ctnnb1*$^{\Delta Hep}$ and *Ctnnb1*$^{fl/fl}$ livers as reference for β-catenin-regulated liver genes.

## Nucleus isolation for snRNA-seq

For snRNA-seq analysis, human and mouse livers (Supplementary Tables 4 and 10) were processed as previously described[14]. In brief, frozen liver tissue was minced with scissors in 1 ml TST buffer in the well of a six-well plate for 10 min on ice. The homogenized solution was then passed through a 40-μm cell strainer. An additional 1 ml of TST buffer and 3 ml of 1× ST buffer were used to wash the well and passed through a 40-μm cell strainer. The resulting 5 ml of nuclei suspension was centrifuged for 5 min at 500g at 4 °C. The supernatant was discarded, and the pellet resuspended in 1 ml of 1× ST buffer. The nuclei suspension was then passed through a 35 μm filter. For single-cell multiome ATAC plus gene expression analysis, the Chromium Nuclei Isolation with RNase Inhibitor Kit (10x Genomics, PN-1000494) was used to isolate nuclei from mouse liver.

## scRNA-seq, snRNA-seq and scATAC and gene expression sequencing

All analysed scRNA-seq data, including HSCs isolated from 1× CCl$_4$-, 2× CCl$_4$-, 4× CCl$_4$- and 12× CCl$_4$-treated mouse livers (GSE172492) and 0, 15, 30 and 34 weeks of HF-HFD-treated mouse livers (GSE166504) as well as whole mouse liver (GSE158183) have been published and deposited previously. Samples for human snRNA-seq analysis (GSE256398) were prepared as previously described[14] using the 10x Chromium Single Cell Platform using a Chromium Single Cell 3′ Library and Gel Bead Kit v.3 and a Chromium Single Cell B Chip kit (10x Genomics, PN-1000074) according to the manufacturer's protocol. Data were aligned to a modified version of the GRCh38 reference genome (counting intronic reads as well as those aligned to exons), and estimated cell-containing partitions and associated unique molecular identifiers (UMIs) using Cell Ranger v.3.1.0 from 10x Genomics.

## snRNA-seq and scRNA-seq analysis

In total, 26 human snRNA-seq datasets and 4 mouse snRNA-seq datasets were analysed (Supplementary Table 10). Technical artefacts such as ambient background RNA and empty droplets in these datasets were removed using the remove-background function in CellBender v.0.2.0 as described (fpr = 0.01 for human datasets; fpr = 0.1 for mouse datasets)[14]. For human datasets, the output raw_feature_bc_matrix_filtered.h5 from CellBender for each sample was further subjected to doublet removal using Scrublet with the default parameters[80]. The resulting singlets from human datasets and the raw_feature_bc_matrix_filtered.h5 from mouse datasets were analysed in Seurat (v.5.0.1).

For each dataset, the samples were combined and low-quality cells or outlier cells were filtered using the same standard (nFeatureRNA < 200 or nFeatureRNA > 6500 or nCount_RNA > 40000 or percent.mt > 20 for human, nFeatureRNA < 200 or nFeatureRNA > 7500 or nCount_RNA > 60000 or percent.mt > 20 for mouse). Each sample was first analysed in parallel using the NormalizeData, FindVariableFeatures (nFeatures = 3000), ScaleData and RunPCA function. The samples were integrated using the SelectIntegrationFeatures, FindIntegrationAnchors (k.anchor = 10, reduction = "rpca") and IntegrateData functions. After integration, a single integrated analysis with a batch-corrected integrated count matrix layer was run on all cells using the ScaleData, RunPCA (npcs = 50), RunUMAP, FindNeighbors and FindClusters functions. All of the default parameters were used unless mentioned otherwise. The main cell types including T cell/natural killer, myeloid cells, HSCs, ECs, cholangiocytes and hepatocytes were identified manually by checking the expression of well-known marker genes as described previously[14]. For each main cell type, the clusters were subset and reclustered using the non-batch-corrected RNA count matrix layer using ScaleData, RunPCA (npcs = 50), RunUMAP, FindNeighbors and FindClusters function. Detailed cell types were identified manually using the expression of well-known marker genes as described previously[81] and markers for each population are provided in the Supplementary Information as dot plots for mouse snRNA-seq data (Supplementary Information 4) and human snRNA-seq data (Supplementary Information 5). Clusters that simultaneously express marker gene sets from two or more cell types were identified and further removed as doublets. For each dataset, the differentially expressed genes (DEGs) in every cell cluster were identified using the FindAllMarkers function in Seurat v.3.

Some published datasets, including human and mouse snRNA-seq from the Henderson lab[82] and from the Livercellatlas[81] were analysed using web-based tools described in the respective publications.

## CellPhoneDB analysis

After identifying cell types in each dataset as described above, we used CellPhoneDB[83] in the most recent v.5 version to identify ligand–receptor interactions in *n* = 6 healthy livers from our human snRNA-seq dataset (GSE256398). To determine mouse ligand–receptor interactions, mouse genes from *n* = 2 healthy control livers (GSE256398) were first converted to human gene symbols (HGNC) using biomaRt (v.2.60.1) package in R, followed by recommended procedures for preparation of input files. All CellPhoneDB statistical analysis was performed using the default parameters and the percentage cell expression threshold of 5%. After ranking cell–cell interactions by the interaction scores, all interactions with a positive interaction score were further ranked by the mean expression. Heat maps showing ligand–receptor interactions, log$_2$-transformed mean (molecule 1, molecule 2) and log$_{10}$[*P*] values were generated using ggplot2 (v.3.4.4) package.

## Gene expression and survival analysis in clinical cohorts of patients with CLDs

To determine survival and liver-related events in the SteatoSITE cohort of patients with MASLD[84], normalized counts per minute of RSPO3 from the biopsy subset of cases in the SteatoSITE data commons were used in survival analysis using R (v.4.3.0) in RStudio (v.2023.12.0 build 369) and the 'survminer' package (v.0.4.9). The optimal cutpoint for normalized RSPO3 counts was separately calculated for overall survival, and hepatic decompensation (when a first coding of any component of the composite outcome occurred after the biopsy date and with death as a competing risk) using surv_cutpoint, applying maximally selected rank statistics of the 'maxstat' package (v.0.7-25) with a minimum proportion of 0.25. Kaplan–Meier estimator curves of all-cause mortality were compared by regular log-rank testing with weights = 1. To determine survival in the dbGaP phs001807.v1.p1 cohort of patients with ALD[85], the median RSPO3 expression was established as a threshold for high and low expression cohorts, for analysis of survival using Kaplan–Meier estimator curves of all-cause mortality and log-rank testing. Moreover, genome-wide hepatic transcriptome datasets of clinical cohorts of MASLD without HCC (Gene Expression Omnibus (GEO): GSE49541 (ref. 86) and GSE193066 (ref. 87)) and with resected or ablated HCC (GSE192959 (ref. 87)), alcoholic-associated cirrhosis (GSE103580 (ref. 88)) and severe alcohol-associated hepatitis (GSE94397 (ref. 88)) were analysed for association with clinical disease phenotypes and outcome for RSPO1, RSPO2, RSPO3 and RSPO4 genes, including association with known WNT pathway target genes (*CYP2E1* and *CYP1A2*), and previously reported transcriptomic signatures of overall survival, decompensation and HCC risk in CLDs (prognostic liver signature (PLS))[89] and

overall survival in severe alcohol-associated hepatitis[88]. For analyses in these cohorts, high expression of the RSPO genes was defined based on a top-quartile cut-off in each cohort; the presence of high-risk pattern of the prognostic transcriptomic signatures was determined by the nearest template prediction algorithm[90]; associations of the high gene expression and presence of the high-risk signatures with clinical phenotypes and outcomes were evaluated using Wilcoxon rank-sum tests, log-rank tests and/or the Kaplan–Meier method depending on the availability of clinical annotations in each cohort.

## Genome-scale metabolic pathway analysis

We performed metabolic pathway analysis using liver transcriptomics data, comparing JEDI HSC-depleted livers to their controls (GSE211370), hepatocytes from $Rspo3^{fl/fl}$ and $Rspo3^{\Delta HSC}$ mouse livers (GSE256398) and livers from $Ctnnb1^{\Delta Hep}$ and $Ctnnb1^{fl/fl}$ mice (GSE68779), as described previously[91]. In brief, we translated the statistically significant DEGs (FDR ≤ 0.05) to enzymatic reaction rate changes according to gene–protein–reaction (GPR) rules. As a database for pathways, reactions and GPR rules, we used the Mouse1 (v.1.3.0) metabolic models, genome-scale metabolic reconstruction[92]. Metabolic pathways were then scored and ranked according to the amount of perturbed reactions that they encompass[91]. We also computed $P$ values for each pathway to evaluate their statistical significance. We used the hypergeometric test, which is based on the hypergeometric distribution[91]. The computed $P$ values were subjected to a FDR correction, using the Benjamini–Hochberg procedure.

## Bulk and spatial metabolomics

Snap-frozen mouse liver samples were analysed at the Roswell Park Comprehensive Cancer Center Bioanalytics, Metabolomics and Pharmacokinetics Shared Resource, using the MxP Quant 500 XL kit (Biocrates Life Sciences) according to the manufacturer's instructions. In brief, liver samples were homogenized at a ratio of 1 mg of tissue to 3 µl of solvent (25% ethanol and 75% 0.01 M phosphate buffer) using optimized settings on the Omni-Bead Ruptor 24 (Omni). After centrifugation, 10 µl of each supernatant, quality control samples, blank, zero sample or calibration standard were added on the filterspot (already containing internal standard) in the appropriate wells of two 96-well plates and dried under a gentle stream of nitrogen. On one plate, the samples were derivatised with phenyl isothiocyanate for the amino acids and biogenic amines, and dried again. Sample elution on both plates was performed with 5 mM ammonium acetate in methanol. Sample extracts were diluted with either water for the HPLC–MS/MS analysis (1:1) or kit running solvent (Biocrates Life Sciences) for flow injection analysis (FIA)–MS/MS (50:1), using the Shimadzu HPLC system interfaced with the Sciex 5500 mass spectrometer. Data were processed using WebIDQ software (Biocrates Life Sciences), and Limma for differential metabolite analysis.

For DESI MS imaging, fresh-frozen mouse liver tissue blocks were embedded in 5% gelatin over dry ice and stored at −80 °C. The tissue blocks were then cryosectioned at a thickness of 8 µm (Leica, CM3050S) and thaw-mounted onto a microscope slide and stored at −80 °C until analysis. The microscope slides were dried in a vacuum desiccator for 8 min. DESI MSI data acquisition was performed on the Synapt G2-XS QToF mass spectrometer coupled to a DESI ion source (Waters) in positive-ion sensitivity mode with a mass range of $m/z$ 100–1,000. The following DESI parameters were used: capillary voltage and sampling cone voltage of 0.5 kV and 50 V, respectively, DESI sprayer angle of 78 °C, source temperature of 150 °C, nebulizing gas ($N_2$) pressure of 0.9 bar and spatial resolution of 40 µm. The solvent used was methanol:water 95:5 (v/v) with 0.01% formic acid and 40 pg µl$^{-1}$ leucine enkephalin, at a flow rate of 1.5 µl min$^{-1}$. Data were processed and visualized in HDI imaging software (Waters, v.1.6) and SCiLS (Bruker, version 2024a). Peak picking and lockmass correction were performed using leucine enkephalin ([M+H]+, $m/z$ 556.2771). Lipid annotations were performed by accurate mass search against the Lipidmaps database[93] and also from annotations previously performed by lipidomics analysis using UPLC with ion mobility ToF MS$^E$ (HDMS$^E$) data-independent acquisition[94]. Co-localizing with glutamine synthetase by IHC was used to determine zonation in images. The distribution of the intensity of the ions and quantification using area under the curve of the corresponding lipid feature, denoting the average intensity of the $m/z$ interval within the tissue, normalized to TIC, were generated within the SCiLS software.

## Quantification and statistical analysis

No statistical methods were used to predetermine sample size. Investigators were blinded for in vivo treatments and post-mortem analyses such as (1) quantification by IHC and (2) determination of gene expression by qPCR. Investigators were not blinded for snRNA-seq analyses studies as there were not separate groups involved or the samples were annotated. For immunoblotting, the investigators were not blinded when loading the gel to display the results in a logical way. Statistical significance was determined using GraphPad Prism (v.9.0) or R (v.4.0.2). After assessing the normal distribution of the data using D'Agostino and Pearson omnibus normality tests and or Shapiro–Wilk's normality test, $P$ values were calculated, and all statistical tests used are described in the figure legends. Survival curves were represented using the Kaplan–Meier method and compared using log-rank statistics.

## Reporting summary

Further information on research design is available in the Nature Portfolio Reporting Summary linked to this article.

## Data availability

RNA-seq data have been deposited at the Gene Expression Omnibus database (GEO), including snRNA-seq data from livers of 26 human patients (GSE256398), snRNA-seq data from healthy mouse livers, $Rspo3^{\Delta HSC}$ and $Rspo3^{fl/fl}$ livers (GSE256398) as well as bulk RNA-seq data from HSC-depleted versus control mice, aged $Rspo3^{\Delta HSC}$ mice versus control mice and $Rspo3^{\Delta EC}$ versus control mice (GSE256377). The previously published datasets GSE68779, GSE211370, GSE172492, GSE158183, GSE49541, GSE193066, GSE192959, GSE103580 and GSE94397 were used for analysis. Source data are provided with this paper.

## Code availability

R markdown scripts enabling the main steps of the analysis have been deposited at GitHub (https://github.com/SchwabeLabcu/HSC_RSPO3).

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

**Acknowledgements** This work was supported by NIH grants 5R01DK128955 (to R.F.S.), 5R01CA262424-03 (to R.F.S.), by the Takeda Development Center Americas through a grant from the Takeda-Columbia-NYU alliance (to R.F.S.), a BIH Visiting Professorship grant by the Private Excellence Initiative Johanna Quandt (to R.F.S.), Deutsche Forschungsgemeinschaft grants Collaborative Research Center CRC1366 'Vascular Control of Organ Function' (project number 39404578; project C5 to H.G.A.), Collaborative Research Center CRC1324 'Wnt signaling' (project number 331351713; project A2 to H.G.A.), the European Research Council Advanced Grant 'AngioMature' (project 787181, to H.G.A.) and NIH 1RO1DK133512 and ACS Research Scholar Grant RSG-22-061-01-MM (both to Y.A.L.). Research was supported by the Columbia University Digestive and Liver Disease Research Center (5P30DK132710) through the use of the Bioimaging Core (subproject 8681) for confocal imaging, the Bioinformatic and Single Cell Core (subproject 8679) for single-cell analyses, the Organoid and Cell Culture Core (subproject 8680) for cell culture studies and assays and the Clinical Biospecimen and Research Core (subproject 8678) for histopathological evaluations, histology and single-nucleus isolation support; the Herbert Irving comprehensive Cancer Center (5P30CA013696) through its flow cytometry and molecular pathology shared resources as well as its pilot grant program; and the Pittsburgh Liver Research Center (5P30DK120531) through the use of its Genomics and Systems Biology Core (subproject 8833) and the 'Multidisciplinary approach to study of patients with Severe Alcoholic Hepatitis Undergoing Liver Transplantation', funded by 5P50AA027054 and the Vanderbilt Digestive Diseases Research Center, funded by NIH grant P30DK058404. The members of the H.G.A. laboratory thank the Genomics, Laboratory Animal and Light Microscopy Core Facilities of the DKFZ for technical support. A.S. was supported by a postdoctoral fellowship from the American Liver Foundation, the Uehara Memorial Foundation and the Cell Science Research Foundation. Construction of the SteatoSITE MASLD data commons by J.A.F. and T.J.K. was funded by Innovate UK (Precision medicine: impacting through innovative technology, TS/R017581/1). We thank T. Wang, K. Yan, J. Albrecht, A. Jain and Ö. Yilmaz for discussions; S. Rosario and the staff at the Bioanalytics, Metabolomics and Pharmacokinetics Shared Resource at the Roswell Park Comprehensive Cancer Center (supported by NCI P30CA16056) for technical support; and D. Brenner for being an outstanding mentor.

**Author contributions** A.S. designed experiments, generated, analysed and interpreted data in HSC-depleted mice and mice with constitutive *Rspo3* knockout and drafted the manuscript. Y.S. designed, conducted and interpreted experiments on liver regeneration in HSC-depleted mice and mice with constitutive *Wls* deletion and drafted the manuscript. G.W. designed experiments, generated, analysed and interpreted data in mice with inducible HSC and EC *Rspo3* knockout and drafted the manuscript. K.H.L. designed experiments, generated, analysed and interpreted data in mice with inducible HSC and EC *Rspo3* knockout. Q.S. designed and performed computational analyses of scRNA-seq and snRNA-seq data. J.Q. performed ligand–receptor analyses. C.Y. designed, performed and interpreted experiments in HSC-depleted mice, *Wls* mice and HSC–hepatocyte co-culture studies. Y.G. performed and evaluated co-culture studies and analysed RNA-seq data in patient cohorts. K.C.R. and Y.A.L. performed immunohistochemical analyses in JEDI-depleted mice. D.M.G. performed zone-specific analyses of expression, proliferation and cell death in stained liver sections. C.H. performed and interpreted PHx experiments in mice with deletion of *Col1a1*, *Pdgfrb* and *Tgfbr1*. A.F. performed and interpreted co-culture experiments and HSC isolations and characterized HSC-depleted mice. M.S. performed in vitro experiments in HSCs. D.Y. generated and maintained colonies of all mice for HSC depletion and conditional HSC knockouts. J.M. and B.I. isolated nuclei and supervised snRNA-seq from patient samples. B.M.L., S.L., T.M.Y. and M.O. designed, performed and analysed spatial transcriptomic studies. P.R. and T.V.N. designed, performed and analysed spatial DESI metabolite imaging, supervised by B.R.S. S.P.M. supervised and interpreted spatial transcriptomics analyses, and interpreted data on β-catenin functions in the liver including studies in *Wls*-deleted mice. T.L., A.G., M.P. and F.T. designed, performed and interpreted multiplex IHC analysis of liver zonation. T.S. and N.A. performed flow cytometry analyses in HSC-depleted and *Rspo3^{ΔHSC}* mice. J.B.S. and U.B.P. performed RNAscope analysis. S.K.B.-P. and R.A.H. performed metabolite analysis. H.K., N.F. and Y.H. performed transcriptomic analyses in human cohorts on *RSPO3* expression and association with outcomes. M.B., J.H., S.R., Z.S. and R.B. provided human samples for snRNA-seq. J.A., J.A.F. and T.J.K. provided and analysed clinical cohort data. H.R. performed histopathological analysis of human liver samples. M.M. analysed and performed metabolic analysis in transcriptomic data. H.G.A. conceived, designed interpreted and oversaw all experiments in mice with inducible HSC and EC deletion and drafted and edited the manuscript. R.F.S. conceived and oversaw the study, and designed and interpreted all experiments in mice with constitutive HSC deletion and conditional deletion of component of the WNT signalling pathways, drafted and edited the manuscript.

**Funding** Open access funding provided by Deutsches Krebsforschungszentrum (DKFZ).

**Competing interests** J.A.F. serves as a consultant or advisory board member for Resolution Therapeutics, Kynos Therapeutics, Ipsen, River 2 Renal, Stimuliver and Global Clinical Trial Partners, and has received research grant funding from Intercept Pharmaceuticals and Genentech. F.T. has received research support from AstraZeneca, MSD and Gilead Consulting, and honoraria for lectures from Novo Nordisk, AstraZeneca, Gilead, Abbvie, Alnylam, BMS, Intercept, Falk, Inventiva, MSD, Pfizer, Novartis, Merz, Sanofi and GSK. B.R.S. is listed as an inventor on patents and patent applications involving small-molecule drug discovery, ferroptosis and immunostaining; holds equity in Sonata Therapeutics; co-founded and serves as a consultant to Exarta Therapeutics and ProJenX; and serves as a consultant to Weatherwax Biotechnologies Corporation and Akin Gump Strauss Hauer & Feld. S.P.M. has received research funding from Alnylam, Fog Pharmaceuticals and serves as a consultant or on the scientific advisory board of Alnylam, Genentech, Surrozen, Vicero, Mermaid Bio, Antlera and UbiquiTx. R.F.S. has received research funding from Takeda via the Takeda-Columbia-NYU alliance. The other authors declare no competing interests.

**Additional information**

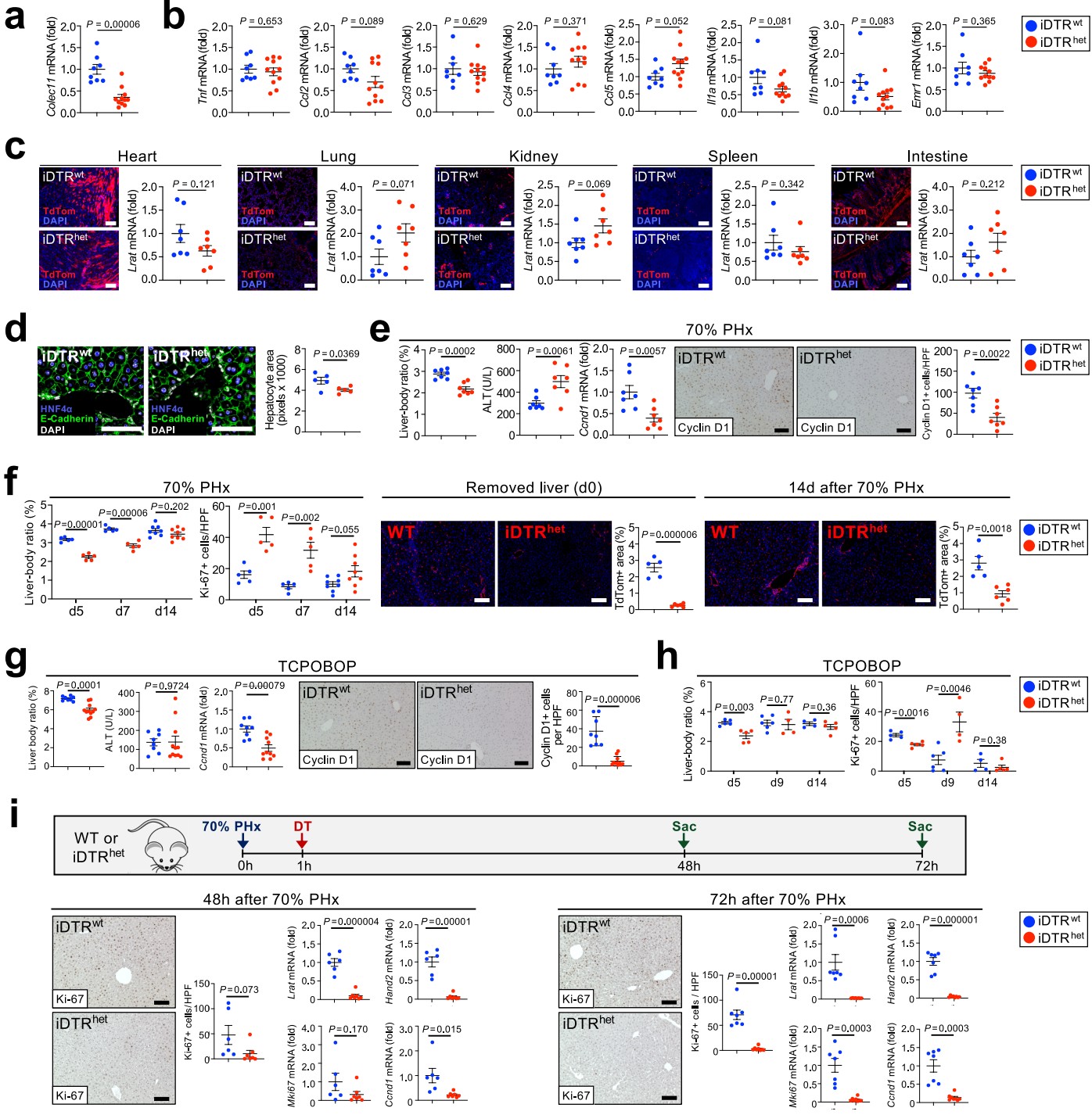

**Extended Data Fig. 1 | Characterization of iDTR x LratCre-mediated HSC depletion. a-b.** qPCR for HSC marker *Colec11* (**a**) or inflammatory genes (**b**) in HSC-depleted iDTR[het] (*n* = 11) and control iDTR[wt] (*n* = 8) mice. **c.** Representative TdTom images and qPCR for *Lrat* in various organs of iDTR[wt] and iDTR[het] mice (n = 7/group). **d.** Hepatocyte size, determined by co-staining for HNF4a and E-cadherin in iDTR[wt] and iDTR[het] mice (*n* = 5/group). **e.** Liver-body weight ratio, qPCR for hepatic *Ccnd1* mRNA, serum ALT and Cyclin D1 IHC in iDTR[het] and iDTR[wt] mice (*n* = 7/group) 48 h after 70% PHx (related to Fig. 1b). **f.** Liver-body weight ratio and quantification of Ki-67+ cells at day 5 (*n* = 5/group), day 7 (*n* = 5/group) and day 14 (iDTR[wt] *n* = 8, iDTR[het] *n* = 7) after 70% PHx as well as determination of HSC depletion efficiency by quantification of TdTom+ HSC at day 0 and day 14

after PHx in iDTR[wt] (*n* = 5) and iDTR[het] (*n* = 6) mice. **g.** Liver-body weight ratio, qPCR for hepatic *Ccnd1* mRNA, serum ALT, and Cyclin D1 IHC in iDTR[wt] (*n* = 8) and iDTR[het] mice (*n* = 11) 48 h after TCPOBOP (3 mg/kg) treatment (related to Fig. 1c). **h.** Liver-body weight ratio and Ki-67+ cells in iDTR[wt] and iDTR[het] mice at day 5 (*n* = 5/group), day 9 (iDTR[wt] *n* = 6, iDTR[het] *n* = 4) and day 14 (iDTR[wt] *n* = 4, iDTR[het] *n* = 5) after TCPOBOP (3 mg/kg) treatment. **i.** iDTR[wt] and iDTR[het] mice were subjected to 70% PHx, injected with DT 1 h later and euthanized 48 h (iDTR[wt] *n* = 6, iDTR[het] *n* = 7) or 72 h (iDTR[wt] *n* = 7, iDTR[het] *n* = 8) after 70% PHx to determine HSC depletion and proliferation by qPCR and Ki-67 IHC. Data are shown as mean ± s.e.m. Each dot represents one biological replicate (**a-i**). Scale bars 100 μm (**c-g,i**). P-values were calculated using unpaired two-tailed t-tests (**a-i**).

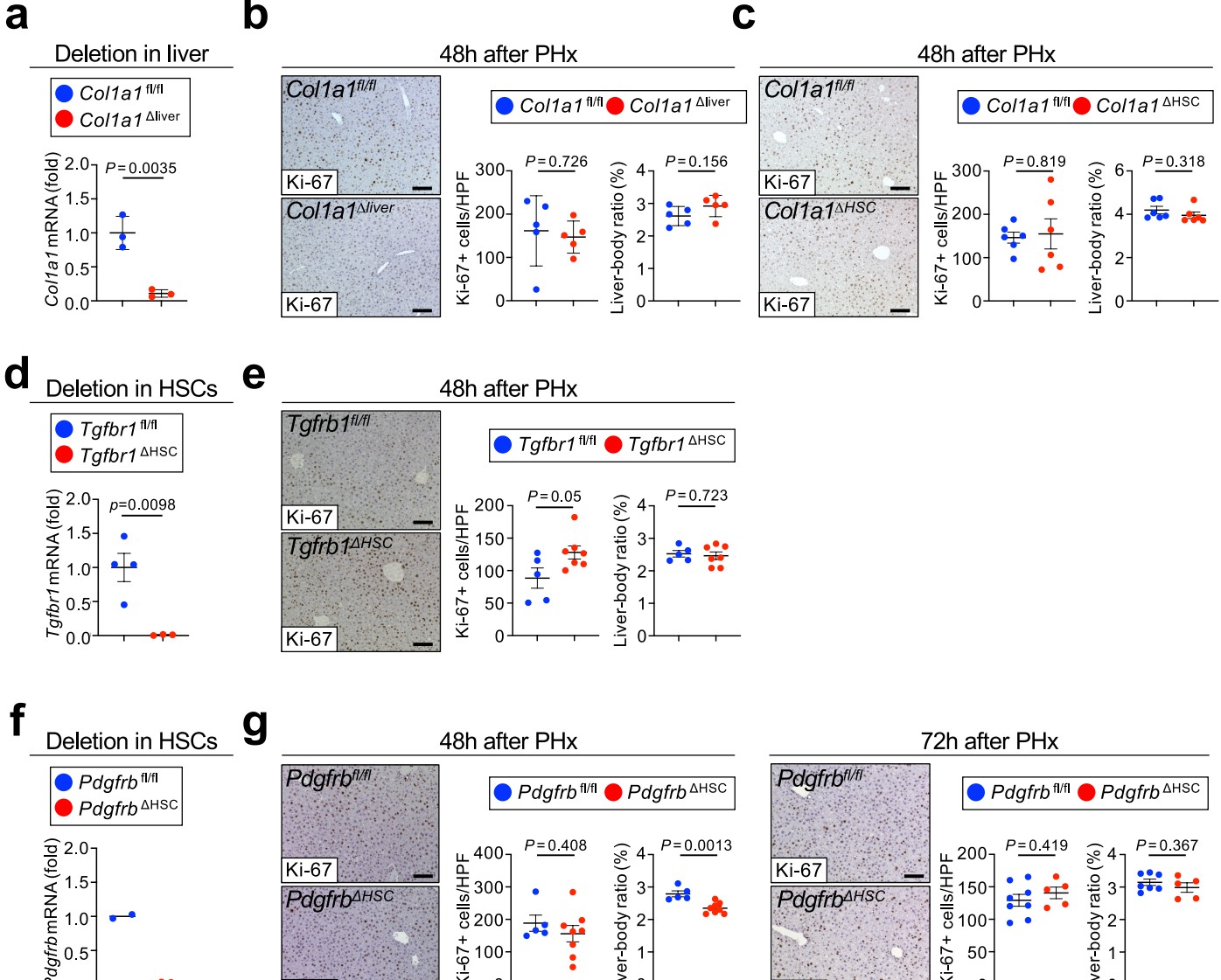

**Extended Data Fig. 2 | Effects of *Col1a1*, *Tgfbr1* and *Pdgfrb* deletion on liver regeneration.** a. qPCR for *Col1a1* livers from *Col1a1*<sup>fl/fl</sup> (*n* = 3) and *Col1a1*<sup>Δliver</sup> mice (*n* = 3). **b-c**. Representative images Ki-67 IHC as well as quantification and determination of the liver-body weight ratio in *Col1a1*<sup>fl/fl</sup> and *Col1a1*<sup>Δliver</sup> mice (**b**, *n* = 5/group) or in *Col1a1*<sup>fl/fl</sup> and *Col1a1*<sup>ΔHSC</sup> mice (**c**, *n* = 6/group) 48 h after 70% PHx. **d**. qPCR for *Tgfrb1* in HSCs isolated from *Tgfrb1*<sup>fl/fl</sup> (*n* = 4) and *Tgfrb1*<sup>ΔHSC</sup> mice (*n* = 3). **e**. Representative images Ki-67 IHC as well as quantification and determination of the liver-body weight ratio in *Tgfrb1*<sup>fl/fl</sup> (*n* = 5) and *Tgfrb1*<sup>ΔHSC</sup>

mice (*n* = 7) 48 h after 70% PHx. **f**. qPCR for *Pdgfrb* in HSCs isolated from *Pdgfrb*<sup>fl/fl</sup> (*n* = 2) and *Pdgfrb*<sup>ΔHSC</sup> mice (*n* = 3). **g**. Representative images Ki-67 IHC as well as quantification and determination of the liver-body weight ratio in *Pdgfrb*<sup>fl/fl</sup> (*n* = 5) and *Pdgfrb*<sup>ΔHSC</sup> mice (*n* = 8) 48 h after 70% PHx or in *Pdgfrb*<sup>fl/fl</sup> (*n* = 8) and *Pdgfrb*<sup>ΔHSC</sup> mice (*n* = 5) 72 h after PHx. Data are shown as mean ± s.e.m. Each dot represents one biological replicate (**a-g**). Scale bars, 100 μm (**b,c,e,g**). P-values were calculated using unpaired two-tailed t-tests (**a-e,g**).

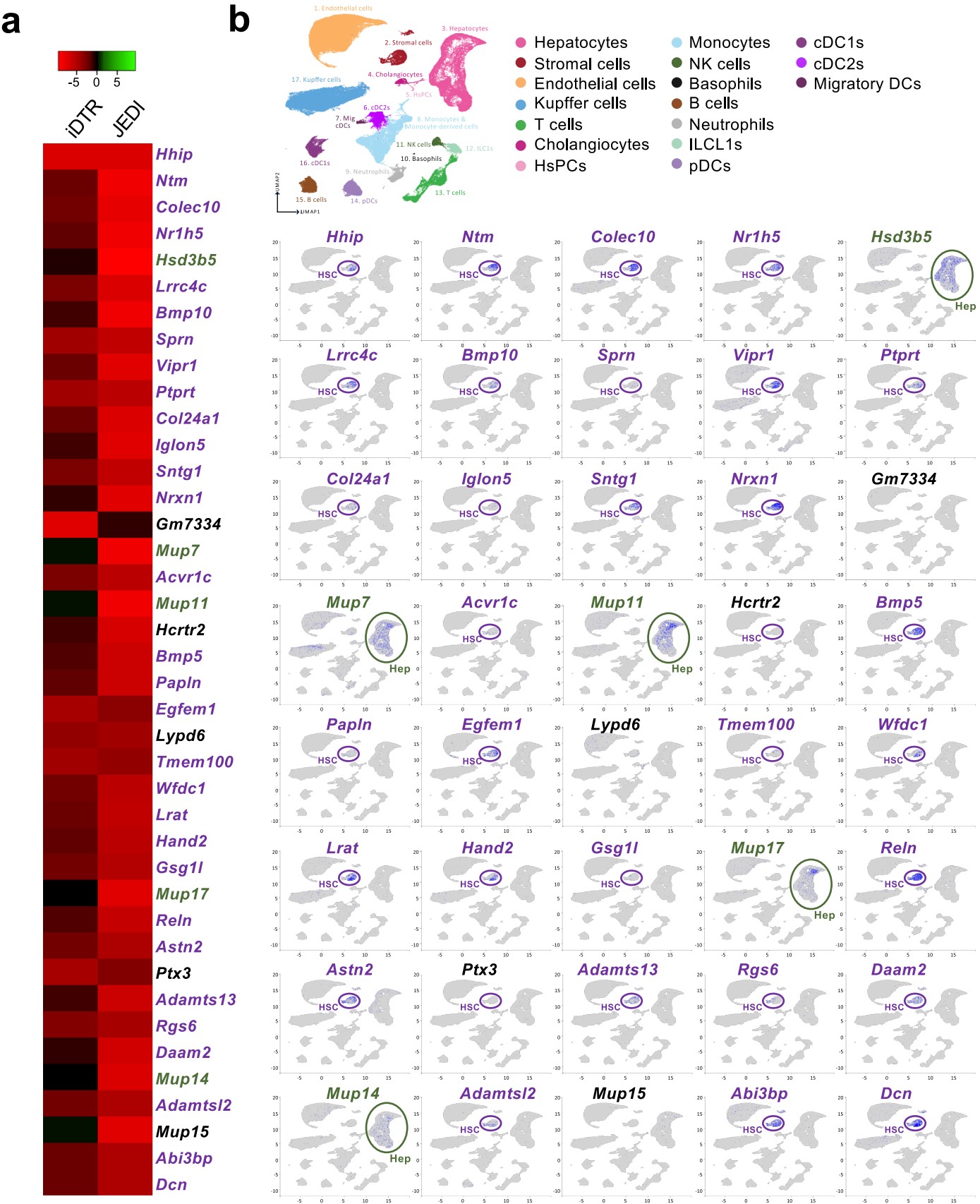

**Extended Data Fig. 3 | Differentially expressed genes and their enrichment in HSC-depleted mice. a-b.** Heatmap showing the top 40 genes with the strongest combined downregulated in RNA-seq from livers of iDTR- and JEDI HSC-depleted mice vs their respective non-depleted controls (**a**) and UMAPs of genes from panel A showing expression in different cell populations in scRNA-seq from mouse liver (**b**). HSC-enriched genes in purple (30 out of 40 genes), hepatocyte-enriched genes in green (5 out 40 genes), non-enriched genes in black (5 out of 40 genes).

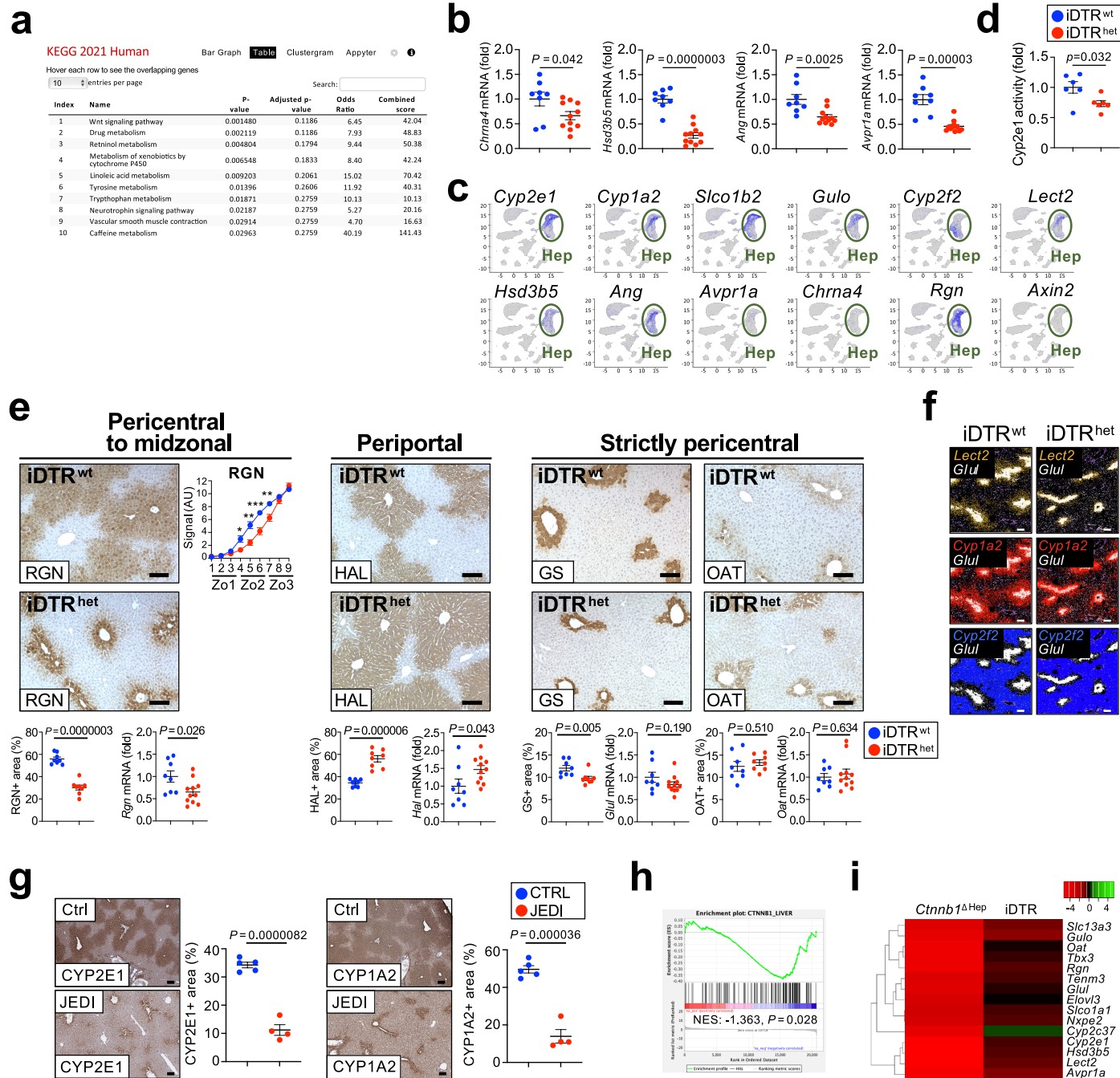

**Extended Data Fig. 4 | Pathway activation and zonation in HSC-depleted mice. a**. KEGG pathway analysis using the top 100 non-HSC genes (<3 logFC HSC enriched) with strongest combined downregulation in RNA-seq from JEDI and iDTR HSC-depleted livers. **b**. qPCR of indicated genes in iDTR HSC-depleted livers ($n$ = 11) vs controls ($n$ = 8). **c**. UMAPs for genes with important hepatocyte functions (related to Fig. 2b) in scRNA-seq from mouse liver. **d**. Cyp2e1 activity in livers from HSC-depleted (iDTR) and control mice ($n$ = 6/group). **e**. IHC, morphometric quantification and qPCR as well as zonal quantification (for RGN only – for zonal quantification of HAL, GS and OAT, see Fig. 2e) for pericentral to midzonal marker RGN, periportal markers HAL, and strictly pericentral markers

GS and OAT in livers from iDTR$^{wt}$ ($n$ = 8) and iDTR$^{het}$ mice ($n$ = 11) **f**. 100 plex spatial transcriptomics showing the indicated zonal genes in iDTR$^{wt}$ and iDTR$^{het}$ mice ($n$ = 1/group) **g**. Representative images and quantification of CYP2E1 and CYP1A2 IHC in HSC JEDI-depleted livers ($n$ = 4) vs controls ($n$ = 5). **h-i**. GSEA (**h**) of liver CTNNB1-regulated genes from RNA-seq of HSC-depleted (iDTR) vs control livers and heatmap (**i**) of the top 15 genes downregulated in *Ctnnb1*$^{ΔHep}$ mice vs *Ctnnb1*$^{fl/fl}$ controls alongside expression of the same genes in HSC-depleted (iDTR) vs control livers. Data are shown as mean ± s.e.m. Each dot represents one biological replicate (**b,d,e,g**). Scale bars, 100 μm (**e,g**). P-values were calculated using unpaired two-tailed t-tests (**b,d,e,g**).

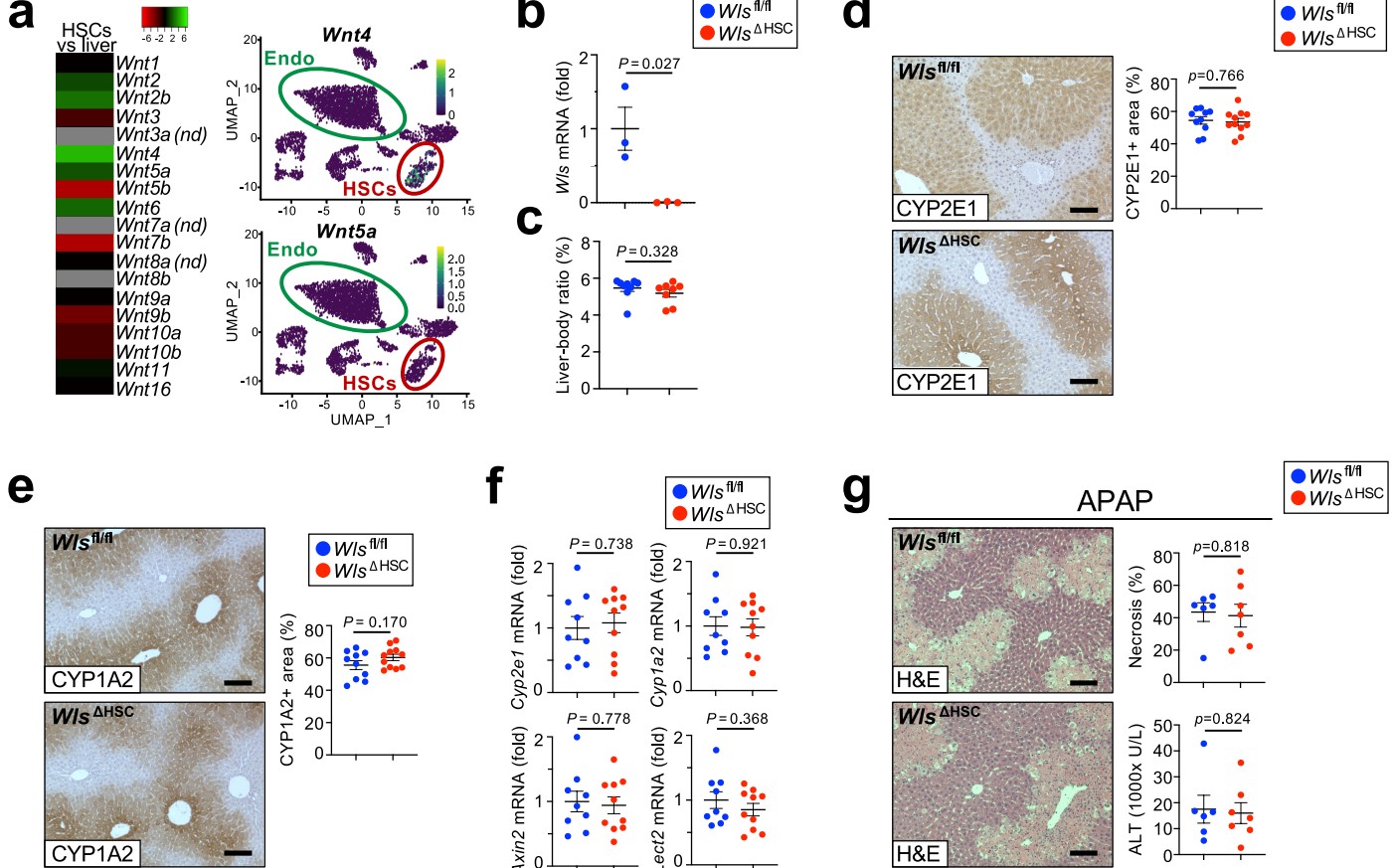

**Extended Data Fig. 5 | HSC-selective Wntless deletion does not alter liver zonation, injury and regeneration. a.** Heatmap of RNA-seq data comparing Wnt gene expression between isolated mouse HSCs and whole liver (left) and *Wnt4* and *Wnt5a* expression in scRNA-seq from whole mouse liver. **b.** *Wls* deletion in HSCs isolated from LratCre+ *Wls*$^{\Delta HSC}$ ($n = 3$) and LratCre- *Wls*$^{fl/fl}$ ($n = 3$) mice, as determined by qPCR. **c.** Liver-body weight ratio in LratCre+ *Wls*$^{\Delta HSC}$ ($n = 4$) and LratCre- *Wls*$^{fl/fl}$ ($n = 4$) mice. **d-e.** CYP2E1 IHC (**d**) and CYP1A2 IHC (**e**) and quantification for *Wls*$^{fl/fl}$ ($n = 10$) and *Wls*$^{\Delta HSC}$ ($n = 11$) mice **f.** qPCRs for Wnt target genes *Cyp2e1*, *Cyp1a2*, *Lect2* and *Axin2* in livers from *Wls*$^{fl/fl}$ ($n = 9$) and *Wls*$^{\Delta HSC}$ ($n = 11$) mice. **g.** Liver injury induced by APAP (300 mg/kg) was determined by ALT measurement and quantification of the necrosis area 24 h after APAP injection in LratCre+ *Wls*$^{\Delta HSC}$ ($n = 7$) and LratCre- *Wls*$^{fl/fl}$ ($n = 6$) mice. Data are shown as mean ± s.e.m. Each dot represents one biological replicate (**b-g**). Scale bars, 100 μm (**d,e,g**). P-values were calculated using unpaired two-tailed t-tests (**b-g**).

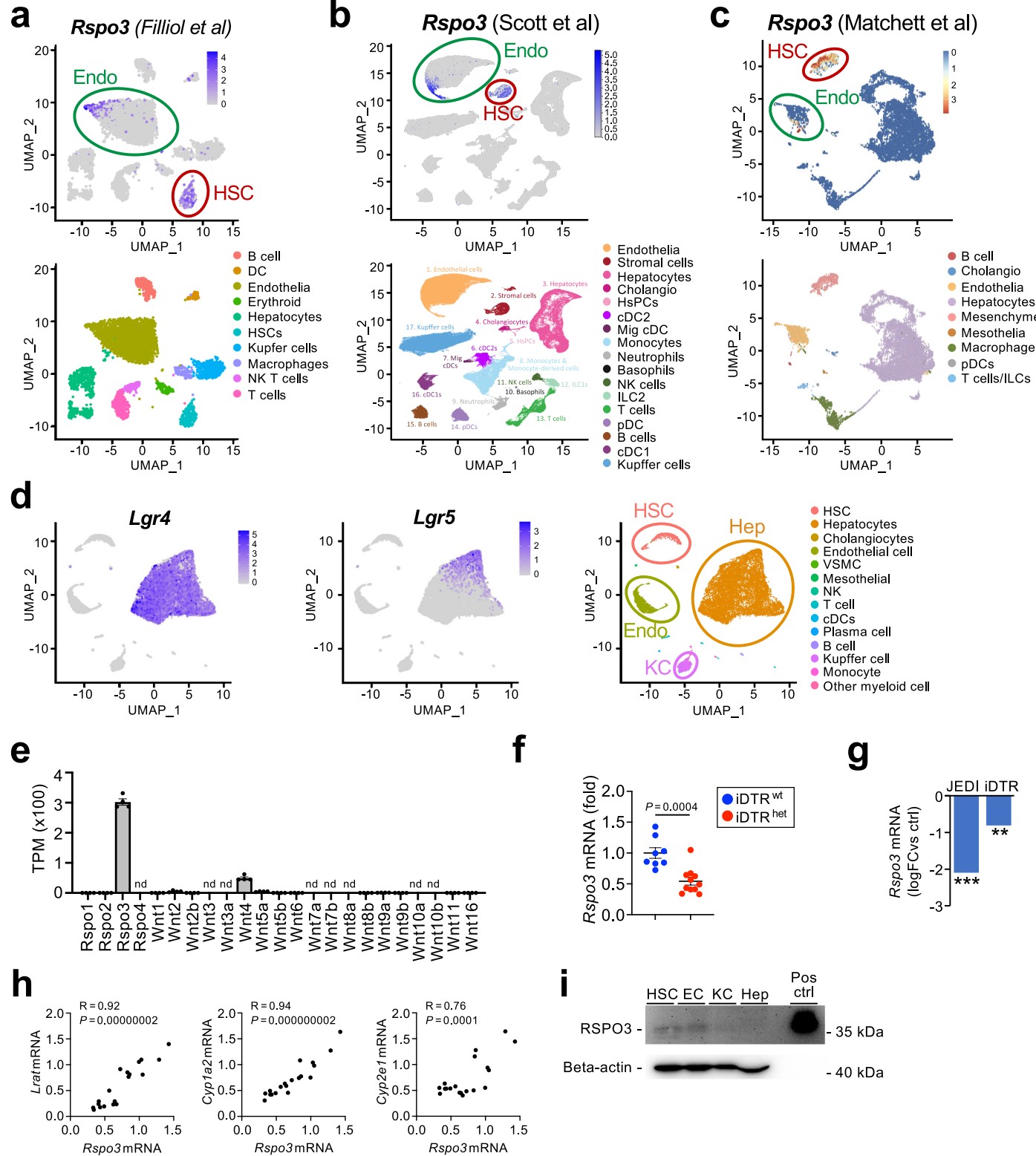

**Extended Data Fig. 6 | R-spondin3 expression in the liver. a-c.** UMAPs of Rspo3 in healthy mouse livers from the indicated sc/snRNA-seq datasets. **d.** UMAPs of Lgr4 and Lgr5 in snRNA-seq from healthy mouse livers (n = 2). **e.** RNA-seq of isolated mouse HSCs showing TPM values for secreted Wnt pathway regulators (n = 4/group). **f.** PCR of *Rspo3* mRNA in livers from HSC-depleted iDTR[het] (n = 11) and control iDTR[wt] (n = 8) mice. **g.** *Rspo3* mRNA expression from bulk RNA-seq in livers with HSCs depleted by the iDTR or JEDI method, compared to control mice. **h.** Correlation of *Rspo3* mRNA with *Lrat*, *Cyp1a2* and *Cyp2e1* mRNA in HSC-depleted mice. **i.** Western blot for RSPO3 using lysates from freshly isolated mouse HSCs, ECs, Kupffer cells (KCs) and hepatocytes (Hep) and rmRSPO3 as positive control (representative image of 2 technical replicates). nd, not detected **P < 0.01 vs ctrl, ***P < 0.001 vs ctrl. Data are shown as mean ± s.e.m. Each dot represents one biological replicate (**f,h**). P-values were calculated using unpaired two-tailed t-tests (**f**) and Wilcoxon rank-sum test (**g**). Correlations were evaluated by pearson correlation coefficient (**h**).

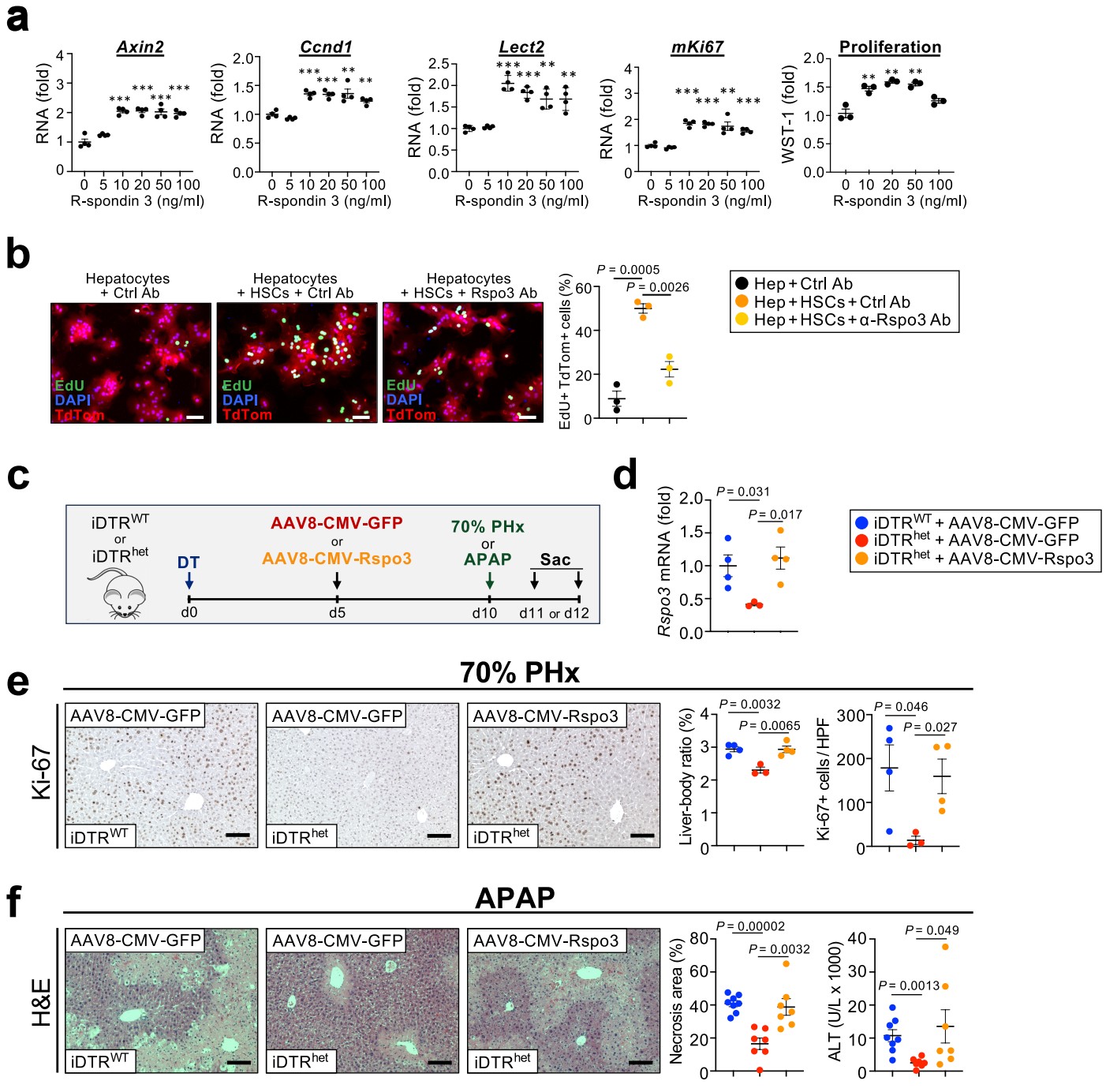

**Extended Data Fig. 7 | R-spondin3 promotes hepatocyte proliferation and rescues HSC-depleted livers. a**. qPCR (*n* = 4/group) for indicated genes and WST-1 assay (*n* = 3/group) in AML12 hepatocytes treated with the indicated concentrations of Rspo3 for 24 h (qPCR) or 48 h (WST-1 assay). **b**. Effect of HSC co-culture on hepatocyte proliferation in the absence or presence of Rspo3-blocking antibody rosmantuzumab or control antibody. **c**. Schematic diagram showing AAV8-CMV-GFP or AAV8-CMV-Rspo3 rescue experiments in HSC-depleted mice followed by 70% PHx or treatment with APAP (300 mg/kg). **d**. *Rspo3* mRNA was determined by qPCR in non-depleted control livers and liver from HSC-depleted iDTR mice after treatment with AAV8-CMV-GFP (iDTR^wt *n* = 4, iDTR^het *n* = 3) or AAV8-CMV-Rspo3 (*n* = 4). **e**. Proliferation was determined 48 h after 70% PHx by Ki-67 IHC in livers from AAV8-CMV-GFP-treated WT (*n* = 4) and iDTR^het (*n* = 3) and in AAV8-CMV-Rspo3 treated iDTR^het mice (*n* = 4). **f**. Liver injury was determined by serum ALT and necrosis quantification in H&E liver sections 24 h after APAP treatment in AAV8-CMV-GFP-treated WT (*n* = 8) and iDTR^het (*n* = 7) and in AAV8-CMV-Rspo3 treated iDTR^het mice (*n* = 7). nd ** P < 0.01 vs ctrl, *** P < 0.001 vs ctrl. Data are shown as mean ± s.e.m. Each dot represents one biological replicate (**a**,**b**,**d-f**). Scale bars 100 μm (**b**,**e**,**f**). P-values were calculated using unpaired two-tailed t-tests (**a-d**).

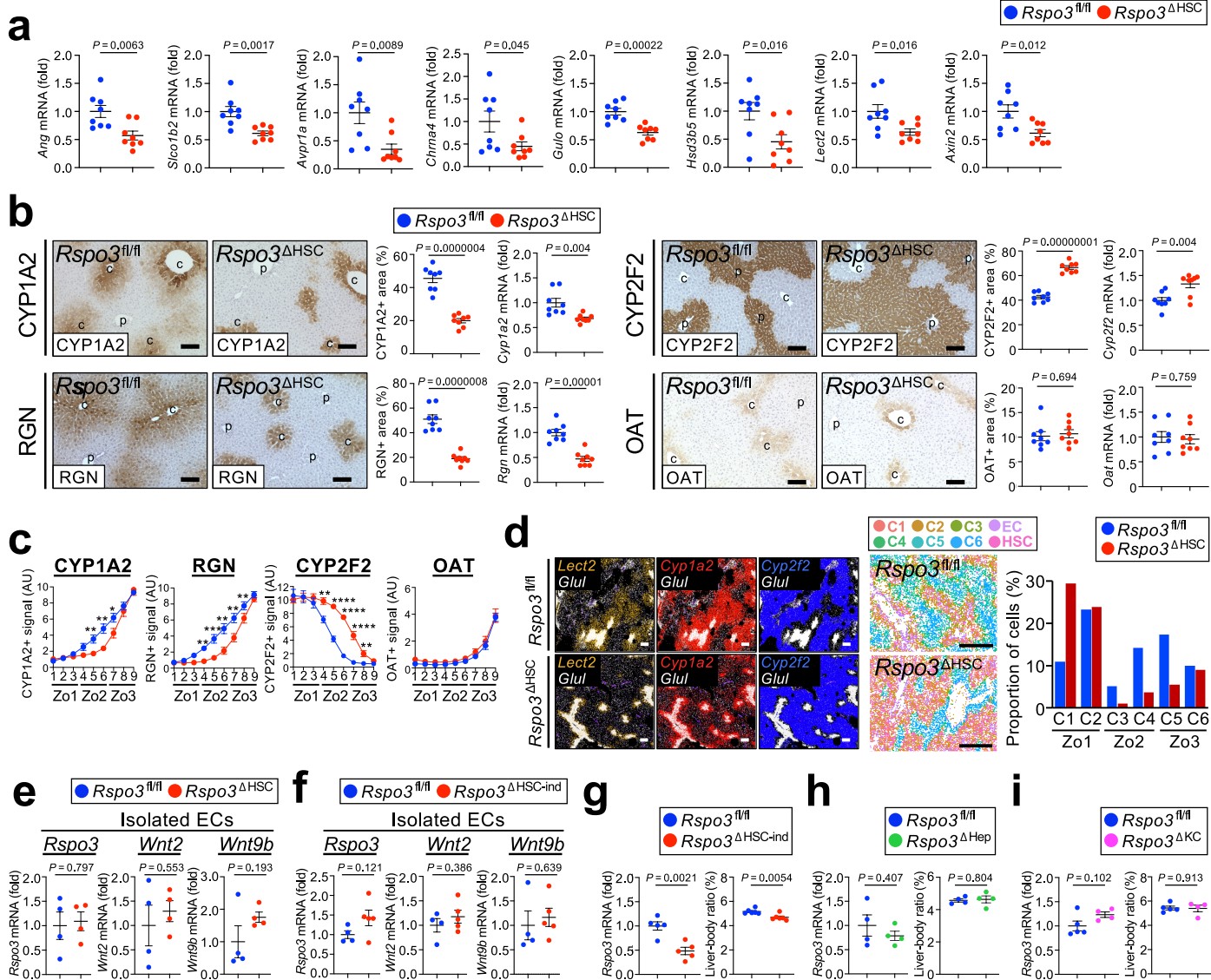

**Extended Data Fig. 8 | Characterization of mice with HSC-, EC-, hepatocyte and KC-specific Rspo3 deletion. a**. qPCR for the indicated genes in livers from $Rspo3^{fl/fl}$ ($n = 8$) and $Rspo3^{\Delta HSC}$ ($n = 8$, 7 male, 1 female) mice. **b**. IHC, morphometric quantification and qPCR for the indicated zonation markers in $Rspo3^{fl/fl}$ and $Rspo3^{\Delta HSC}$ mice ($n = 8$/group). **c**. Zonal quantification based on IHC from b ($n = 8$). **d**. Wnt-focused 100plex spatial transcriptomics and cluster-based zonal quantification in $Rspo3^{fl/fl}$ ($n = 1$) and $Rspo3^{\Delta HSC}$ ($n = 1$) mice. **e-f**. qPCR for $Rspo3$,

$Wnt2$ and $Wnt9b$ in ECs from $Rspo3^{\Delta HSC}$ ($n = 4$) (**e**) and $Rspo3^{\Delta HSC-ind}$ ($n = 5$) (**f**) mice or $Rspo3^{fl/fl}$ littermates ($n = 4$). **g-i**. Hepatic $Rspo3$ mRNA and liver-body weight ratio in $Rspo3^{\Delta HSC-ind}$ ($n = 5$) (**g**), $Rspo3^{\Delta Hep}$ ($n = 4$) (**h**) and $Rspo3^{\Delta KC}$ ($Rspo3^{fl/fl}$ $n = 5$, $Rspo3^{\Delta KC}$ $n = 4$) (**i**) mice or floxed controls. Data are shown as mean ± s.e.m. Each dot represents one biological replicate (**a-b,e-i**). Scale bars 100 μm (**c,k,l,m,n,o**), 100 μm (**d**, white), 1 mm (**d**, black). P-values were calculated using unpaired two-tailed t-tests (**a-c,e-i**).

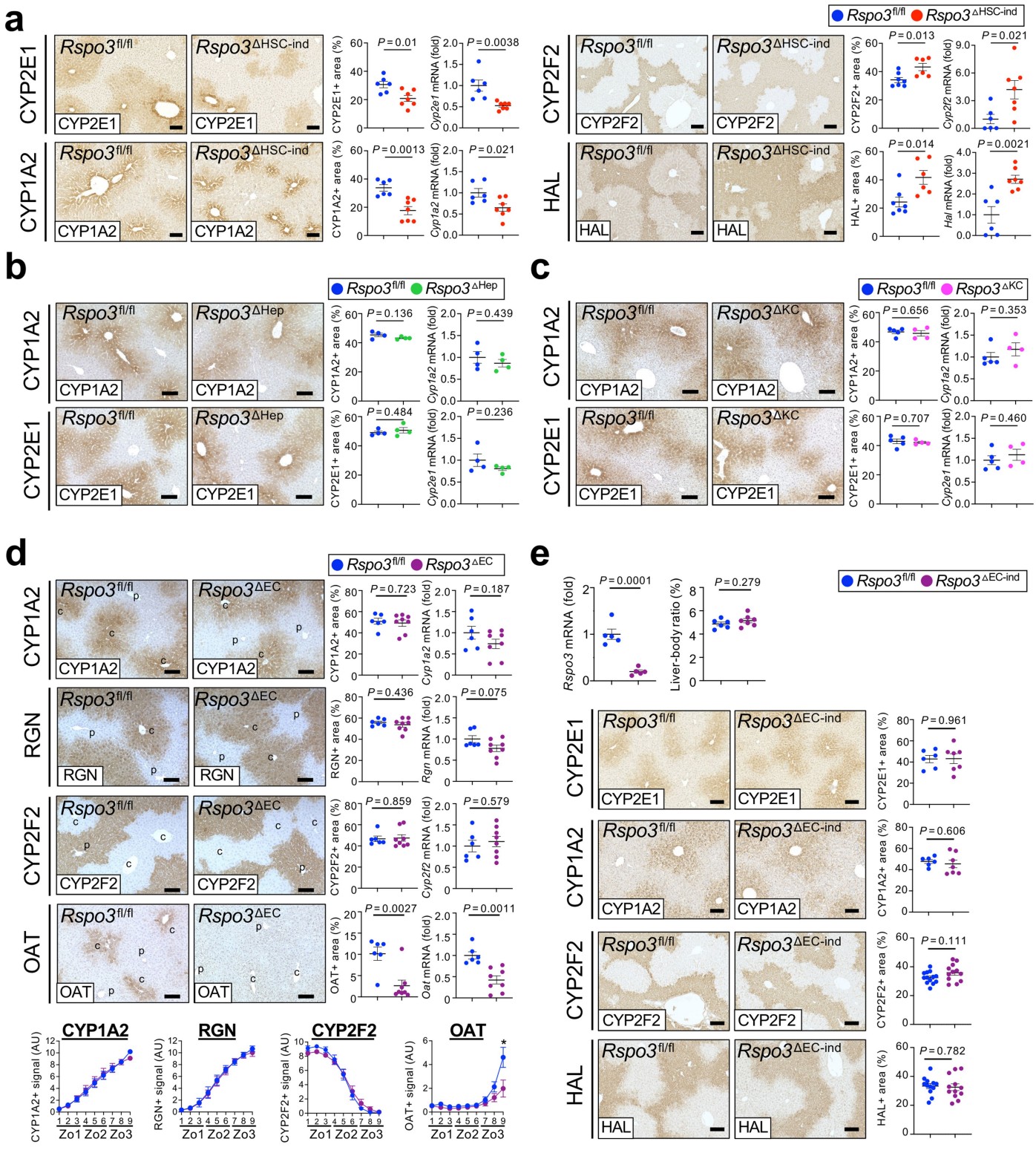

**Extended Data Fig. 9 | Liver zonation in mice with HSC-, EC-, hepatocyte and KC-specific Rspo3 deletion. a-d**. IHC, morphometric quantification and qPCR of zonation markers and, in some cases, zonal quantification of IHC in livers from *Rspo3*[ΔHSC-ind] (*n* = 7) or *Rspo3*[fl/fl] (*n* = 6) (**a**), *Rspo3*[ΔHep] (*n* = 4) or *Rspo3*[fl/fl] (*n* = 4) (**b**), *Rspo3*[ΔKC] (*n* = 4) or *Rspo3*[fl/fl] (*n* = 5) (**c**), and *Rspo3*[ΔEC] (*n* = 8) or *Rspo3*[fl/fl] (*n* = 6) (**d**) littermates. **e**. *Rspo3* qPCR in ECs (*n* = 5), the liver-body weight ratio

(*Rspo3*[fl/fl] *n* = 6, *Rspo3*[ΔEC-ind] *n* = 7), IHC, morphometric quantification of zonation markers CYP2E1 and CYP1A2 (*Rspo3*[fl/fl] *n* = 6, *Rspo3*[ΔEC-ind] *n* = 7), CYP2F2 (*Rspo3*[fl/fl] *n* = 13, *Rspo3*[ΔEC-ind] *n* = 12) and HAL (*Rspo3*[fl/fl] *n* = 13, *Rspo3*[ΔEC-ind] *n* = 12) in livers of *Rspo3*[ΔEC-ind] mice. Data are shown as mean ± s.e.m. Each dot represents one biological replicate (**a-e**). Scale bars 100 μm (**a-e**), P-values were calculated using unpaired two-tailed t-tests (**a-e**).

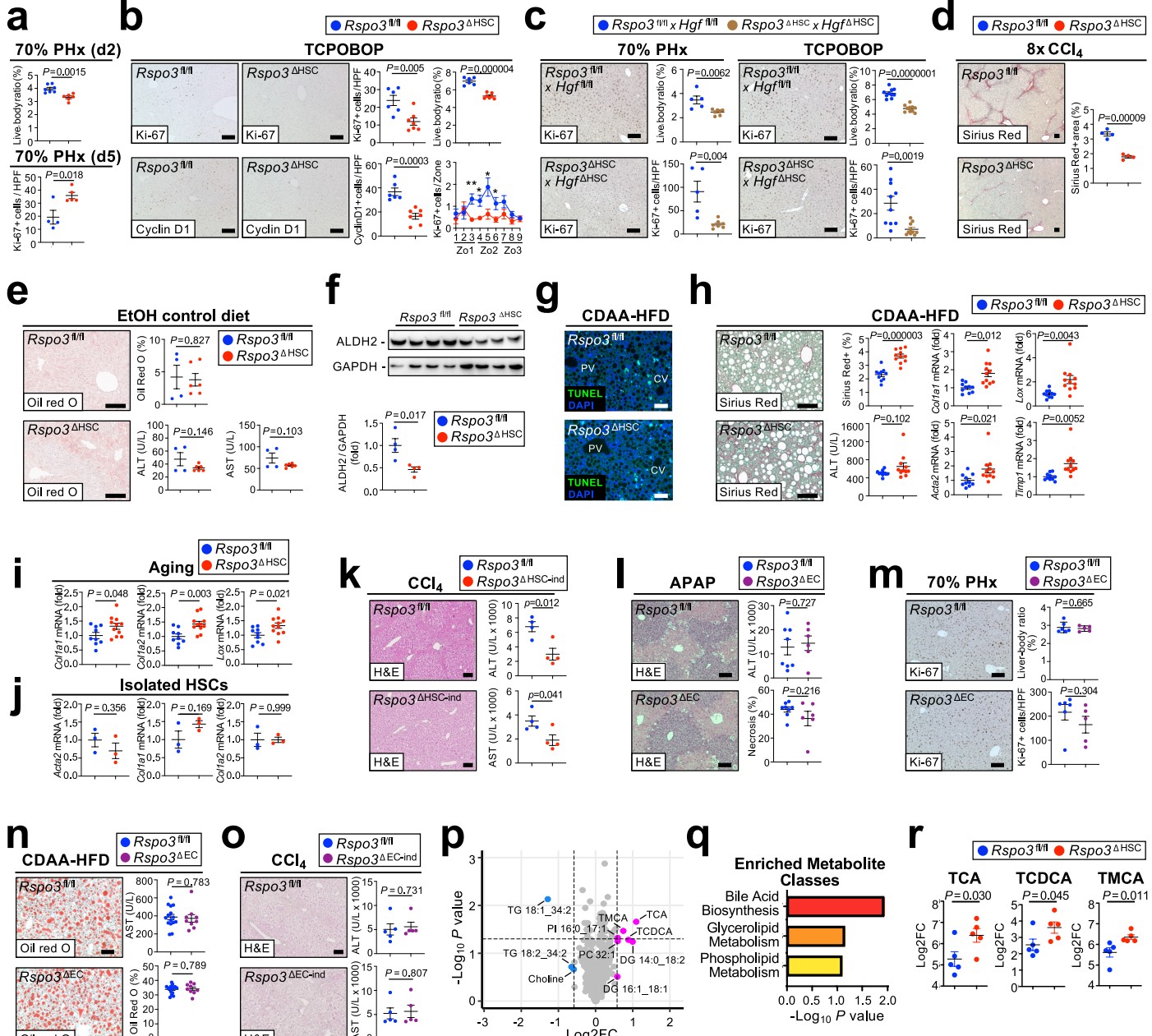

**Extended Data Fig. 10 | Rspo3 functions in the liver. a**. Liver-body weight ratio 48 h after 70% PHx (*Rspo3*^fl/fl^ *n* = 7, *Rspo3*^ΔHSC^ *n* = 6) and Ki-67 IHC 5 days after 70% PHx (*Rspo3*^fl/fl^ *n* = 4, *Rspo3*^ΔHSC^ *n* = 5) in *Rspo3*^fl/fl^ and *Rspo3*^ΔHSC^ mice. **b**. *Rspo3*^fl/fl^ (*n* = 6) and *Rspo3*^ΔHSC^ (*n* = 7) mice were treated with TCPOBOP. The liver-body weight ratio, total Ki-67 and zonal Ki-67 were quantified 48 h later. **c**. *Rspo3*^fl/fl^ *x Hgf*^fl/fl^ (*n* = 5 for 70% PHx, n = 10 for TCPOBOP), and *Rspo3*^ΔHSC^ *x Hgf*^ΔHSC^ dko mice (*n* = 7 for 70% PHx, n = 10 for TCPOBOP) were subjected to 70% PHx or TCPOBOP treatment, followed by determination of the liver-body weight ratio, Ki-67 IHC and morphometric quantification 48 h later. **d**. *Rspo3*^fl/fl^ and *Rspo3*^ΔHSC^ mice (*n* = 4/group) were treated with 8xCCl₄, followed by Sirus red staining and morphometric quantification. **e**. *Rspo3*^fl/fl^ (*n* = 4) and *Rspo3*^ΔHSC^ (*n* = 6) mice were treated with alcohol-free control liquid diet, followed by Oil red O staining, ALT and AST determinations. **f**. Alcohol dehydrogenase 2 (ALDH2) western blot in *Rspo3*^fl/fl^ (*n* = 4) and *Rspo3*^ΔHSC^ (*n* = 4) livers and quantification. **g**. Representative TUNEL images in *Rspo3*^fl/fl^ and *Rspo3*^ΔHSC^ mice treated with CDAA-HFD for 6 weeks. **h**. qPCR for fibrogenic genes, Sirius red staining and quantification and serum ALT in *Rspo3*^fl/fl^ (*n* = 10) and *Rspo3*^ΔHSC^ (*n* = 11) mice treated with CDAA-HFD for 6 weeks. **i**. Determination of fibrogenic genes by qPCR in 32-42 weeks old aged *Rspo3*^fl/fl^ (*n* = 9) and *Rspo3*^ΔHSC^ mice (*n* = 11). **j**. qPCR for fibrogenic genes in HSCs from *Rspo3*^fl/fl^ and *Rspo3*^ΔHSC^ mice (*n* = 3 each). **k**. Determination of CCl₄-induced liver injury in *Rspo3*^fl/fl^ mice (*n* = 4) and mice with inducible HSC-specific Rspo3 deletion (*Rspo3*^ΔHSC-ind^, *n* = 4). **l-n**. Liver injury, Ki-67 IHC and Oil Red O staining in *Rspo3*^fl/fl^ (*n* = 8) and *Rspo3*^ΔEC^ mice (*n* = 6) subjected to APAP treatment (**l**), *Rspo3*^fl/fl^ (*n* = 6) and *Rspo3*^ΔEC^ *n* = 5) mice subjected to 70% PHx (**m**), or *Rspo3*^fl/fl^ (*n* = 14) and *Rspo3*^ΔEC^ (*n* = 9) mice subjected to 6 weeks of CDAA-HFD (**n**). **o**. CCl₄-induced liver injury in *Rspo3*^fl/fl^ (*n* = 5) and *Rspo3*^ΔEC-ind^ mice (*n* = 5). **p-r**. Biocrates 500 XL metabolomic analysis of livers from *Rspo3*^ΔHSC^ (*n* = 5) and *Rspo3*^fl/fl^ (*n* = 5) mice showing a volcano plot of metabolites (**p**), enriched metabolite classes (**q**) and bile acids taurocholic acid (TCA), taurochenodeoxycholic acid (TCDCA) and tauromuricholic acid (TMCA) (**r**). Data are shown as mean ± s.e.m. Each dot represents one biological replicate (**a-f,h-o,r**). Scale bars 100 μm (**b-e,g-h,k-o**). P-values were calculated using unpaired two-tailed t-tests (**a-f,h-o,r**).

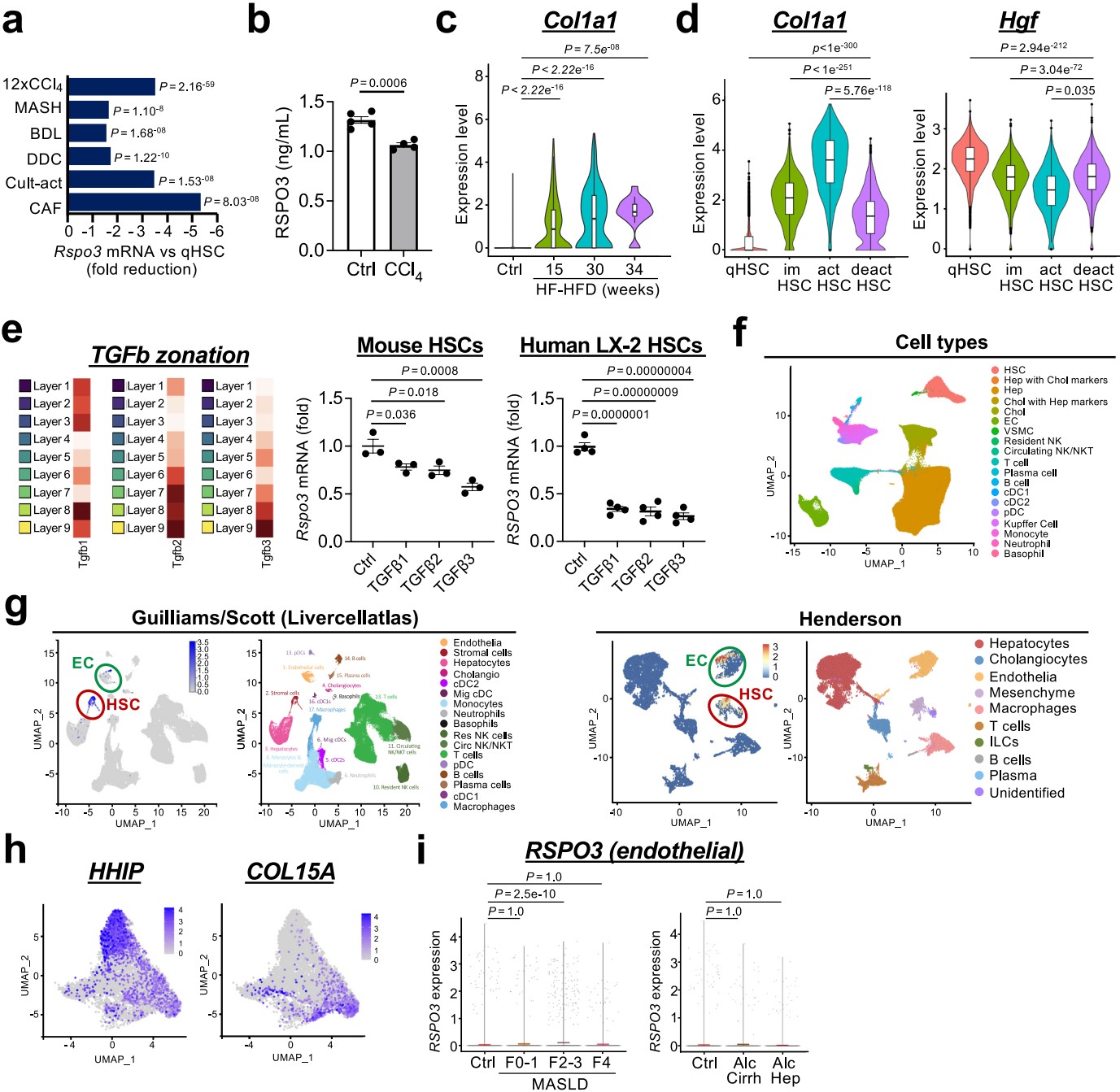

**Extended Data Fig. 11 | Expression and regulation of Rspo3. a.** *Rspo3* mRNA in bulk RNA-seq from mouse HSCs isolated from different fibrosis models. qHSC, quiescent HSC. **b.** Determination of RSPO3 protein in liver extracts from healthy mice (*n* = 5) and mice after treatment with 40xCCl₄ injections (*n* = 4) by ELISA. **c.** scRNA-seq data showing *Col1a1* expression from mice on HF-HFD at different stages of MASLD (related to Fig. 5b). **d.** scRNA-seq showing *Col1a1* and *Hgf* mRNA in quiescent, activated and deactivated mouse HSCs (related to Fig. 5c). **e.** Zonation of TGFB1, TGFB2 and TGFB3 (left panel) from Xu et al, Nat Genet 2024 56:953-969) and qPCR for *Rspo3* in primary mouse HSCs (*n* = 3/group) and *RSPO3* in human LX-2 HSCs (*n* = 4/group) after treatment with rhTGFb1, rhTGFb2 and rhTGFb3 (right panel). **f.** UMAP showing cell annotations in human snRNA-seq from healthy, MASLD and ALD livers (related to Fig. 5e). **g.** sc/snRNA-seq showing *RSPO3* expression and cell annotations in human livers from the indicated dataset. **h.** UMAP showing quiescence marker HHIP (higher in cyHSC) and activation marker *COL15A1* (high in myHSC) in human snRNA-seq (related to Fig. 5g). **i.** *RSPO3* mRNA in endothelial cells from snRNA-seq data of healthy controls (Ctrl), MASLD and patients with alcoholic cirrhosis or alcoholic hepatitis (mean + 95% confidence interval in purple). Data are shown as mean ± s.e.m. Each dot represents one biological replicate (**b,e**). In the violin plots, box plots represent the interquartile range (IQR), Q1, median and Q3, whiskers as minimum (Q1-1.5xIQR) and maximum (Q3 + 1.5xIQR), and outlier data as individual dots. Each data point represents one cell (**c,d**). P values were calculated using one-way ANOVA followed by Dunnett's multiple comparisons test (**a,e**), unpaired two-tailed t-tests (**b**) or two-way ANOVA followed by Tukey's multiple comparison test (**c,d,i**).

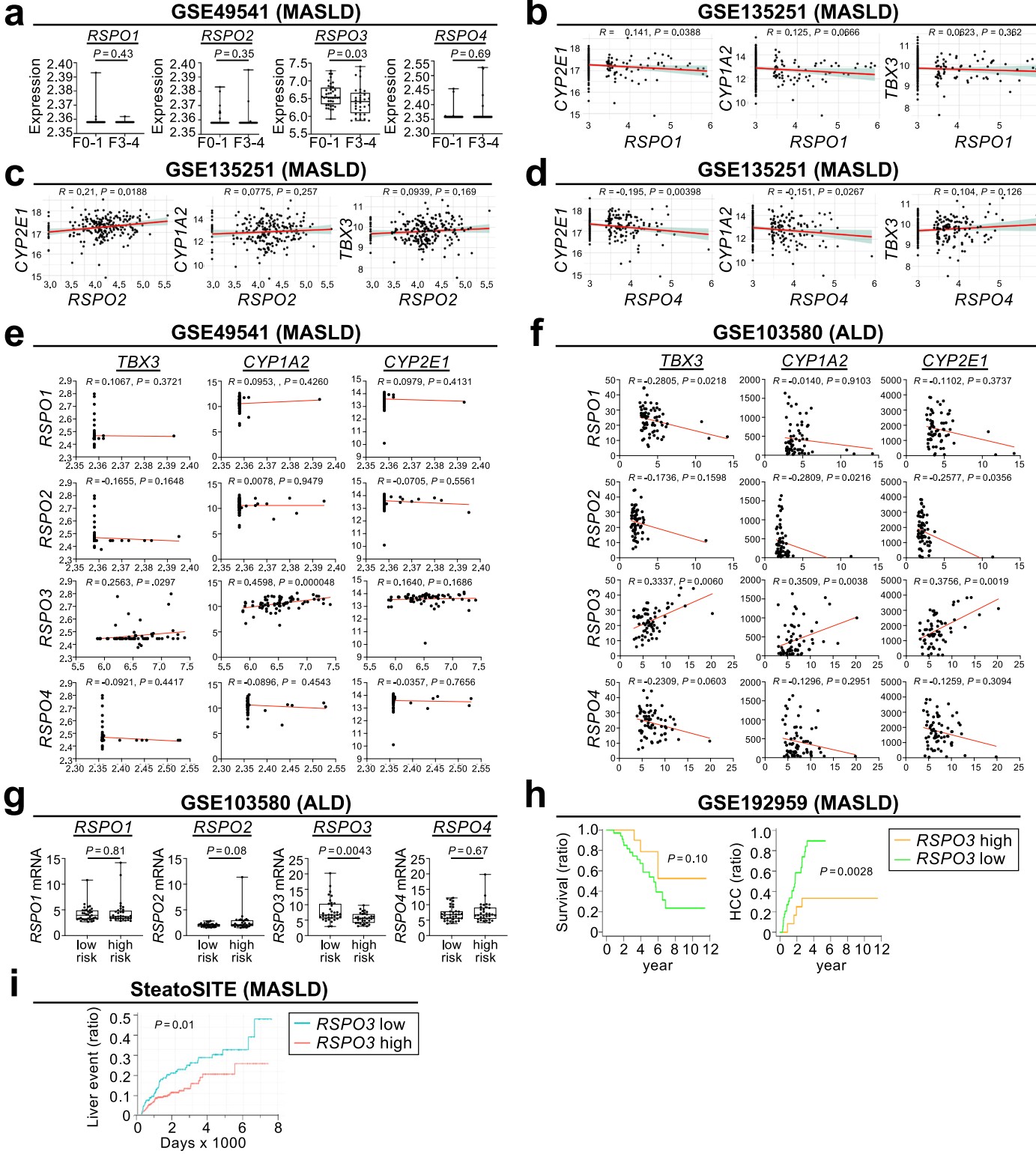

**Extended Data Fig. 12 | Expression and correlation with WNT target gene expression and outcomes of R-spondins in human cohorts with MASLD and ALD. a**. Expression of *RSPO1*, *RSPO2*, *RSPO3* and *RSPO4* in the GSE49541 MASLD cohort. **b-d**. Correlation of *RSPO1* (**b**), *RSPO2* (**c**) and *RSPO4* (**d**) mRNA with WNT target genes *CYP2E1*, *CYP1A2* and *TBX3* in the GSE135251 MASLD cohort. **e-f**. Correlation of *RSPO1*, *RSPO2*, *RSPO3* and *RSPO4* mRNA with the expression of WNT target genes *TBX3*, *CYP1A2* and *CYP2E1* in the GSE49541 MASLD (**e**) and the GSE103580 ALD (hASWX) (**f**) cohorts. **g**. Expression of *RSPO1*, *RSPO2*, *RSPO3*

and *RSPO4* in the GSE103580 ALD cohort. **h**. Survival and HCC development stratified by *RSPO3* expression in the GSE192959 cohort. **i**. Liver-related coding events by *RSPO3* expression in the SteatoSITE MASLD cohort. Box plots represent the interquartile Range (IQR), Q1, median and Q3, each data point represents one individual (**a,g**). P-values were calculated using Wilcoxon rank-sum test (**a,g**) and log-rank test (**h,i**). Correlations were evaluated by pearson correlation coefficient (**b-f**).

# Reporting Summary

## Statistics

For all statistical analyses, confirm that the following items are present in the figure legend, table legend, main text, or Methods section.

| n/a | Confirmed | |
|---|---|---|
| ☐ | ☒ | The exact sample size (*n*) for each experimental group/condition, given as a discrete number and unit of measurement |
| ☐ | ☒ | A statement on whether measurements were taken from distinct samples or whether the same sample was measured repeatedly |
| ☐ | ☒ | The statistical test(s) used AND whether they are one- or two-sided<br>*Only common tests should be described solely by name; describe more complex techniques in the Methods section.* |
| ☒ | ☐ | A description of all covariates tested |
| ☐ | ☒ | A description of any assumptions or corrections, such as tests of normality and adjustment for multiple comparisons |
| ☐ | ☒ | A full description of the statistical parameters including central tendency (e.g. means) or other basic estimates (e.g. regression coefficient) AND variation (e.g. standard deviation) or associated estimates of uncertainty (e.g. confidence intervals) |
| ☐ | ☒ | For null hypothesis testing, the test statistic (e.g. *F*, *t*, *r*) with confidence intervals, effect sizes, degrees of freedom and *P* value noted<br>*Give P values as exact values whenever suitable.* |
| ☒ | ☐ | For Bayesian analysis, information on the choice of priors and Markov chain Monte Carlo settings |
| ☒ | ☐ | For hierarchical and complex designs, identification of the appropriate level for tests and full reporting of outcomes |
| ☐ | ☒ | Estimates of effect sizes (e.g. Cohen's *d*, Pearson's *r*), indicating how they were calculated |

*Our web collection on statistics for biologists contains articles on many of the points above.*

## Software and code

Policy information about availability of computer code

| Data collection | Bulk and single cell RNA-sequencing data was generated from frozen mouse or frozen human livers as described in the method section. The following instruments were used for data collection: NovaSeq 6000 (Illumina); FluorChem M system instrument (ProteinSimple); QuantStudioTM5 Real-Time PCR System (Applied Biosystems); NanoDropTM 1000 Spectrophotometer (Thermo Scientific); Hybridization System (ACDbio); Bioanalyzer 2100 (Agilent Technologies); iMarkTM Microplate Reader (BioRad). |
|---|---|
| Data analysis | Details for data analysis are described in the Material and Methods. The following softwares/packages were used for data analysis: Kallisto (0.44.0); Cell Ranger (v3.1.0); CellBender (v.0.2.0); R (v4.0.2); RStudio (2023.12.0 build 369); R package survminer (0.4.9); R package maxstat (v.0.7-25); R package DESeq2 (v 1.40.2); R package Seurat (v5.0.1); R package ggplot2 (v3.4.4); R package dplyr (1.1.4); WebIDQ software; Limma; GraphPad Prism (v.9.0); FIJI (v2.14.0); ilastik (v1.3.3post3); CellProfiler (v4.2.1); GSEA (v4.3.2); ggplot2 (v 3.4.4);HDI imaging software (v1.6); SCiLS (v2024a);Scrublet; CellphoneDB (v5)<br>The previously published datasets GSE68779, GSE211370, GSE172492, GSE158183, GSE49541, GSE193066, GSE192959, GSE103580 and GSE94397 were used for analysis. R markdown scripts enabling the main steps of the analysis have been deposited into GitHub (https://github.com/SchwabeLabcu/HSC_RSPO3). |

For manuscripts utilizing custom algorithms or software that are central to the research but not yet described in published literature, software must be made available to editors and reviewers. We strongly encourage code deposition in a community repository (e.g. GitHub). See the Nature Portfolio guidelines for submitting code & software for further information.

## Data

Policy information about availability of data

All manuscripts must include a data availability statement. This statement should provide the following information, where applicable:

- Accession codes, unique identifiers, or web links for publicly available datasets
- A description of any restrictions on data availability
- For clinical datasets or third party data, please ensure that the statement adheres to our policy

The snRNA-seq data from 26 human patients and 4 untreated mouse livers with the indicated genotypes (GSE256398); as well as bulk RNA-seq data from HSC-depleted vs control mice, Rspo3ΔHSC mice vs control mice, aged Rspo3ΔHSC mice vs control mice, and Rspo3ΔEC vs control mice (GSE256377) have been deposited.

## Research involving human participants, their data, or biological material

Policy information about studies with human participants or human data. See also policy information about sex, gender (identity/presentation), and sexual orientation and race, ethnicity and racism.

| | |
|---|---|
| Reporting on sex and gender | Sex and gender were not considered in the study design. The respective sex and gender information are detailed in the methods and figure legend and Supplementary Table 10 |
| Reporting on race, ethnicity, or other socially relevant groupings | No further information about race, ethnicity, or other socially relevant groupings are reported. |
| Population characteristics | Human characteristics are reported in Supplementary Table 10 |
| Recruitment | The respective recruitment process is listed in the Methods section. All patients provided written informed consent. Liver samples for alcoholic hepatitis, alcohol-associated cirrhosis and MASLD-associated cirrhosis were obtained from liver explants in patients undergoing liver transplant. All other liver samples from living donors were obtained intraoperatively in patients in whom an intraoperative liver biopsy was indicated on clinical grounds such as during scheduled liver resection, exclusion of liver malignancy during major oncologic surgery, or assessment of liver histology during bariatric surgery. |
| Ethics oversight | The respective ethics committee is listed in the Methods section. Human Liver samples were collected under IRB-approved protocols at the University of Pittsburgh (IRB protocol 19120198), Johns Hopkins University School of Medicine (IRB protocol 00107893), the University of Kiel, Germany (Ethikkommission der Medizinischen Fakultät der Universität Kiel, D425/07, A111/99) or obtained from LifeNet Health, an organ procurement organization (operating under the Anatomical Gift Act) |

Note that full information on the approval of the study protocol must also be provided in the manuscript.

# Field-specific reporting

Please select the one below that is the best fit for your research. If you are not sure, read the appropriate sections before making your selection.

☒ Life sciences  ☐ Behavioural & social sciences  ☐ Ecological, evolutionary & environmental sciences

For a reference copy of the document with all sections, see nature.com/documents/nr-reporting-summary-flat.pdf

# Life sciences study design

All studies must disclose on these points even when the disclosure is negative.

| | |
|---|---|
| Sample size | Pilot experiments and previously published results were used to estimate the sample size such that appropreate statistical tests could yield significant results. For some experiments, results were confirmed using a second cohort.  The sample size and exact n numbers used in the study are indicated in the Methods or in each individual figure/figure legend. |
| Data exclusions | No mice were excluded because they were statistical outlier. However, some mice were excluded of the analysis if they presented sickness not related to the study (abscess, malocclusion of their teeth, infections due to fight, extensive hydronephrosis). Some samples were excluded if they did not meet quality standards, e.g. for qPCR when the house keeping gene 18s was more than 1 cycle lower than the mean, or for IHC when the staining was extremely faint or had high background, e.g. due to bad sample preparation or fixation. Some mouse samples were not analyzed if not enough tissues was collected. |
| Replication | All experiments presented were conducted with sufficient mouse numbers to ensure statistical significance could be reached, particularly for experiments involving tumor studies. Biochemical or image based data were reproduced in multiple mice; e.g. immunostainings, qPCR analysis, measurements of transaminase levels, immunoblotting experiments. All atempts of replicating data were successful. Number of mice, biological replicate and number of experiment represented in the figures are indicated in the figure legends. |

| Randomization | Individual were allocated in different experimental groups based on the expression of the Cre recombinase. Group were designed to have mice with similar weight in each group. To induce hepatocyte gene deletion, half of each litters (Rspo3 fl/fl mice) received AAV8-TBG-Null and the other half received a similar dose of AAV8-TBG-CRE. To induce Rspo3 gene hyperexpression, half of each litters (LratCre iDTRhet) received AAV8-CMV-GFP and the other half received a similar dose of AAV8-CMV-Rspo3. |
|---|---|
| Blinding | For mouse treatment and euthanasia as well as post euthanasia analysis such as (i) quantification by IHC and (ii) determination of gene expression by qPCR, the investigators were blinded. Investigators were not blinded for the analysis of single cell or single nucleus RNA-seq analyses. For immunoblotting, the investigators were not blinding when loading the gel to display the results in a logical way. |

# Reporting for specific materials, systems and methods

We require information from authors about some types of materials, experimental systems and methods used in many studies. Here, indicate whether each material, system or method listed is relevant to your study. If you are not sure if a list item applies to your research, read the appropriate section before selecting a response.

## Materials & experimental systems

| n/a | Involved in the study |
|---|---|
| ☐ | ☒ Antibodies |
| ☐ | ☒ Eukaryotic cell lines |
| ☒ | ☐ Palaeontology and archaeology |
| ☐ | ☒ Animals and other organisms |
| ☒ | ☐ Clinical data |
| ☒ | ☐ Dual use research of concern |
| ☒ | ☐ Plants |

## Methods

| n/a | Involved in the study |
|---|---|
| ☒ | ☐ ChIP-seq |
| ☐ | ☒ Flow cytometry |
| ☒ | ☐ MRI-based neuroimaging |

## Antibodies

| Antibodies used | Description of all antibodies used in the study are provided in the Materials&Methods and listed below:

In vitro studies:
RSPO3 neutralizing antibody (ProteoGenix, PX-TA1446)
Isotype control antibody (ProteoGenix, PTX17885)

Immunostaining studies:
Anti-Ki67 antibody (Abcam, ab16667)
Anti-Cyclin D1 (Abcam, ab134175)
Anti-CYP1A2 (Santa Cruz, sc-53241)
Anti-CYP2E1 (Abcam, ab28146)
Anti-RGN (Thermofisher, PA5-56057)
Anti-HAL (Sigma, HPA038547)
Anti-OAT (antibodies.com, A15120)
Anti-GS (Abcam, ab176562)

Multiplex IHC
Anti-CYP1A2 (Santa Cruz, sc-532410)
Anti-RGN (Thermofisher, PA5-56057)
Anti-GS (Abcam, ab176562)
Anti-CYP2F2 (Santa Cruz, sc-374540)
Anti-HNF4α (Thermofisher, MAI-199)
Anti-E-cadherin (Cell Signaling, 3195)
Anti-Na:K ATPase (Abcam, ab7671)
Anti-Rabbit IgG Alexa750 (Thermofisher, A21039)
Anti-Mouse IgG Alexa 647 (Cell Signaling, 4410S)

Spatial metabolomics study:
Anti-GS (Abcam, ab176562)

Immunoblotting studies:
Anti-ALDH2 (Proteintech, 15310-1-AP)
Aanti-RSPO3 (Proteintech, 17193-1-AP)
Anti-GAPDH (Sigma, G9295)
Anti-b actin (Sigma, A3854)
HRP anti-rabbit (Santa Cruz, sc-2004)

Flow cytometry
anti-CD16/32 (Tonbo, 2.4G2)
anti-CD45 (BD and BioLegend, clone 30-F11)
anti-CD19 (Tonbo, clone 1D3) |
|---|---|

anti-CD3e (Tonbo, clone 145-2C11)
anti-CD4 (BD, clone RM4-5)
anti-CD8a (Tonbo, clone 53-6.7)
anti-NK1.1 (BD, clone PK136)
anti-CD11b (BD, clone M1/70)
anti-CD11c (BD, clone HL3)
anti-F4/80 (Tonbo, clone BM8.1)
anti-Ly6C (BioLegend, clone HK1.4)
anti-Ly6G (BioLegend, clone 1A8)
anti-B220 (BD, RA3-6B2)
anti-CD44 (Biolegend, IM7)
anti-CD64 (Biolegend, X54-5/7.1)
anti-CD80 (Tonbo, 16-10A1)
anti-CD86 (BD, GL1)
anti-VSIG4 (eBioscience, NLA14)
anti-MHCII (Tonbo, clone M5/114.15.2).
anti-CD3e (BD, clone 145-2C11)
anti-TCRβ (BD, clone H57-597)
anti-FOXP3 (eBioscience, FJK-16s)
anti-Ki67 (Thermo, clone SolA15)
anti-granzyme-B (BioLegend, clone QA16A02)

| | |
|---|---|
| Validation | All antibodies are commercially available and have been validated by supplier.<br>The validation information of primary antibodies used for imunostaining and immunoblotting found on supplier's webpage is as follows:<br><br>Anti Ki-67 antibody (Suitable for: Flow Cyt (Intra), IHC-P, WB, mIHC, ICC/IF; Knockout validated; Reacts with: Mouse, Rat, Human; https://www.abcam.com/products/primary-antibodies/ki67-antibody-sp6-ab16667.html)<br>Anti-Cyclin D1 (Suitable for: WB, IP, ICC/IF, IHC-P; Reacts with: Mouse, Rat, Human; https://www.abcam.com/products/primary-antibodies/cyclin-d1-antibody-epr2241-c-terminal-ab134175.html?productWallTab=ShowAll)<br>Anti-CYP1A2 (Suitable for: WB, IP, IF and IHC(P); Reacts with: mouse, rat and human; https://www.scbt.com/p/cyp1a2-antibody-d15)<br>Anti-CYP2E1 (Suitable for: WB, ICC/IF; Reacts with: Mouse, Rat, Rabbit, Human; https://www.abcam.com/products/primary-antibodies/cytochrome-p450-2e1-antibody-ab28146.html)<br>Anti-RGN (Suitable for: WB, IHC; Reacts with:Human, Mouse, Rat; https://www.thermofisher.com/antibody/product/RGN-Antibody-Polyclonal/PA5-56057)<br>Anti-HAL (Suitable for: IHC; Reacts with:Human, Mouse, Rat; https://www.sigmaaldrich.com/US/en/product/sigma/hpa038547?srsltid=AfmBOooa3rc-NNbpKdlt1MwLxU3JZH4lH9kbvhlnQrLx_VM6fJ3s6Mhi)<br>Anti-OAT (Suitable for: WB, IHC, IP; Reacts with:Human, Mouse, Rat; https://www.antibodies.com/ornithine-aminotransferase-antibody-a15120)<br>Anti-GS (Suitable for: mIHC, IHC-Fr, WB, IHC-P; Knockout validated; Reacts with: Mouse, Rat, Human; https://www.abcam.com/products/primary-antibodies/glutamine-synthetase-antibody-epr13022b-ab176562.html?productWallTab=ShowAll )<br>Anti-CYP2F2 (Suitable for: WB, IP, IF and ELISA; Reacts with: mouse and rat; https://www.scbt.com/p/cyp2f2-antibody-f-9)<br>Anti-HNF4α (Suitable for: WB, IHC, IHC (P), ICC/IF, Flow, ELISA, IP, ChIP, FN; Reacts with : Human, Mouse, Rat; https://www.thermofisher.com/antibody/product/HNF4A-Antibody-clone-K9218-Monoclonal/MA1-199)<br>Anti-E-cadherin (Suitable for: WB, IHC, IHC (P), IF, Flow; Reacts with : Human, Mouse; https://www.cellsignal.com/products/primary-antibodies/e-cadherin-24e10-rabbit-mab/3195)<br>Anti-Na:K ATPase (Suitable for: ICC/IF, IHC-P, WB; Reacts with: Mouse, Rat, Rabbit, Human, Pig; https://www.abcam.com/products/primary-antibodies/alpha-1-sodium-potassium-atpase-antibody-4646-ab7671.html)<br>Anti-Rabbit IgG Alexa750 (Suitable for: WB, ICC/IF, Flow; Reacts with: Rabbit; https://www.thermofisher.com/antibody/product/Goat-anti-Rabbit-IgG-H-L-Cross-Adsorbed-Secondary-Antibody-Polyclonal/A-21039)<br>Anti-Mouse IgG Alexa 647 (Suitable for: IF, Flow; Reacts with: Mouse; https://www.cellsignal.com/products/secondary-antibodies/anti-mouse-igg-h-l-f-ab-2-fragment-alexa-fluor-647-conjugate/4410)<br>Anti-ALDH2 (Suitable for: WB, IP, IHC, IF, CoIP, ELISA; Reacts with: Human, Mouse, Rat; https://www.ptglab.com/products/ALDH2-Antibody-15310-1-AP.htm)<br>Anti-RSPO3 (Suitable for: WB, IHC, IF, IP, ELISA; Reacts with:Human, Mouse, Rat; https://www.ptglab.com/products/RSPO3-Antibody-17193-1-AP.htm)<br>Anti-GAPDH (Suitable for: WB; Reacts with: rabbit, canine, rat, hamster, monkey, mouse, turkey, bovine, mink, human, chicken; https://www.sigmaaldrich.com/US/en/product/sigma/g9295)<br>Anti-b actin (Suitable for: WB; Reacts with:sheep, carp, feline, chicken, rat, mouse, Hirudo medicinalis, rabbit, canine, pig, human, bovine, guinea pig; https://www.sigmaaldrich.com/US/en/product/sigma/a3854?srsltid=AfmBOophnMsldF-su_vD-JDiPSVtGISuPLTuS3csBxH9uJHLOsBOIpT2)<br>HRP-anti-rabbit (Suitable for: WB; Reacts with: Rabbit; https://www.scbt.com/p/goat-anti-rabbit-igg-hrp)<br>anti-CD16/32 antibody (Suitable for: FC, IF, IP; Reacts with: Mouse; https://www.fishersci.com/shop/products/pure-ms-cd16-cd32-2-4g2-1mg/501055032)<br>anti-CD45 (Suitable for: FC; Reacts with: Mouse; https://www.biolegend.com/fr-ch/products/brilliant-violet-510-anti-mouse-cd45-antibody-7995)<br>anti-CD19 (Suitable for: Flow Cytometry; Reacts with: Mouse; https://cytekbio.com/products/percp-cyanine5-5-anti-mouse-cd19-1d3?variant=40581196709924)<br>anti-CD3e (Suitable for: Flow Cytometry, IHC, IHC-P, IHC-F, Flow, FN; Reacts with: Human, Mouse; https://www.thermofisher.com/antibody/product/CD3e-Antibody-clone-145-2C11-Monoclonal/45-0031-82)<br>anti-CD4 (Suitable for: Flow cytometry; Reacts with: Mouse; https://www.bdbiosciences.com/en-be/products/reagents/flow-cytometry-reagents/research-reagents/single-color-antibodies-ruo/buv737-rat-anti-mouse-cd4.612844)<br>anti-CD8a (Suitable for: Flow cytometry ; Reacts with: Mouse; https://cytekbio.com/products/apc-anti-mouse-cd8a-53-6-7?variant=40581236555812)<br>anti-NK1.1 (Suitable for: Flow cytometry ; Reacts with: Mouse; https://www.bdbiosciences.com/en-us/products/reagents/flow-cytometry-reagents/research-reagents/single-color-antibodies-ruo/buv395-mouse-anti-mouse-nk-1-1.564144) |

anti-CD11b (Suitable for: Flow cytometry ; Reacts with: Mouse, Human; https://www.bdbiosciences.com/en-us/products/reagents/flow-cytometry-reagents/research-reagents/single-color-antibodies-ruo/bv650-rat-anti-cd11b.563402)
anti-CD11c (Suitable for: Flow cytometry, Immunohistochemistry-frozen, Immunohistochemistry-formalin, Immunohistochemistry-paraffin, Immunohistochemistry-zinc-fixed; Reacts with: Mouse; https://www.bdbiosciences.com/en-us/products/reagents/flow-cytometry-reagents/research-reagents/single-color-antibodies-ruo/buv737-hamster-anti-mouse-cd11c.612796)
anti-F4/80 (Suitable for: Flow Cytometry; Reacts with: Mouse; https://cytekbio.com/products/apc-anti-mouse-f4-80-antigen-bm8-1?variant=40581236424740)
anti-Ly6C (Suitable for: FC, IHC-F; Reacts with: Mouse; https://www.biolegend.com/fr-lu/products/brilliant-violet-510-anti-mouse-ly-6c-antibody-8726)
anti-Ly6G (Suitable for:FC; Reacts with: Mouse; https://www.biolegend.com/nl-nl/products/purified-anti-mouse-ly-6g-antibody-4767?GroupID=BLG7232)
anti-B220 (Suitable for:Flow cytometry; Reacts with: Mouse, Human; https://www.bdbiosciences.com/en-us/products/reagents/flow-cytometry-reagents/research-reagents/single-color-antibodies-ruo/buv496-rat-anti-mouse-cd45r-b220.612950)
anti-CD44 (Suitable for:FC; Reacts with: Mouse, Human; https://www.biolegend.com/en-ie/products/brilliant-violet-650-anti-mouse-human-cd44-antibody-8923)
anti-CD64 (Suitable for:FC; Reacts with: Mouse; https://www.biolegend.com/fr-ch/products/pe-cyanine7-anti-mouse-cd64-fcgammari-antibody-10062)
anti-CD80 (Suitable for:Flow Cytometry; Reacts with: Mouse; https://cytekbio.com/products/fitc-anti-mouse-cd80-b7-1-16-10a1?variant=40581223252004)
anti-CD86 (Suitable for:Flow Cytometry; Reacts with: Mouse; https://www.fishersci.com/shop/products/anti-cd86-clone-gl1-bd-3/BDB563055)
anti-VSIG4 (Suitable for:WB, IHC-F, FC; Reacts with: Human, Mouse; https://www.thermofisher.com/antibody/product/VSIG4-Antibody-clone-NLA14-Monoclonal/17-5752-82)
anti-MHCII (Suitable for:Flow Cytometry; Reacts with: Mouse; https://cytekbio.com/products/violetfluor-450-anti-mouse-mhc-class-ii-i-a-i-e-m5-114-15-2?variant=40581180981284)
anti-CD3e (Suitable for:Flow Cytometry; Reacts with: Mouse; https://www.bdbiosciences.com/en-us/products/reagents/flow-cytometry-reagents/research-reagents/single-color-antibodies-ruo/buv496-hamster-anti-mouse-cd3e.612955)
anti-TCRβ (Suitable for:Flow Cytometry; Reacts with: Mouse; https://www.bdbiosciences.com/en-us/products/reagents/flow-cytometry-reagents/research-reagents/single-color-antibodies-ruo/bv711-hamster-anti-mouse-tcr-chain.563135)
anti-FOXP3 (Suitable for:IHC, IHC-F, ICC/IF, Flow; Reacts with: Bovine, Dog, Cat, Mouse, Pig, Rat; https://www.thermofisher.com/antibody/product/FOXP3-Antibody-clone-FJK-16s-Monoclonal/11-5773-82)
anti Ki-67 (Suitable for:IHC, IHC-P, IHC-PFA, IHC-F, ICC/IF, Flow, FN; Reacts with: Dog, Cynomolgus monkey, Human, Mouse, Non-human primate, Rat; https://www.thermofisher.com/antibody/product/Ki-67-Antibody-clone-SolA15-Monoclonal/56-5698-82)
anti-granzyme-B (Suitable for:ICFC; Reacts with: Human, Mouse; https://www.biolegend.com/nl-be/products/apc-anti-human-mouse-granzyme-b-recombinant-antibody-14429)

# Eukaryotic cell lines

Policy information about cell lines and Sex and Gender in Research

| | |
|---|---|
| Cell line source(s) | The mouse hepatocyte cell line (AML12) used in this study were obtained from ATCC. The human hepatic stellate cell line (LX-2) used in this study was obtained from Millipore. |
| Authentication | The cell lines were authenticated by the morphology. |
| Mycoplasma contamination | The cell lines were sporadically tested for mycoplasma contamination. |
| Commonly misidentified lines (See ICLAC register) | No commonly misindentified cell lines were used. |

# Animals and other research organisms

Policy information about studies involving animals; ARRIVE guidelines recommended for reporting animal research, and Sex and Gender in Research

| | |
|---|---|
| Laboratory animals | All the mice were in a C57Bl/6 background with at least 5 backcrosses the exception of LratCre Wls floxed mice that were in a mixed background C57Bl/6 - 129/Sv, backcrossed twice to C57Bl/6.. <br> The following strains were used for the experiment described in the manuscript: <br> LratCre, Lyve1Cre, Clec4fCre, Mx1Cre, Pdgfrβ-P2A-CreERT2, Cdh5-CreERT2, Tdtomato (TdTom) Ai14 reporter, Rosa26-HBEGF (iDTR), Rspo3 floxed, Wls floxed, Col1a1 floxed, Tgfbr1 floxed, Pdgfrb floxed, Hgf floxed and BALB/c mice. <br> The age for the mice are as below. LratCre/Tdtom/iDTR 7-34 weeks old, LratCre/Rspo3 floxed 7-42 weeks old (these include "aged mice"), Lyve1Cre/Rspo3 floxed 7-16 weeks old, Clec4fCre/Rspo3 floxed 8 weeks old, Mx1Cre/Col1a1 floxed 11-12 weeks old, LratCre/Wls floxed 8-11 weeks old, LratCre/Tgfbr1 floxed 9-10 weeks old, LratCre/Col1a1 floxed 9-11 weeks old, LratCre/Rspo3 floxed/HGF floxed 7-8 weeks old, Pdgfrβ-P2A-CreERT2/Rspo3 floxed 9-18 weeks old, Cdh5-CreERT2/Rspo3 floxed 11-16 weeks od. |
| Wild animals | No wild animals were used in the study. |
| Reporting on sex | All experiments were performed on male with the exception of: <br> -Some Rspo3 floxed mice were females as indicated in the figure legends. |
| Field-collected samples | No field collected samples were used in the study. |
| Ethics oversight | All animal procedures were performed with approval by Columbia University Institutional Animal Care and Use Committee |

| Ethics oversight | (protocolsAC-AABQ5565, AC-AABP3560 and AC-AABQ5566), the local institutional or the Vanderbilt University Institutional Animal Care and Use Committee (protocol M2000054-01) and in accordance with the Guide for the Care and Use of Laboratory Animals; or with approval from the governmental animal care and use committees Karlsruhe, Germany (in accordance with German national guidelines on animal welfare and the regulations of the regional council Karlsruhe under permit number G-251/20). |
|---|---|

Note that full information on the approval of the study protocol must also be provided in the manuscript.

# Plants

| Seed stocks | N/A |
|---|---|
| Novel plant genotypes | N/A |
| Authentication | N/A |

# Flow Cytometry

## Plots

Confirm that:

☒ The axis labels state the marker and fluorochrome used (e.g. CD4-FITC).

☒ The axis scales are clearly visible. Include numbers along axes only for bottom left plot of group (a 'group' is an analysis of identical markers).

☒ All plots are contour plots with outliers or pseudocolor plots.

☒ A numerical value for number of cells or percentage (with statistics) is provided.

## Methodology

| Sample preparation | Liver tissues were mechanically homogenised followed by an enzymatic digestion with 1 mg/ml of collagenase A (Roche, 10103578001) and 0.5 µg/ml DNase I (Roche, 10104159001) in isolation buffer (RPMI 1640, 5% FBS, 1% L-glutamine, 1% penicillin–streptomycin and 10 mM HEPES) for 45 min at 150 r.p.m. at 37°C. Cells were filtered through a 100-µm cell strainer, washed and separated in 2 parts to analyse the myeloid and the lymphocytes cell subsets. For the latter, cells were loaded onto a Percoll gradient (67% overlay with 40%) followed by red blood cell lysis using ammonium-chloride-potassium buffer and stained. |
|---|---|
| Instrument | Samples were analysed using a BD LSR Fortessa cell analyser. |
| Software | Flow cytometry analysis was performed using FlowJo (v.10.10.0). |
| Cell population abundance | For immune cells analysis, all the cells of the sample were sorted or analyzed. |
| Gating strategy | For immune cells analysis:<br>Debris exclusion by FSC-A/SSC-A. Dounlets were excluded using FSC-A/FSC-H, Life/Dead exclusion was performed using Ghost Dye Red 780 cell viavility reagent. Remaining cells were analyzed according to displayed markers and following the gating strategy provided in Supplementary information. |

☒ Tick this box to confirm that a figure exemplifying the gating strategy is provided in the Supplementary Information.

