## [Peer Review file · Nature]

Hepatic stellate cells control liver zonation, size and functions via Rspo3

Corresponding Author: Dr Robert Schwabe

Version 0:

Reviewer comments:

Referee #1

(Remarks to the Author)

In this very interesting manuscript entitled “Hepatic stellate cells control liver zonation, size and 1 functions via R-spondin 3” Sugimoto, Saito, Wang and colleagues uncover a novel and important role of hepatic stellate cells (HSCs) in controlling hepatic WNT signaling, metabolic zonation and regeneration. While studying the role of HSCs in supporting liver regeneration, the authors found significantly less injury following APAP or CCl4 intoxication in mice with depleted HSCs. They observed depressed metabolic function in zone3, including downregulation of the enzymes which metabolize APAP and CCl4, suggesting protection from injury in HSC-depleted mice and highlighting a novel role for HSCs in controlling metabolic zonation. Using elegant cell-type-specific gene deletion and profiling experiments the authors identify RSPO3, rather than WNT ligands, as the HSC-derived factor instructing WNT-pathway-controlled metabolic function. The authors further use patient sample profiling to translate their findings into human pathobiology, showing that RSPO3 levels correlate with survival and lipid metabolism status. Overall, this is an exciting and highly relevant work conducted by a multidisciplinary team of experts in the field. The manuscript is well written, and the conclusions are supported by solid data. However, there are some points that should be addressed before publication.

Major comments

- 1) The authors convincingly demonstrate that HSCs-derived RSPO3 regulates WNT-pathway-regulated metabolic gene expression and metabolic function. Since ectopic activation or inactivation of hepatic WNT signaling had shown indirect regulation of zone1 metabolism, it would be important to also stain for zone1 metabolic enzymes (e.g. HAL and PCK1) and include zone1 metabolic signatures in the gene expression analyses. Presumably, zone1 metabolism will be increased in HSC-depleted mice and mice with HSC-specific RSPO3 deletion due to reduced hepatic WNT signaling. This simple analysis could possibly expand the role of HSCs in regulating metabolic function across all lobular zones and not just in zone3.
- 2) HSC depletion or HSC-specific RSPO3 deletion show significant reduction in CYP1A2 and CYP2E1 staining, but the hepatocytes adjacent to the periportal niche remain positive. This is consistent with significant RSPO3 expression in portal vein endothelial cells and zone1 LSCEs. Lack of an effect on the expression of these enzymes in the EC-specific RSPO3-KO is therefore surprising. Also, the paper from Neil Henderson’s lab (reference 31), which shows RSPO ISH staining in zone3 HSCs, clearly shows substantial RSPO3 in endothelial cells. It is difficult to imagine that HSC-derived RSPO3 but not EC-derived RSPO3 plays a role in metabolic zonation. Maybe improving the staining quality in Figure 3g could highlight reduced expression levels in hepatocytes adjacent to the portal vein (the authors should quantify zonal staining intensity here).
- 3) The authors provide elegant multiplexed ISH, antibody staining, scRNAseq data and re-analyzed multi-omics profiling data in their manuscript. Given the pre-existing work showing RSPO3 expression in ECs, it would be helpful to use one the experiments above to quantify the distribution of RSPO3 per cell type and zone. Moreover, it would be great to check whether HSC-depletion affected RSPO3 expression in ECs, possibly explaining the stronger reduction of WNT-regulated metabolic enzymes in mice with HSC depletion compared with HSC-specific RSPO3 KO mice.

4) Why do only zone3-HSCs express RSPO3 and not HSCs across the lobule? Maybe the authors can use their spatially-resolved scRNAseq HSC dataset to develop a hypothesis or at least speculate in the discussion section. Similarly, the authors show that inactivated HSCs express RSPO3, whereas activated HSCs express downregulated RSPO3 expression. Also here, the scRNAseq data could be further mined to compare these HSC subsets in light of their differential RSPO3 expression status. More detailed comparison of the HSC subsets by GSEA or other analysis could highlight important differences explaining the findings above.

5) The discussion implies that reduction of RSPO3-induced WNT signaling in the liver can be protective in acute liver injuries. This is correct when considering toxic agents activated by WNT-regulated metabolic enzymes. While this may be practical when hepatotoxic drugs, activated by WNT-regulated metabolic enzymes, are given to patients, it is rather unpractical for the most frequent acute injuries (e.g. APAP overdose) since by the time the patients enter the hospital the toxic substance is discontinued. In these patients the pro-regenerative effect of WNT signaling is quite important, so RSPO3 blockade would be counterproductive. This discussion section should be revised accordingly.

Minor comments

6) Figure 3F shows strong CYP1A2 and CYP2E1 staining around the portal vein and weaker staining towards zone2, as expected. In contrast, the stainings in Figure 3g show CYP1A2 in the majority of hepatocytes, even in the control group, and the staining has a lot of background. The stainings should be repeated.

7) Page 12, lines 15-17: "Together, these findings align with the role of hepatocyte b-catenin signaling in promoting acute toxic liver injury³⁴ but protecting from chronic metabolic injury in alcohol-associated liver disease (ALD)^{35,36}". This sentence should be revised according to the discussion on page 16, lines 6-10, where the authors correctly mention that CCl4 and APAP are activated by WNT-regulated metabolic enzymes. The general conclusion about the role of b-catenin signaling in promoting acute toxic liver injury on page 12 is therefore misleading, since this highly depends in the toxin and how it is metabolized.

8) Page 2, lines 21-22: None of the [word missing] represent competing interests.

Referee #2

(Remarks to the Author)

The manuscript by Sugimoto et al explores the function of LRat1+ hepatic stellate cells in liver homeostatic growth and regeneration following a variety of models including the zonal patterning of the parenchyma. The authors propose that RSPO3, a ligand of LGR receptors which modulate Wnt sensitivity through the regulation of RNF43/ZNRF3 mediated Fzd receptor turnover, is produced by HSCs and these potentiate hepatocellular patterning and regeneration.

There are a large number of complex and interesting experiments in this manuscript; however, there are a few comments I have on this manuscript, below:

Major:

1. The concepts that Wnt signalling regulates regeneration and enzymatic/metabolic patterning of the liver is not new, indeed there are several manuscripts showing that RSPO in the liver regulates the zonation and regeneration of liver following damage (PMID: 26655896 and PMID: 27088858). I do agree that there is an interesting advance here suggesting that while endothelial cells of the vasculature can produce RSPO and Wnt ligands, perhaps HSCs perpetuate and enhance this signal to enhance the range of a diffusible ligand.
2. In the first section of the paper the authors show that loss of LRAT1+ HSCs (by DTR) limits the ability of hepatocytes to regenerate following PHx and that CCl4 and Acetaminophen damage is limited. Given the LRAT+ cells are deleted prior to either of these models it is difficult to demonstrate that limited PHx regeneration is due to lack of signals from the HSC rather than due to prior patterning defects. Can the authors hepatectomize the animals first and then delete the HSCs to show that lack of regeneration is directly influenced by HSC-loss?
3. Furthermore, how essential are the HSCs to induce hepatocyte proliferation? Can the lack of HSCs (and indeed RSPO3 later in the paper) be overcome by exogenous HGF/T3 injection? Or is the response completely HSC dependant.
4. In the Acetaminophen and CCl4 treatments, surely the most likely conclusion (based on the remaining paper) is that the loss of metabolic zonation means that the CYP enzymes etc are lost in Zone 3 and therefore CCl4 and Acetaminophen are not processed normally ergo damage is reduced – the authors would need to dissect whether this change in injury is not simply because the animals lacking HSCs are more resistant to zone 3 chemical damage.
5. In the manuscript the authors reasonably show loss of peri-central hepatocyte markers during loss of HSCs, however presumably LRAT1+ HSCs are lost across the lobule? What happens to markers of peri-portal hepatocytes, do they expand or are they also lost? I.e., is there general loss of zonal patterning across the lobule or is this specific to the Wnt responsive cells within peri-venous zone 3.
6. When the HSCs are deleted, what happens to the transcriptional signature of vascular cells which normally produce Wnt ligand or RSPO? Could many of the phenotypes seen be secondary to changes in the vasculature?
7. The identification of Wnt4 and Wnt5a by HSCs is interesting (in addition to RSPO) these are traditionally seen as non-canonical Wnts and not upstream of b-catenin signalling rather activate PCP/Wnt-Ca2+-NFAT signalling (likely in addition to RSPO). Do the authors have evidence that Wnt4 and Wnt5a are playing a role in patterning the hepatocytes or could this be that the vascular endothelial cells express canonical Wnt ligands and that RSPO is increasing the range of which these signals work?
8. In the paper by Jan Tchorz, it appears that different b-catenin responsive genes vary depending on proximity to the CV

where TCF:LEF-GFP reporting mice have a very tight expression zone whereas Cyp enzymes are broader. Can the authors define whether they are losing very highly active Wnt responsive cells or the lower activity cells which are further from the central vein vasculature.

9. In the experiments where the authors knockout RSPO3 in HSCs with CreERT have they shown that tamoxifen itself doesn't induce the changes in protein and gene expression described? There are a number of reports showing that tamoxifen induces changes in hepatocyte genes.

As a general note, it is clear that the authors have been very thorough and worked to replicate their data across a number of models. However there are points where the paper becomes quite repetitive and could be streamlined to make the story clearer.

Referee #3

(Remarks to the Author)

This manuscript reports that quiescent HSC are required for Wnt-mediated control of liver zonation and homeostasis. To note, this is in itself is not an entirely novel observation since Trinh and colleagues previously reported this requirement of HSC for hepatocyte Wnt-beta-catenin signalling, zonation and hepatic homeostasis but with the difference they identified neurotrophin-3 as the soluble HSC-derived factor responsible for these functions in the normal developing liver (<https://www.science.org/doi/10.1126/scisignal.adf6696>). Instead, in this manuscript the HSC soluble mediator in the regenerative and liver damage models is proposed as the Wnt enhancer R-spondin 3 (Rspo3). The manuscript, which is well written and technically strong, proposes that depletion of HSC causes a transient blunting of liver regeneration followed by hyperproliferation caused by defective induction of *Ccnd1* and more severe liver damage in response to acute toxic injury due to blunted induction of beta-catenin regulated transcripts for metabolic regulators such as *Cyp2e1*, *Rgn*, *Lect2* and *Gul*, and upregulation of the lipid catabolism suppressor *Cyp2f2*. Many of these effects are phenocopied in mice in which *Rspo3* is genetically deleted in HSC. Loss of *Cyp2e1* which was confirmed at protein level at least partially explains the more severe toxic damage observed in HSC deleted livers. *Rspo3* transcript expression is then shown to be dynamically regulated in models of chronic liver damage (toxic and metabolic models), with a progressive decline in expression which was reversed upon cessation of injury, these effects shown to be regulated by TGF-beta. To translate these findings the authors, show with observational data that RSPO3 transcript is decreased in human MASLD and in the livers of patients with alcoholic cirrhosis and hepatitis. Interestingly higher levels of RSPO3 correlated with reduced mortality in these conditions albeit this effect being quite modest in MASLD.

Major comments:

1. What is the explanation for the transient effect of HSC depletion on hepatocyte regeneration? This is not addressed in the manuscript. At day 2 post-hepatectomy there is a profound suppression of Ki67+ hepatocytes (Fig 1b) which at day 5 appears to be over-compensated with a hyperproliferative response (Ext Data 1f). Is this dynamic effect also seen in the TCPOBOP model? Is it seen in the Jedi depleted mice and in *Rspo3* KO HSC mice where it appears that the effects on regeneration, as determined by Ki67 stain and *Ccnd1* expression (Fig 4a) are less impressive than seen with depletion of HSC. I do not see liver mass or liver/body weight data for any of these experiments, these data seem important to include. Could the more profound effect of HSC loss be due to some lasting toxic consequence of killing these cells in the DTR model? Does exogenous delivery of recombinant R-spondin 3 rescue the initial suppression of regeneration and/or suppress the subsequent hyperproliferative effect?

2. The paper by Trinh suggested that neurotrophin-3 is important for mediating the stimulatory effects of HSC on liver homeostasis and zonation. Given the authors have phenocopied the effects of DTR deletion of HSC with the Jedi deletion model used by Trinh it is surprising that the potential contribution of HSC-derived neurotrophin-3 has not been investigated, ideally by targeted genetic deletion so as to have technical parity with the *Rspo3* investigations.

3. The authors have not documented effects of HSC deletion or loss of HSC-expressed *Rspo3* on the immune responses to hepatectomy or liver damage. They cannot therefore rule out an indirect mechanism rather than the proposed HSC/*Rspo3*-Hepatocyte/*Lgr4* model. The requirement for hepatocyte *Lgr4* is also not formally tested and as such the mechanism suggested by the authors is speculative.

4. The dynamic changes in expression of *Rspo3* are critical to the conclusions of the paper, however the authors have not shown data for expression of the R-spondin 3 protein and have also not formally proven that secretion of biologically active R-spondin 3 is dynamically controlled or are selective for HSC in the model. It is well established that changes at the transcript level are often not matched by a similar change at the protein level or that a soluble factor will become available in the extracellular space at a higher or lower level simply because the transcript or protein levels in the cell change. It is therefore critical to this manuscript that the authors demonstrate in as many ways as possible (IHC, IMC, Western blot, functional assays etc) that the dynamic changes in *Rspo3* transcript correlate with changes in protein expression and biologically active R-spondin 3 in the extracellular compartment. In respect to this Zhang et al (<https://doi.org/10.1371/journal.pone.0229445>) report that R-spondin 3 protein levels increase in the CCl4 model and R-spondin 3 is expressed in HSC, hepatocytes and inflammatory cells, including macrophages. In human MASLD, R-spondin 3 increases with NASH grade, and again is expressed in HSC hepatocytes and immune cells when assessed by IHC. They also show that R-spondin 3 promotes fibrosis, as neutralisation with an anti-RSPO3 antibody, OMP-131R10, attenuated CCl4 and bleomycin induced fibrosis. This major discrepancy is not addressed.

Referee #4

(Remarks to the Author)

In this study, the authors used HSC-depleted mice as a model to investigate how HSCs contribute to liver regeneration and injury. They determined the critical role of HSCs in regulating the expression of liver zonation markers through WNT/ β -catenin signaling. In addition, they identified HSC derived R-spondin 3 as regulator of WNT signaling and liver zonation. The authors have previously developed HSC depletion methods and used these models to study the roles of HSCs in hepatocarcinogenesis (Nature 2022 Oct;610:356-365). In the current study, the authors further defined the functions of HSCs under the physiological and pathological conditions and identified R-spondin 3 as an important mediator for HSC functions in controlling liver zonation, size and functions.

Here are some comments.

1. In the paper, the authors claim that HSCs control liver zonation; however, the evidence for this conclusion is a little bit weak. In the manuscript, the authors mainly focused on how HSC depletion downregulated the expression of several zone 3 genes but have not examined whether depletion of HSCs affects the expression of zone 2 and zone 1 genes. These data suggest that HSCs are required for the expression of some zone 3 genes, more data are required to make the conclusion that HSCs control liver zonation or that HSCs specifically control zone 1 gene expression.
2. Another major finding in this paper is an important function of HSC-derived R-spondin 3 in controlling liver zonation and functions. As stated by the authors, HSC derived R-spondin 3 is critical for the zonation marker (Cyp2e1, Cyp1a2) expression in pericentral region. scRNA seq data in Fig. 3a revealed a portion of endothelial cells express R-spondin 3, but almost all HSCs express R-spondin 3. The question is whether expression of R-spondin 3 in HSCs has a zonation pattern, and how does the HSC R-spondin 3 affect hepatocytes in different zones?
3. In the paper, the authors provide data suggesting that R-spondin 3 controls liver zonation and repair via the activation of Wnt signaling. However, R-spondin 3 has been well characterized to be a modulator of Wnt signaling, and plays an important role in controlling tissue regeneration and repair (PMID: 31554819, PMID: 24532711), which reduces the enthusiasm of the novelty of the study.
4. Zone 2 has been reported to be the main driving force for liver regeneration after PHx. How about the effects of HSC depletion on the hepatocyte proliferation in each zone?
5. In Fig. 1, the authors examined liver regeneration and toxic liver injury in HSC-depleted mice by using several models including APAP and CCl₄. The authors proposed that HSCs produce soluble factors that protect against hepatocyte injury and promote liver regeneration. Later the authors showed that depletion of HSCs reduced the expression of Cyp2e1, a key enzyme involved in the hepatotoxicity of APAP and CCl₄, which explained why these mice were resistant to APAP and CCl₄ induced liver injury. This should be mentioned in results section.
6. In Fig.3f and g, the expression pattern of Cyp1a2 is not consistent in Rspo3 fl/fl control mice.

Version 1:

Reviewer comments:

Referee #1

(Remarks to the Author)

This is an excellent revision and the authors addressed all my concerns. I am very much looking forward to seeing this story published soon.

Referee #2

(Remarks to the Author)

The authors should be commended for their thorough response to the initial review. Providing clarification and new data.

I would recommend including the TUNEL data in a supplementary to show that there is not a failure of the pericentral zone to become injured (as queried by the authors), but other than this I have no further questions.

Referee #3

(Remarks to the Author)

The authors are to be congratulated on their revised manuscript which is greatly improved and is undoubtedly a technical tour de force. Similar, the point-by-point response document is very impressive and provides a considerable body of additional work. I have no further technical concerns to raise other than to ask whether the lower baseline liver-to-body weight ratio for mice lacking HSC expression of Rspo3 may have influenced some of the findings in the various liver injury models. This phenotype observation might also be included in the manuscript.

The authors may wish to also comment on the work of Zhou Y et al recently published in Metabolism (V154, May 2024) which suggests that Rspo3 in NASH perturbs liver zonation. How does this report impact on novelty of the authors study?

The relevance of HSC-derived Rspo3 in liver homeostasis is only experimentally tested in mice. The authors should at least discuss this limitation since we are provided with no evidence that Rspo3 protein is differentially expressed by HSC in the damaged human liver or that it has the same physiological role in human liver as described here in mice.

Referee #4

(Remarks to the Author)

The authors have adequately addressed the concerns raised during the review. The manuscript is now significantly improved. I have no further comments.

We would sincerely like to thank the referees for their helpful constructive comments and suggestions. All referees' comments and suggestions have wholeheartedly been addressed through extensive additional experimentation and editorial changes. As a result, the manuscript is very much improved for which we are very thankful!

We prepared this point-by-point response to make review easy and displayed relevant data/figures next to each of the referees' comment – hence, some pieces of data may be shown multiple times in this point-by-point response.

NOTE TO REFEREES: All key data shown in the Figures to Reviewers have been integrated into the manuscript.

REFEREE #1

In this very interesting manuscript entitled “Hepatic stellate cells control liver zonation, size and functions via R-spondin 3” Sugimoto, Saito, Wang and colleagues uncover a novel and important role of hepatic stellate cells (HSCs) in controlling hepatic WNT signaling, metabolic zonation and regeneration. While studying the role of HSCs in supporting liver regeneration, the authors found significantly less injury following APAP or CCl₄ intoxication in mice with depleted HSCs. They observed depressed metabolic function in zone3, including downregulation of the enzymes which metabolize APAP and CCl₄, suggesting protection from injury in HSC-depleted mice and highlighting a novel role for HSCs in controlling metabolic zonation. Using elegant cell-type-specific gene deletion and profiling experiments the authors identify RSPO3, rather than WNT ligands, as the HSC-derived factor instructing WNT-pathway-controlled metabolic function. The authors further use patient sample profiling to translate their findings into human pathobiology, showing that RSPO3 levels correlate with survival and lipid metabolism status. Overall, this is an exciting and highly relevant work conducted by a multidisciplinary team of experts in the field. The manuscript is well written, and the conclusions are supported by solid data. However, there are some points that should be addressed before publication.

Response: We would sincerely like to thank the referee for the positive assessment of our work. As spelled out in more detail below, all referee's comments could be translated into experiments, which have very much helped to further improve the manuscript. All key data shown in the **Figures to Reviewers have been integrated into the manuscript.**

Major comments

1) The authors convincingly demonstrate that HSCs-derived RSPO3 regulates WNT-pathway-regulated metabolic gene expression and metabolic function. Since ectopic activation or inactivation of hepatic WNT signaling had shown indirect regulation of zone1 metabolism, it would be important to also stain for zone1 metabolic enzymes (e.g. HAL and PCK1) and include zone1 metabolic signatures in the gene expression analyses. Presumably, zone1 metabolism will be increased in HSC-depleted mice and mice with HSC-specific RSPO3 deletion due to reduced hepatic WNT signaling. This simple analysis could possibly expand the role of HSCs in regulating metabolic function across all lobular zones and not just in zone3.

Response to comment 1: We fully agree with this excellent suggestion. Our previous manuscript had a very limited analysis of zone 1 (spatial transcriptomics, in HSC-depleted and *Rspo3*^{ΔHSC} mice, original Fig.2g and Fig.3i; Cyp2f2 IHC in HSC-depleted mice, original Fig.2d). As suggested by the referee, we have substantially expanded this section:

- As expected, IHC plus zone-specific quantification as well as qPCR show an expansion of zone 1 markers Hal and Cyp2f2 in HSC-depleted and *Rspo3*^{ΔHSC} mice but not in *Rspo3*^{ΔEC} mice (**Figure 1 for Reviewers**).

- We furthermore added a model with predominantly portal injury, induced by allyl alcohol. Consistent with the expansion of zone 1, we find a strong increase in allyl alcohol-induced liver injury in HSC-depleted and *Rspo3*-deleted mice (**Figure 2 for Reviewers**). Zone-specific injury quantification shows an extension of necrosis into the midzone. Thus, the contracted zone 3 appears to cause decreased injury by zone 3 toxins, whereas the expanded zone 1 triggers an increase by zone 1 toxins. We cannot fully explain the extremely strong increase in allyl alcohol-induced liver injury in HSC-depleted and *Rspo3*-deleted mice – there seems to be threshold with a strong increase in injury beyond this threshold. This is similar for our survival study, where we see striking mortality in HSC-depleted and *Rspo3*-deleted mice (see manuscript).

2) HSC depletion or HSC-specific *RSPO3* deletion show significant reduction in *CYP1A2* and *CYP2E1* staining, but the hepatocytes adjacent to the periportal niche remain positive. This is consistent with significant *RSPO3* expression in portal vein endothelial cells and zone1 LSCs. Lack of an effect on the expression of these enzymes in the EC-specific *RSPO3*-KO is therefore surprising. Also, the paper from Neil Henderson's lab (reference 31), which shows *RSPO* ISH staining in zone3 HSCs, clearly shows substantial *RSPO3* in endothelial cells. It is difficult to imagine that HSC-derived *RSPO3* but not EC-derived *RSPO3* plays a role in metabolic zonation. Maybe improving the staining quality in Figure 3g could highlight reduced expression levels in hepatocytes adjacent to the portal vein (the authors should quantify zonal staining intensity here).

Response to comment 2: This comment was extremely helpful as it stimulated us to perform additional experiments, uncovering distinct and zone-specific functions of HSC- and EC-derived *Rspo3* as suggested by the referee and as spelled out in more detail below. However, we would like to emphasize that pericentral, not periportal ECs express high levels of *Rspo3* (Halpern et al, Nat Biotechnol 2018, Dobie et al, Cell Report 2019) and that we see pericentral hepatocytes adjacent to this *Rspo3*⁺ EC niche remaining in the *Rspo3*^{ΔHSC} and HSC-depleted mice:

- As described in the manuscript and summarized by the referee, HSCs and HSC-derived *Rspo3* regulate Wnt target genes in the majority of hepatocytes (*Cyp2e1*, *Cyp1a2*, *Rgn*, *Lect2* etc in zone 2-3; *Hal*, *Cyp2f2* etc in Zone 1). The referee is 100% right in that EC-derived *Rspo3* should also fulfil functions. We do not see changes in *Cyp2e1*, *Cyp1a2* or *Rgn* by IHC (**Figure 3 for reviewers**) or RNA-seq (**Suppl. Table 5**) in *Rspo3*^{ΔEC} mice as these WNT target genes are expressed across zones 2-3 and therefore mostly regulated by HSC-derived *Rspo3*.

- Consistent with the pericentral location of Rspo3+ECs, we find that very pericentral Wnt-regulated target genes GS and Oat are nearly absent in Rspo3^{ΔEC} livers (Figure 4C for Reviewers), but not altered in Rspo3^{ΔHSC} livers or HSC-depleted livers (Figure 4A-B for Reviewers). RNA-seq confirmed this with significant decreases of *Glul*, *Oat*, *Lgr5*, *Slc1a2* and *Slc13a3* in Rspo3^{ΔEC} livers (Suppl. Table 5).

Figure 4 for Reviewers. Regulation of the pericentral zone 3 layers by EC-derived Rspo3. A-C. Zone 3 markers GS and Oat were analyzed by IHC with zone-specific zonation quantification and by qPCR. A significant suppression of these very pericentral zone 1 markers was observed in Rspo3^{ΔEC} livers (C) but not in HSC-depleted (A) and Rspo3^{ΔHSC} (B) livers.

- To highlight the distinct zone-specific effects of HSC- and EC-derived Rspo3, we generated mice with deletion in EC and HSC (LratCre+ Lyve1Cre+ Rspo3^{fl/fl}). We did not include this data in the manuscript as some mice escaped from deletion for reasons we do not currently understand. Mice without escape showed nearly complete absence of Rspo3 and Wnt target genes (*Glul*, *Oat*, *Cyp1a2*, *Cyp2e1*) and lacked the Cyp2e1- and Cyp1a2-positive pericentral rim that remained in Rspo3^{ΔHSC} mice (Figure 5 for Reviewers).

Figure 5 for Reviewers. Absent Rspo3 and Wnt target gene expression in mice with combined deletion of Rspo3 in HSCs and ECs. A. Rspo3 and Wnt target genes were determined in Rspo3^{ΔHSC+EC} livers and Rspo3^{fl/fl} controls. B. Representative images for IHC of Cyp1a2 and Cyp2e1 in Rspo3^{ΔHSC+EC} livers and Rspo3^{fl/fl} controls.

- We subjected Rspo3^{ΔEC} mice to additional disease models, including APAP-induced liver injury, 70% PHx and CDAA-HFD. Consistent with our finding that HSC- but not EC-derived Rspo3 regulated Wnt signaling in the majority of hepatocytes and EC only a few ‘Wnt high’ pericentral hepatocytes, we did not observe altered liver injury or regeneration in Rspo3^{ΔEC} mice (Figure 6 for Reviewers).

Figure 6 for Reviewers. Rspo3^{ΔEC} mice do not display altered liver injury or regeneration. A-C. Rspo3^{ΔEC} mice were subjected to 70% PHx (A), APAP-induced liver injury or 6 weeks of CDAA-HFD (C). Rspo3^{ΔEC} mice did not show alterations in liver injury (ALT, necrosis), proliferation (Ki67 IHC), or steatosis (Oil-Red-O) compared to their Rspo3^{fl/fl} littermates.

- As the focus of our paper is on the non-fibrotic role of HSCs in the liver rather than on Rspo3 expression in ECs, which is long known (e.g. Halpern et al, Nat Biotechnol 2018), we did not follow up with additional functional studies in *Rspo3*^{AEC} mice, but are confident that the pericentral alterations and decreased expression of GS in these mice will also impact liver or systemic health and disease processes, especially in diseases where glutamine exerts a key role such as ammonia detoxification (a topic of ongoing long-term studies). Indeed, decreased detoxification of ammonia and increased encephalopathy has been demonstrated mice with hepatocyte-specific GS knockout (Qvartskhava et al, PNAS 2015;112(17):5521-6).

3) The authors provide elegant multiplexed ISH, antibody staining, scRNAseq data and re-analyzed multi-omics profiling data in their manuscript. Given the pre-existing work showing RSPO3 expression in ECs, it would be helpful to use one the experiments above to quantify the distribution of RSPO3 per cell type and zone. Moreover, it would be great to check whether HSC-depletion affected RSPO3 expression in ECs, possibly explaining the stronger reduction of WNT-regulated metabolic enzymes in mice with HSC depletion compared with HSC-specific RSPO3 KO mice.

Response to comment 3: Thank you for these helpful suggestions.

- Quantification of *Rspo3* across cell types and zones shows the expected pattern of very pericentral ECs with high *Rspo3* expression (as reported by the Henderson and Itzkovitz groups) and a gradient of *Rspo3* expression in HSCs, with high expression in zone 3 and a progressive decline towards zone 1 (**Figure 7 for Reviewers**).
- The data are consistent with other published studies (Dobie et al, Cell Rep. 2019 Nov 12;29(7):1832-1847 [FIGURE REDACTED]; Xu et al, Nat Genet. 2024 May;56(5):953-969) and our sc/snRNA-seq data showing expression in most HSCs but at different levels and one a small subset of ECs with very high *Rspo3* expression (**Figure 8 for Reviewers**).

4) Why do only zone3-HSCs express RSPO3 and not HSCs across the lobule? Maybe the authors can use their spatially-resolved scRNAseq HSC dataset to develop a hypothesis or at least speculate in the discussion section. Similarly, the authors show that inactivated HSCs express RSPO3, whereas activated HSCs express downregulated RSPO3 expression. Also here, the scRNAseq data could be further mined to compare these HSC subsets in light of their differential RSPO3 expression status. More detailed comparison of the HSC subsets by GSEA or other analysis could highlight important differences explaining the findings above.

Response to comment 4: The referee points to an interesting but complex scientific question, which we tackled by multiple approaches.

- As already demonstrated in our previous submission, TGF β , the strongest activator of HSCs, is also the most potent regulator of *Rspo3* expression in mouse and human HSCs. Alterations of TGF β may not only contribute to the different levels of *Rspo3* expression in healthy and fibrotic livers, but also have a role in the zoned expression of *Rspo3*. We found that TGF β 2 and TGF β 3 display a zoned expression in the liver and that

recombinant TGFβ2 and TGFβ3 potently downregulated RSPO3 in primary mouseHSCs and human LX2 HSCs slightly more potently than TGFβ1 (**Figure 9A-B for Reviewers**).

Figure 9 for Reviewers. Potential role of TGFβ2 and TGFβ3 in the zonation Rspo3 zonation. **A.** The Liver Spatio-Temporal Atlas revealed a strong zonation of *Tgfb2* and *Tgfb3* but not *Tgfb1* in the liver (Layer 1 = pericentral, Layer 9 = periportal). **B.** Treatment of primary mouse HSCs or human LX2 HsCs with TGFβ2 or TGFβ3 downregulated *Rspo3* mRNA slightly more potently than TGFβ1. Together, these data suggest that TGFβ2 and TGFβ3 might downregulate *Rspo3* in zone 1.

Figure 10 for Reviewers. Potential role of hypoxia in the regulation of Rspo3 expression. Primary mouse HSCs were subjected to hypoxia for 72h shortly after isolation. **A.** Hypoxia increased *Hif1a* expression. **B.** Hypoxia led to a mild but non-significant increase of *Rspo3* when normalized to b-actin. **C.** Hypoxia led to a significant increase of *Rspo3* when normalized to *Lrat* (b-actin also increased with hypoxia).

- Oxygen represents another key gradient across the lobule, with the pericentral zone being relatively hypoxic (30-35 mm Hg) compared to the periportal zone (60-65 mm Hg). To test whether hypoxia could contribute to the altered *Rspo3* expression, we subjected primary HSCs to hypoxia in a hypoxia chamber, which increased *Hif1a* (**Figure 10A for Reviewers**). We observed a small but non-significant increase of *Rspo3* when normalized to beta-actin (**Figure 10B for Reviewers**). As beta-actin levels increase with hypoxia and activation, we also normalized with HSC markers like *Lrat*, which revealed a stronger and significant increase (**Figure 10C for Reviewers**). While we consider these experiments interesting, they need additional replicates, time points and also functional investigations of how hypoxia-inducible transcription factors such as HIF1a may regulate *Rspo3*. **Due to the preliminary nature of these studies, we opted to not include the data in the manuscript but included a sentence discussing oxygen gradients as one possible regulator of HSC Rspo3 expression.**

- A recent study reported a key influence of nutritional/metabolic gradients in liver zonation (Plata-Gomez *et al*, Nat Commun 2024 Mar 18;15(1):1878). This study used a total parenteral nutrition (TPN) model and showed a fetal and unorganized pattern in livers from piglets receiving TPN (no nutrients entering the normal portal vein route). We analyzed *Rspo3* mRNA expression in these samples and found that *Rspo3* mRNA was strongly decreased (65% reduction) in TPN mice and among the top 10 gene with the most significant alterations (**Figure 11 for Reviewers**). Studies using RNAscope and co-staining of HSCs by IHC are planned but will take a long time (preliminary studies were not successful as samples preps were not ideal for RNAscope). At this time, we cannot determine whether the altered expression is mainly due to changes in HSCs or ECs. Hence, we would like to study this more thoroughly in the future. We have included a sentence discussing nutrient gradients as one possible regulator of HSC *Rspo3* expression and cited the Plata-Gomez *et al*. paper.

TOP 15 genes altered in livers from piglets receiving total parental nutrition (TPN) (Plata-Gomez et al, Nat Commun 2024;15(1):1878.)				
Name	Milk average	TPN average	Ratio TPN/Milk	p-value
1 CTSB	545.507	1811.503	3.321	5.42E-07
2 PSMA6	106.983	191.627	1.791	1.54E-06
3 RB1	21.566	8.416	0.390	5.49E-06
4 DEGS1	40.763	81.509	2.000	6.48E-06
5 CSTB	228.837	656.956	2.871	7.20E-06
6 TPT1	13599.379	23741.446	1.746	7.20E-06
7 FMO1	1554.715	327.526	0.211	8.19E-06
8 INPP1	13.321	30.247	2.271	1.47E-05
9 PEBP4	26.721	3.757	0.141	1.51E-05
10 RSPO3	77.992	27.659	0.355	1.54E-05
11 ANGPTL3	1595.667	961.114	0.602	1.72E-05
12 FBXO6	17.112	52.520	3.069	1.95E-05
13 CTSH	541.989	1145.056	2.113	2.15E-05
14 HSD17B4	1479.599	760.931	0.514	2.36E-05
15 PSIP1	276.866	149.128	0.539	2.62E-05

Figure 11 for Reviewers. Potential role of nutrient gradients in Rspo3 zonation. Shown are the top 15 genes with the most significant changes in liver from newborn piglets that were fed either orally (milk) or parenterally (TPN).

5) The discussion implies that reduction of RSPO3-induced WNT signaling in the liver can be protective in acute liver injuries. This is correct when considering toxic agents activated by WNT-regulated metabolic enzymes. While this may be practical when hepatotoxic drugs, activated by WNT-regulated metabolic enzymes, are given to patients, it is rather unpractical for the most frequent acute injuries (e.g. APAP overdose) since by the time the patients enter the hospital the toxic substance is discontinued. In these patients the pro-regenerative effect of WNT signaling is quite important, so RSPO3 blockade would be counterproductive. This discussion section should be revised accordingly.

Response to comment 5: We completely agree with these comments and have rewritten our manuscript to avoid misunderstanding. Our main point – also emphasized in the last section on *Rspo3* restoration – is that the loss of *Rspo3*

in chronic liver disease is associated with worsened injury, steatosis and fibrosis (in mice) and worsened outcomes (in patients) in chronic metabolic liver disease (MASLD, ALD). We agree that modulating WNT signaling in acute toxic liver disease would be too late, and impractical (plus NAC works the same way). We also completely agree that the pro-generative role of the WNT pathway is extremely important. Finally, it is our belief that CYPs generally perform important functions that promote organismal health. Besides aflatoxin-induced liver injury (which is mild), CYP-mediated toxicity is largely the result of man-made drugs/chemicals that do not exist in Nature. The CYP P450 system has evolved - in the absence of evolutionary pressure from naturally occurring CYP P450-activated hepatotoxins - as a protective system by which ingested toxins are inactivated in the liver and protect sensitive organs such as the brain from damage.

Minor comments

6) Figure 3F shows strong CYP1A2 and CYP2E1 staining around the portal vein and weaker staining towards zone2, as expected. In contrast, the stainings in Figure 3g show CYP1A2 in the majority of hepatocytes, even in the control group, and the staining has a lot of background. The stainings should be repeated.

Response to minor comment 6: We completely agree that the quality was subpar. Besides strong staining in $Rspo3^{\Delta HSC}$ mice (see **Figure 12c for Reviewers**– now Extended Data Fig.8c in the manuscript) we now use constitutive EC deletion (via $Lyve1-Cre$, $Rspo3^{\Delta EC}$) with strongly improved IHC (**Figure 12n for Reviewers** - now Extended Data Fig.8c in the manuscript). We also improved IHC for the inducible EC knockout ($Rspo3^{\Delta EC-ind}$, **Figure 12o for Reviewers** - now Extended Data Fig.8o in the manuscript). The IHC protocols differ slightly between the Schwabe and Augustin labs, hence the staining intensity for Cyp1a2 is not identical but the pattern and the positive area are very similar.

7) Page 12, lines 15-17: “Together, these findings align with the role of hepatocyte b-catenin signaling in promoting acute toxic liver injury³⁴ but protecting from chronic metabolic injury in alcohol-associated liver disease (ALD)^{35,36}”. This sentence should be revised according to the discussion on page 16, lines 6-10, where the authors correctly mention that CCl₄ and APAP are activated by WNT-regulated metabolic enzymes. The general conclusion about the role of b-catenin signaling in promoting acute toxic liver injury on page 12 is therefore misleading, since this highly depends in the toxin and how it is metabolized.

Response to minor comment 7: We have revised the sentence to emphasize its role in specific Cyp P450-activated toxins such as APAP and CCl₄. We fully agree that this is not true for all acute toxic injuries as we saw increased injury by allyl alcohol in HSC-depleted and $Rspo3^{\Delta HSC}$ mice (**Figure 2 for Reviewers**). We have also expanded the discussion as it appears that the majority of Cyp P450-activated toxins are man-made (e.g. APAP and CCl₄ do not occur naturally) and there has been no evolutionary pressure to protect the liver against these. The Cyp P450 is, in our opinion, exerts beneficial functions that improve organismal health and protect other more sensitive organs such as the brain from toxin-induced damage.

8) Page 2, lines 21-22: None of the [word missing] represent competing interests.

Response to minor comment 8: We have corrected the sentence to “None of these represent competing interests”.

REFeree #2

The manuscript by Sugimoto et al explores the function of LRat1+ hepatic stellate cells in liver homeostatic growth and regeneration following a variety of models including the zonal patterning of the parenchyma. The authors propose that RSPO3, a ligand of LGR receptors which modulate Wnt sensitivity through the regulation of RNF43/ZNRF3 mediated Fzd receptor turnover, is produced by HSCs and these potentiate hepatocellular patterning and regeneration. There are a large number of complex and interesting experiments in this manuscript; however, there are a few comments I have on this manuscript, below:

Response: We would sincerely like to thank the referee for the positive assessment of our work. The referee's inspiring comments could be translated to additional experiments, which helped to further improve the manuscript. All key data shown in the **Figures to Reviewers** have been integrated into the manuscript.

1. The concepts that Wnt signalling regulates regeneration and enzymatic/metabolic patterning of the liver is not new, indeed there are several manuscripts showing that RSPO in the liver regulates the zonation and regeneration of liver following damage (PMID: 26655896 and PMID: 27088858). I do agree that there is an interesting advance here suggesting that while endothelial cells of the vasculature can produce RSPO and Wnt ligands, perhaps HSCs perpetuate and enhance this signal to enhance the range of a diffusible ligand.

Response to comment 1: We completely agree with the referee's comment. Our study does not claim that we found novel roles for Wnt in the liver. Rather, the novelty of our study lies in the unexpected roles of HSCs, uncovered in HSC-depleted mice and followed up by studies on *Rspo3* as a key HSC mediator. These potent functions of HSCs, regulating liver zonation, injury, metabolism and regeneration by HSCs, affect key aspects of liver health. Hence, the impact of HSCs on the liver is much greater than previously thought, and distinct from their well-described fibrogenic role.

- Our manuscript now shows that HSC- and EC-derived *Rspo3* play distinct roles and that HSCs are not simply enhancing signals coming from *Rspo3*+ Wnt+ ECs (**Figure 13 for Reviewers**): *Rspo3*-expressing HSCs regulate

1. HSCs regulate the WNT target and Wnt-repressed genes in the majority of hepatocytes

Regulated by HSCs and HSC-derived *Rspo3*

2. ECs regulate only a small number of “WNT high” hepatocytes, located very pericentrally

Regulated by EC-derived *Rspo3*

Figure 13 for Reviewers. Distinct functions of HSC and HSC- and EC-derived *Rspo3* in liver zonation and gene regulation. **A-C.** HSCs and HSC-derived *Rspo3* but not EC-derived regulate Wnt target (*Cyp2e1*, *Cyp1a2*) in the pericentral to midzone and Wnt-repressed genes (*Hal*) in periportal zone, comprising the majority of the liver's hepatocytes. **D-F.** EC-derived *Rspo3* but not HSC or HSC-derived *Rspo3* regulate Wnt target genes (*Glu1*, *Oat*) in a few layers of the most pericentral “Wnt high” hepatocytes, but do not affect the other zones.

Wnt signals in the majority of hepatocyte (together with Wnt ligands, mainly coming from EC, in a “menage a trois”), upregulating target genes in the pericentral to the midzone and suppressing Wnt-repressed target genes in the periportal zone (**Figure 13A-C for Reviewers**). *Rspo3*⁺ ECs only regulate a few layers of “WNT high” pericentral hepatocytes. The most prominent feature of these hepatocytes is their high expression of glutathione synthase (GS). *Rspo3* ko in ECs almost completely wipes out GS expression and also other pericentral genes (*Oat* and others) but does not affect the other zones (**Figure 13D-E for Reviewers**).

- Consistent with the above-discussed strong effects of HSC and HSC-derived *Rspo3* on the majority of hepatocytes in the liver, HSCs HSC-derived *Rspo3* exerts a profound effect on liver size, liver regeneration following 70% PHx, APAP-, *CCl*₄- and allyl alcohol-induced liver injury. Additionally, there are profound effects on ALD and MASLD-induced HCC in *Rspo3*^{ΔHSC} mice (**Figure 14A-E for Reviewers**, additional data in the manuscript). In contrast, liver size, injury, regeneration and MASLD are not altered in *Rspo3*^{ΔEC} mice (**Figure 14F-J for Reviewers**). This is consistent with the very limited effect of EC-derived *Rspo3*, regulating only a few “Wnt high hepatocytes”.

1. HSCs regulate, via *Rspo3*, regulate liver size, metabolism, regeneration or MASLD

2. EC-derived *Rspo3* does not regulate liver size, metabolism, regeneration or MASLD

Figure 14 for Reviewers. Distinct functions of HSC and HSC- and EC-derived *Rspo3* in regulating liver size, injury, regeneration and metabolism. A-H. Mice with HSC-selective deletion of *Rspo3* (*Rspo3*^{ΔHSC}) but not EC-selective deletion of *Rspo3* (*Rspo3*^{ΔEC}) show decreased liver size (**A,E**), blunted regeneration after 70% PHx (**B,F**), decreased liver injury after APAP (**C,G**) and increased metabolic liver injury (**D,H**).

2. In the first section of the paper the authors show that loss of LRAT1+ HSCs (by DTR) limits the ability of hepatocytes to regenerate following PHx and that *CCl*₄ and Acetaminophen damage is limited. Given the LRAT+ cells are deleted prior to either of these models it is difficult to demonstrate that limited PHx regeneration is due to lack of signals from the HSC rather than due to prior patterning defects. Can the authors hepatectomize the animals first and then delete the HSCs to show that lack of regeneration is directly influenced by HSC-loss?

Response to comment 2: We thank the reviewer for this helpful suggestion. We completely agree that this is an important point and have performed experiments suggested by the referee.

- Depletion of HSCs after PHx leads to a similar profound inhibition of liver regeneration as depletion one week before 70% PHx. The depletion one hour after 70% PHx was highly efficient and relatively rapid as demonstrated by strong reductions of *Lrat* and *Hand2* 48h and 72h after 70% PHx. There was a strong trend towards reduced proliferation at 48h (variability is a bit higher in this cohort) and a highly significant reduction

at 72h (**Figure 15 for Reviewers**). These findings are consistent with the key role of HSCs shown in various models in our manuscript. The finding that effects of depletion at the 72h time point were stronger also suggests that some time is required for the absence of HSCs to take full effect (which is expected as HSC mediators and pathways activated by HSCs will persist for a while, depending on their biological half-life).

3. Furthermore, how essential are the HSCs to induce hepatocyte proliferation? Can the lack of HSCs (and indeed Rspo3 later in the paper) be overcome by exogenous HGF/T3 injection? Or is the response completely HSC dependent.

Response to comment 3: This point addresses a key question. A main mechanism of suppressed proliferation – in *Rspo3*^{ΔHSC} as well as in *Ctnnb1*^{ΔHep} mice – appears to be suppressed *Ccnd1* expression. We have followed up on this point via two approaches, which include (i) the suggested T3 stimulation and (ii) *Ccnd1* overexpression. The latter allows to overcome the primary defect rather. Our data suggest that powerful exogenous stimulators like 70% PHX, TCPOBOP or T3 seem to be unable to trigger full hepatocyte proliferation when β-catenin and *Ccnd1* are suppressed.

- After discussing HGF/T3 experiments extensively with George Michalopoulos (who has nearly 4 decades experience in HGF and liver regeneration) and co-author Monga (multiple papers on T3-driven liver regeneration), we decided to test thyroid receptor stimulation alone. HGF treatment would require exceedingly high doses (as most HGF will not reach the liver but bind to ECM when systemically injected). Moreover, HGF injection could lead to a bias as HGF-binding ECM may be lower in HSC-depleted liver (leading to relatively higher levels of HGF). Consistent with previous studies from co-author Monga (Alvarado et al, Gene Expr. 2016; 17:19–34), we find decreased proliferation in HSC-depleted and *Rspo3*^{ΔHSC} mice when treated with thyroid receptor agonist GC-1 (which is more stable than T3). However, the decrease in *Rspo3*^{ΔHSC} mice was not as profound as after 70% PHX or TCPOBOP in *Rspo3*^{ΔHSC} mice (**Figure 16 for Reviewers**).

- To determine if hepatocytes can proliferate in the absence of HSCs, we treated HSC-depleted livers with adenoviral Ccnd1. Notably, both HSC-depleted and *Rspo3* livers display decreased levels of the Wnt target genes *Ccnd1*, which is likely responsible for the lower proliferation of hepatocytes in HSC-depleted, *Rspo3*^{ΔHSC} or *Ctnnb1*^{ΔHep} mice. Indeed, we AdCnd1 induced hepatocyte proliferation in HSC-depleted mice to the same level as in non-depleted controls (**Figure 17 for Reviewers**). These findings demonstrate that hepatocytes are still able to proliferate in the absence of HSCs, are also consistent with the hyperproliferation observed in HSC-depleted and *Rspo3*^{ΔHSC} livers at late time points following 70% PHx or TCPOBOP treatment.

We did not include the GC1 and Ccnd1 data in the very busy manuscript, which already contains 2 models of regeneration/hepatocyte proliferation - but are happy to add these data if the reviewer deems it important.

4. In the Acetaminophen and CCl4 treatments, surely the most likely conclusion (based on the remaining paper) is that the loss of metabolic zonation means that the CYP enzymes etc are lost in Zone 3 and therefore CCl4 and Acetaminophen are not processed normally ergo damage is reduced – the authors would need to dissect whether this change in injury is not simply because the animals lacking HSCs are more resistant to zone 3 chemical damage.

Response to comment 4: Again, the reviewer raises an important point, which we have addressed in multiple models:

- ALD and MASLD promote injury predominantly in zone 3. We observed increased liver injury in both models in *Rspo3*^{ΔHSC} mice, excluding resistance of zone 3 hepatocytes to injury **Figure 18A-B for Reviewers**). Further, TUNEL staining revealed that hepatocyte death indeed occurred in zone 3 in the CDAA-HFD model to at least the same degree in *Rspo3*^{ΔHSC} mice as in *Rspo3*^{fl/fl} mice (**Figure 18C for Reviewers**). We also saw no difference in injury after injection of Jo2 (**Figure 18D for Reviewers**; the data are a bit variable, which is not uncommon for Jo2). **TUNEL and Jo2 data were not included in the manuscript but can be added if deemed important.**
- We additionally investigated allyl alcohol as a zone 3 injury model and found that there is an increase in liver injury in both the *Rspo3*-deleted and HSC-depleted mice (**Figure 2/19 for Reviewers**), corresponding to the expanded zone 3 and arguing against a generally lower susceptibility to injury (albeit in a different zone).

5. In the manuscript the authors reasonably show loss of peri-central hepatocyte markers during loss of HSCs, however presumably LRAT1+ HSCs are lost across the lobule? What happens to markers of peri-portal hepatocytes, do they expand or are they also lost? I.e., is there general loss of zonal patterning across the lobule or is this specific to the Wnt responsive cells within peri-venous zone 3.

Response to comment 5: The original manuscript included limited data on periportal markers (Hal, Cyp2f2 by qPCR, Cyp2f2 IHC and 100 plex spatial transcriptomics). We now have substantially expanded this part with additional staining for periportal markers Hal and Cyp2f2 as well as strictly pericentral markers GS and Oat, both with zone-specific quantification. We analyzed the expression of *Rspo3* across zones, demonstrating a strong pericentral to periportal gradient and high expression in zones 2-3 HSCs, explaining the predominant effects on the pericentral to midzone regions.

- We expanded this analysis significantly, showing zonal expression patterns of *Rspo3* in HSCs and ECs, explaining the zonal effects of *Rspo3* (**Figure 20A-B for Reviewers**).
- Our findings are consistent with published studies from the Henderson group (Dobie et al, Cell Reports 2019) and the Esteban group (Xu et al, Nat Genetics 2024) (**Figure 20C for Reviewers**).
- IHC with zone-specific quantification of periportal markers Hal and Cyp2f2, show a significant expansion of both markers into the midzone (**Figure 20D-F for Reviewers**).

1. Zonation of *Rspo3* in HSCs and ECs

2. Expanded zone 1 in HSC-depleted and *Rspo3*^{ΔHSC} but not in *Rspo3*^{ΔEC} mice

Figure 20 for Reviewers. *Rspo3* zonation in HSCs and ECs and effects on zone 1. A-B. Zonal *Rspo3* expression was determined using spatial transcriptomics data in HSCs (A) and ECs (B). **C.** Similar zonal *Rspo3* expression in HSCs demonstrated in the literature (Dobie et al, Cell Rep 2019, 29:1832-1847; Xu et al, Nat Genet 2024, 5:953-969). **D-F.** HSC-depleted and *Rspo3*^{ΔHSC} mice but not *Rspo3*^{ΔEC} mice show an expanded zone 1, as determined by Cyp2f2 and Hal IHC and zonal quantification. * $p < 0.05$; ** $p < 0.01$; *** $p < 0.001$; **** $p < 0.0001$.

- Consistent with the expansion of zone 1, we find a strong increase in allyl alcohol-induced liver injury in HSC-depleted and *Rspo3*-deleted mice (**Figure 2/19 for Reviewers – see previous page**). Zone-specific injury quantification shows an extension of necrosis into the midzone. Thus, the contracted zone 3 appears to cause decreased injury by zone 3 toxins, whereas the expanded zone 1 triggers increased injury by zone 1 toxins.

6. When the HSCs are deleted, what happens to the transcriptional signature of vascular cells which normally produce Wnt ligand or RSPO? Could many of the phenotypes seen be secondary to changes in the vasculature?

Response to comment 6: Again, this is an important question. We determined the expression of Wnts in bulk RNA-seq data. We also isolated ECs to measure the expression of RSPO3 and WNTS in ECs. We do not see any evidence that changes in EC could cause the observed phenotypes in HSC-depleted or *Rspo3*^{ΔHSC} mice:

- Bulk RNA-seq in HSC-depleted (normal age and aged to 8 months) and *Rspo3*^{ΔHSC} mice showed the expected downregulation of Wnt target genes but did not show changes in Wnts that would explain altered zonation (Figure 21 for Reviewers). As such, most Wnts are either unaltered or slightly increased as shown below, with the exception of *Wnt4*, which was decreased in HSC-depleted livers (expected as it is HSC-enriched).

- To specifically address the question of secondary changes in ECs that might mediate zonation changes independent of HSC-derived *Rspo3*, we isolated EC and performed qPCR. Similar to our RNA-seq data from the liver, we saw no consistent alterations in *Rspo3* or Wnts in ECs that would explain the altered zonation of in *Rspo3*^{ΔHSC} mice (Figure 22 for Reviewers).

- These findings are consistent with our data showing that HSC-depleted and *Rspo3*^{ΔHSC} mice have a remaining rim of pericentral Wnt-regulated genes such as *GS*, *Cyp2e1* and *Cyp1a2* (see Figure 13 for Reviewers in response to comment 1). If the effect of HSC-derived *Rspo3* was via Wnt expression in ECs, then this pericentral rim should have been lost (as in *Lgr4/Lgr5*^{ΔHep} or *Ctnnb1*^{ΔHep} mice).
- Our snRNA-seq and CellPhoneDB analyses, which showed high expression of *Rspo3* in HSCs and ECs and of *Lgr4* in hepatocytes and identified *Rspo3*-*Lgr4* as one of the top ligand-receptor pairs in HSC-hepatocyte

communication (**Figure 23 for Reviewers**), suggest a direct mechanism via Rspo3-Lgr4 rather than indirect effects via secondary changes in vascular cells. Indeed, *Lgr4*^{Hep} mice (Planas-Paz, Nat Cell Biol. 2016,5:467-79) show alterations in zonation as in HSC-depleted and *Rspo3*^{HSC} mice. In conjunction, these studies support a direct interaction between HSCs and hepatocytes via Rspo3-Lgr4 rather than indirect mechanisms.

7. The identification of Wnt4 and Wnt5a by HSCs is interesting (in addition to RSPO) these are traditionally seen as non-canonical Wnts and not upstream of b-catenin signalling rather activate PCP/Wnt-Ca²⁺-NFAT signalling (likely in addition to RSPO). Do the authors have evidence that Wnt4 and Wnt5a are playing a role in patterning the hepatocytes or could this be that the vascular endothelial cells express canonical Wnt ligands and that RSPO is increasing the range of which these signals work?

Response to comment 7: The high expression of Wnt4 is indeed intriguing. As suggested by the referee, data from the literature, in conjunction with our studies, provide strong evidence that canonical Wnt ligands from ECs act as key regulators of β -catenin signaling in the liver and work together with Rspo3 from HSCs (regulating the majority of hepatocytes in the liver) and Rspo3 from ECs (regulating the most pericentral “WNT high” hepatocytes). However, our data from HSC-specific Wntless knockout mice do not support a major role for HSC-derived Wnts, such as Wnt4 and Wnt 5a, in patterning:

- Despite highly efficient knockout of Wls, we did not see effects on zonation, determined by IHC for Cyp2e1 and Cyp1a2 and qPCR for Cyp2e1, Cyp1a2, Axin2 and Lect2 (**Figure 24 for Reviewers**). Moreover, there were no differences in APAP-induced liver, regeneration after 70% PHx or the liver-body weight ratio (**Figure 24 for**

Reviewers) – all these parameters/responses are regulated by the Wnt pathway and altered in *Ctnnb1^{ΔHep}*, HSC-depleted and *Rspo3^{ΔHSC}* mice. Consistent with these findings, previous studies identified Wnts from ECs as regulators of liver zonation (using EC-specific knockout of *Wls*, *Wnt2*, *Wnt9b*). Thus, Wnt ligands from HSCs do play a role in patterning. It is possible that *Wnt4* and *Wnt5a* in HSCs are fulfilling other functions. Co-author Monga plans a HSC-specific *Wnt4* ko but this will require 14-16 months as mice are currently cryopreserved.

8. In the paper by Jan Tchorz, it appears that different b-catenin responsive genes vary depending on proximity to the CV where TCF:LEF-GFP reporting mice have a very tight expression zone whereas Cyp enzymes are broader. Can the authors define whether they are losing very highly active Wnt responsive cells or the lower activity cells which are further from the central vein vasculature.

Response to comment 8: We are thankful for this suggestion – also brought up by referee #1. We have analyzed this in depth and already discussed this in our response to point 1 for this referee.

- In additional experiments, we demonstrate that HSC-derived *Rspo3* regulates the “lower activity cells” (expressing *Cyp1a2* and *Cyp2e1*) with larger distance to the central vein, whereas EC-derived *Rspo3* regulates the “very highly active Wnt responsive cells” (expressing *GS* and *Oat* - similar to TCF:LEF-GFP+ cells) surrounding the central vein (**Figure 13 for Reviewers**). The “lower activity cells”, regulated by HSC-derived *Rspo3*, represent the majority of hepatocytes and regulate liver injury, the majority of liver metabolism and regeneration, resulting in alterations shown for *CCL4*, *APAP*, 70% *PHx*, *TCPOBOP*, *CDAA-HFD* and *Lieber-De Carli EtOH* models. The “high activity cells”, regulated by EC-derived *Rspo3*, are only a very small fraction of all hepatocytes and hence there is no effect of EC-specific ko in *CCL4*, *APAP*, 70% *PHx*, and *CDAA-HFD* models (see response to point 1 and the associated **Figure 14 for Reviewers**).

- 9. In the experiments where the authors knockout Rspo3 in HSCs with CreERT have they shown that tamoxifen itself doesn't induce the changes in protein and gene expression described? There are a number of reports showing that tamoxifen induces changes in hepatocyte genes.

Response to comment 9: We thank the referee for raising the concern about tamoxifen usage and agree that tamoxifen may introduce false-positive effects, including changes of gene expression.

- As tamoxifen is given to both groups of mice, the effects on zonation such cannot be attributed to tamoxifen, but are a consequence of Rspo3 deletion after induction of CreERT activity. A condensed zone 3 and expanded zone 1 is not observed in mice that are Cre-negative despite receiving tamoxifen (**Figure 25A-B for Reviewers**).
- Likewise, the data in Rspo3 inducible HSC knockout mice are consistent with our data with constitutive Rspo3 HSC knockout where tamoxifen was not used (**Figure 25A-B for Reviewers**).
- Finally, zone-specific changes seen in HSC-specific inducible knockout of Rspo3 such as reduced Cyp2e1 and Cyp1a2 in zone 2-3 and increased Cyp2f2 and Hal expression in zone 1 are not observed in the in EC-specific inducible knockout of Rspo3, which also received tamoxifen, thus excluding that tamoxifen cause non-specific changes in zonation (**Figure 25C-D for Reviewers**).
- In summary, zonal changes in protein expression and alterations in gene expression cannot be attributed to tamoxifen as they (i) only occur in Cre+ mice receiving tamoxifen and not in Cre- receiving tamoxifen; (ii) changes are similar in constitutive ko without tamoxifen and tamoxifen-inducible knockout; and (iii) as zone-specific changes in Cyp1a2, Cyp2a1, Cyp2f2 and Hal are found in mice with HSC-specific inducible knockout mice but not in EC-specific inducible knockout mice despite all mice receiving tamoxifen. **We have not included specific data or figures dedicated to tamoxifen in the manuscript – but can add if deemed important.**

- 10. As a general note, it is clear that the authors have been very thorough and worked to replicate their data across a number of models. However, there are points where the paper becomes quite repetitive and could be streamlined to make the story clearer.

Response to comment 10: Thank you – we have further streamlined the manuscript and hope that integration of zone-specific quantifications of gene expression, necrosis and proliferation, CellphoneDB analysis, Rspo3 rescue experiments as well as the distinct functions of HSC- and EC-derived Rspo3 reduced repetition, improved the flow and made the story clearer.

REFEREE #3

This manuscript reports that quiescent HSC are required for Wnt-mediated control of liver zonation and homeostasis. To note, this is in itself is not an entirely novel observation since Trinh and colleagues previously reported this requirement of HSC for hepatocyte Wnt-beta-catenin signalling, zonation and hepatic homeostasis but with the difference they identified neutrotropin-3 as the soluble HSC-derived factor responsible for these functions in the normal developing liver (<https://www.science.org/doi/10.1126/scisignal.adf6696>). Instead, in this manuscript the HSC soluble mediator in the regenerative and liver damage models is proposed as the Wnt enhancer R-spondin 3 (Rspo3). The manuscript, which is well written and technically strong, proposes that depletion of HSC causes a transient blunting of liver regeneration followed by hyperproliferation caused by defective induction of *Ccnd1* and more severe liver damage in response to acute toxic injury due to blunted induction of beta-catenin regulated transcripts for metabolic regulators such as *Cyp2e1*, *Rgn*, *Lect2* and *Gul*, and upregulation of the lipid catabolism suppressor *Cyp2f2*. Many of these effects are phenocopied in mice in which *Rspo3* is genetically deleted in HSC. Loss of *Cyp2e1* which was confirmed at protein level at least partially explains the more severe toxic damage observed in HSC deleted livers. *Rspo3* transcript expression is then shown to be dynamically regulated in models of chronic liver damage (toxic and metabolic models), with a progressive decline in expression which was reversed upon cessation of injury, these effects shown to be regulated by TGF-beta. To translate these findings the authors, show with observational data that *RSPO3* transcript is decreased in human MASLD and in the livers of patients with alcoholic cirrhosis and hepatitis. Interestingly higher levels of *RSPO3* correlated with reduced mortality in these conditions albeit this effect being quite modest in MASLD.

Response: We thank the referee for these positive comments and an excellent summary. In response to the reviewer's excellent comments, we could significantly improve the manuscript. All key data shown in the **Figures to Reviewers have been integrated into the manuscript**. As a general comment, we would like to add that the role of HSCs in metabolic liver zonation was not demonstrated by Trinh et al and not a focus of their paper:

- Trinh et al reported unaltered zone 1 and zone 3 in HSC-depleted mice (Fig.S4F-G,S5H in their manuscript), i.e. no changes in the zones affected by HSC depletion and HSC-derived *Rspo3* in our current study. This was clearly stated in their paper: **"Because staining for GS and E-cadherin or similar markers has been commonly used as an indicator of liver zonation, these findings suggest that zone 1 and zone 3 hepatocytes were preserved in HSC-depleted livers despite decreased Wnt-β-catenin signaling based on gene expression analysis. Our data suggest that assessment of these markers alone might not be sufficient to characterize liver zonation and that more detailed analyses are required."**
- Further, Trinh et al did not report an effect of NTF3 on β-catenin activity and we are not aware of other studies showing NTF-3 as a regulator of hepatic Wnt signaling. Please see further details in our response to point 2.

1A. What is the explanation for the transient effect of HSC depletion on hepatocyte regeneration? This is not addressed in the manuscript. At day 2 post-hepatectomy there is a profound suppression of Ki67+ hepatocytes (Fig 1b) which at day 5 appears to be over-compensated with a hyperproliferative response (Ext Data 1f). Is this dynamic effect also seen in the TCPOBOP model? Is it seen in the Jedi depleted mice and in *Rspo3* KO HSC mice where it appears that the effects on regeneration, as determined by Ki67 stain and *Ccnd1* expression (Fig 4a) are less impressive than seen with depletion of HSC.

Response to comment 1A: The transient effect and compensatory hyperproliferation is a well-established finding in mice with inhibited β-catenin pathway. We now cite two key references and performed the suggested experiments:

- A similar transient effect with late hyperproliferation has been reported in mice with defective β-catenin signaling such as *Lgr4/Lgr5*^{ΔHep} (Planas-Paz et al, Nat Cell Biol. 2016;467-79) and *Ctnnb1*^{ΔHep} mice (Tan et al, Gastroenterology 2006;131:1561-72). Hyperproliferation in *Lgr4/Lgr5*^{ΔHep} and *Ctnnb1*^{ΔHep} mice occurs at d3, d4 and d7 after 70% PHx (**Figure 26 for Reviewers**). The concept that mice with impaired liver regeneration after 70% PHx catch up is likely not specific to the β-catenin pathway as it has also been demonstrated for *Stat3ko* (Haga et al, J Hepatol 2005, 5:799-807) and *db/db* mice (Yamauchi et al, Exp Toxicol Pathol. 2003;54:281-6). Liver regeneration is essential for life and some degree of redundancy with compensatory activation of other pathways may ensure liver function and survival.

- As suggested by the referee, we investigated proliferation in the TCPOBOP model at later time points. The kinetics are slower than after 70% PHx. Importantly, we see lower proliferation in HSC-depleted mice at day 2 and day 5 after TCPOBOP but significant increase in proliferation at d9 (**Figure 27 for Reviewers**).
- As suggested by the referee, we have also determined hyperproliferation in *Rspo3*^{ΔHSC} mice and see a similar extent of hyperproliferation at d5 after 70% PHx as in the HSC-depleted mice (**Figure 28 for Reviewers**).
- We have not been able to study late time points after PHx in the JEDI model due to logistic reasons (these mice are not in the Schwabe and Augustin labs; collaboration with the Lee lab has been limited to the exchange of data and IHC). We trust that we have sufficiently addressed this point with our own experiments and examples from the literature, showing that hyperproliferation is common in mice with suppressed Wnt signaling.

Figure 27 for Reviewers. Determination of proliferation in HSC-depleted mice at late time points after TCPOBOP treatment. One week after DT injection, HSC-depleted iDTRhet and control mice were subjected to a single injection with TPOBOP (3 mg/kg) and euthanized at different time points to perform Ki67 IHC. There was a significant increase in Ki67+ cells in HSC-depleted livers 9 days after TCPOBOP. At this time, the liver-body weight ratio had also caught up.

Figure 28 for Reviewers. Hyperproliferation in *Rspo3*^{ΔHSC} mice at late time points after 70% PHx. *Rspo3*^{ΔHSC} and *Rspo3*^{fl/fl} mice were treated with a single injection of TPOBOP (3 mg/kg) and euthanized 5 day later to perform Ki67 IHC. There was a significant increase in Ki67+ cells in *Rspo3*^{ΔHSC} livers 5 days after TCPOBOP. At this time point, the liver body weight ratio was still decreased compared to *Rspo3*^{fl/fl} mice (expectedly, as hyperproliferation only starts around d5).

1B. I do not see liver mass or liver/body weight data for any of these experiments, these data seem important to include.

Response to comment 1B: We apologize not including this data in most figures and corrected this point in the revised manuscript. The liver/body weight ratio is shown for all experiments as requested (Fig.3g,j; Extended Data Fig.1e,f,g,h,i; Extended Data Fig.2b,c,e,g; Extended Data Fig.5c; Extended Data Fig.7e; Extended Data Fig.8h,i,j,o; Extended Data Fig.9a,b,c,l). We initially had not included the liver-body ratio in all experiments as it is lower in the HSC-depleted and *Rspo3*^{ΔHSC} mice at baseline and interpretation after 70% PHx more difficult than for Ki67 IHC.

1C. Could the more profound effect of HSC loss be due to some lasting toxic consequence of killing these cells in the DTR model?

Response to comment 1C: While we cannot fully exclude this, we consider toxic effects very unlikely:

- To avoid confounding effects of inflammation iDTR-mediated HSC killing, we waited 7 days prior to initiating 70% PHx or TCPOBOP.
- As overexpression of *Ccnd1* (**Figure 17/29 for Reviewers**) or overexpression of *Rspo3* (**Figure 31 for Reviewers**) could completely rescue hepatocyte proliferation in HSC-depleted livers, hepatocytes seem to be able to respond normally – rather than suffering from a “toxic” hit.
- We believe that there are additional HSC mediators promoting hepatocyte proliferation. We tested the role of one candidate, HGF. While HSC knockout of HGF alone did not significantly affect liver regeneration (Filliol et al, Nature 2022), we observed that double knockout (dko) of HGF

Figure 17/29 for Reviewers. Unaltered proliferation after Cyclin D1 overexpression in HSC depleted mice. A. Col or HSC-depleted mice were injected with AdCnd1 (0.3×10^9 GC i.v./mouse) six days after HSC depletion and euthanized 48h after AdCnd1 injection. **B.** HSC depletion was confirmed by qPCR for *Lrat*. Proliferation was determined by IHC for Ki67.

and Rspo3 led to an almost as strong reduction of liver weight and Ki67+ cells in homeostasis and after 70% PHx and TCPOBOP as HSC depletion (**Figure 30 for Reviewers**). We cannot exclude that additional HSC-enriched mediators (e.g. NTF3, epiregulin, BMPs and others) exert additional effects but these are likely weaker. Rspo3 is clearly the most potent HSC mediators and achieves even stronger effects together with HGF.

1D. Does exogenous delivery of recombinant R-spondin 3 rescue the initial suppression of regeneration and/or suppress the subsequent hyperproliferative effect?

Response to comment 1D: We employed AAV8-Rspo3 rather than recombinant Rspo3 (systemic injection may not achieve sufficiently high long-term concentrations as not all Rspo3 will reach the liver; Rspo3 has a short half-life).

- AAV8-Rspo3 rescued liver regeneration after 70% PHx in HSC-depleted mice (**Figure 31 for Reviewers**).

- We also employed AAV8-Rspo3 to determine effects on liver injury and also observed that AAV8-Rspo3 reverted injury in HSC-depleted completely to the level of non-depleted mice (**Figure 32 for Reviewers**).

2. The paper by Trinh suggested that neurotrophin-3 is important for mediating the stimulatory effects of HSC on liver homeostasis and zonation. Given the authors have phenocopied the effects of DTR deletion of HSC with the Jedi deletion model used by Trinh it is surprising that the potential contribution of HSC-derived neurotrophin-3 has not been investigated, ideally by targeted genetic deletion so as to have technical parity with the Rspo3 investigations.

Response to comment 2: We would like to add that Trinh et al did not demonstrate alterations in metabolic zonation but focused on Cyclin D1. It also did not show an effect of neurotrophin-3 (NTF3) on metabolic zonation. We have performed additional analyses and explain why, based on sc/snRNA-seq data, pathway analysis, CellPhoneDB analysis and receptor expression pattern, we do not think that NTF3 is a key mediator that regulates metabolic zonation.

- Trinh et al did not demonstrate spatial alterations in metabolic zonation (e.g. by IHC). Their study showed unaltered zone 1 and zone 3 markers (Fig.S4F-G,S5H) stating: *“these findings suggest that zone 1 and zone 3 hepatocytes were preserved in HSC-depleted livers despite decreased Wnt-β-catenin signaling based on gene expression analysis. Our data suggest that assessment of these markers alone might not be sufficient to characterize liver zonation and that more detailed analyses are required”*. The transcriptomic data in JEDI depletion model is indeed similar to the iDTR model and has been further and more deeply analyzed in this study, followed spatial analysis of zonation in JEDI mice in collaboration with senior author Lee. We consider this a strength as it increases the rigor of study and answers question raised in the Trinh et al paper.
- The Trinh et al paper showed effects of NTF3 on proliferation but not on metabolic zonation: *“We identified neurotrophin-3 (Ntf-3) as an HSC-produced factor that induced the proliferation of midlobular hepatocytes through the activation of tropomyosin receptor kinase B (TrkB). Treating HSC-depleted mice with Ntf-3 restored CCND1+ hepatocytes in the midlobular region and increased liver mass. These findings establish that HSCs form the mitogenic niche for midlobular hepatocytes and identify Ntf-3 as a hepatocyte growth factor.”*
- Trinh et al. did not show effects on liver regeneration, liver injury, ALD or MASLD for Ntf3 or JEDI depletion.
- Our focus on Rspo3 results from **a systematic and unbiased analysis of HSC mediators and pathways** altered in HSC-depleted livers that led us to the Wnt pathway and Rspo3. While we cannot exclude an additional role for HSC-derived NTF3 in zonation, our analyses identified Rspo3-Lgr4 in the top 10 ligand-receptor pairs mediating HSC-hepatocyte communication in mice and humans (**Figure 33A,D for Reviewers**).
- Accordingly, Rspo3 receptors Lgr4 and Lgr5 were highly enriched in hepatocytes (**Figure 33 B,E for Reviewers**).
- Although Ntf3 was enriched in HSCs, its receptors Ntrk1, Ntrk2 and Ntrk3 appear to be largely enriched in HSCs (**Figure 33C,F for Reviewers**), suggesting NTF3 and its receptor mediating autocrine HSC-HSC communication, as suggested by the Friedman group (Wang et al, Sci Transl Med 2023: “An autocrine signaling circuit in hepatic stellate cells underlies advanced fibrosis in nonalcoholic steatohepatitis”).
- Accordingly, NTF3 interactions were not among the top hits in our CellPhoneDB analysis (not in the top 1000 ligand-receptor pairs – whereas Rspo3-Lgr4/Lgr5 were in the top 10, both in mice and human snRNA-seq).

Figure 33 for Reviewers. CellPhoneDB uncovers Rspo3 and its receptors among the top HSC-hepatocyte ligand-receptor interactions. A-C. Analysis of snRNA-seq from mouse livers to determine the top ligand-receptor interactions between HSC and hepatocytes (A), expression of R-spondin 3 and its receptors (B) and neurotrophin 3 and its receptors (C). **D-F.** Analysis of snRNA-seq from human livers to determine the top ligand-receptor interactions between HSCs and hepatocytes (D), expression of R-spondin 3 and its receptors (E) and neurotrophin 3 and its receptors (F).

- Co-author Lee continues to study the NTF3 pathway in her lab. We provided her with LratCre mice to generate mice with conditional ablation of NTF3 in HSCs. It will take her 9-12 months to generate a colony large enough for experiments. These studies are completely separate from studies by the Schwabe and Augustin labs and will likely reveal different and completely unrelated data. We are looking forward to learn about HSC-derived NTF3 in liver biology and pathobiology. We do not anticipate redundant or overlapping functions of Rspo3 and NTF3 and hence do not expect key roles for NTF3 in the regulation of the Wnt pathway or zonation of the liver.

3A. The authors have not documented effects of HSC deletion or loss of HSC-expressed *Rspo3* on the immune responses to hepatectomy or liver damage. They cannot therefore rule out an indirect mechanism rather than the proposed HSC/*Rspo3*-Hepatocyte/*Lgr4* model.

Response to comment 3A: We completely agree with the reviewer that characterization of immune responses is important. We had previously characterized immune cells in HSC depleted mice (via α SMA-TK) with liver injury (Filliol et al, *Nature* 2022) and only found increased neutrophils and Ly6C^{high} Ly6G^{neg} myeloid cells but no alterations in other myeloid and lymphocyte population. We now included an extensive analysis of lymphoid and myeloid cell populations from livers of HSC-depleted and *Rspo3* ^{Δ HSC} mice. We determined immune cell alterations in the normal liver rather than after CCl₄ (already done – Filliol et al), APAP or 70% PHx as the strong alterations in the response to injury and regeneration certainly might reflect secondary changes in immune cells rather than studying a potential contributory role of the immune system (especially in the injury models – less injury will lead to less immune cell recruitment).

- Consistent with our previous data in injured liver (Filliol et al, *Nature* 2022) and also studies performed in cholangiocarcinoma (Affo et al, *Cancer Cell* 2021) and liver metastasis (Bhattacharjee et al, *J Clin Invest* 2021), we did not observe major effects of HSC depletion on hepatic immune cells except for an increase in neutrophils (expected – neutrophils needed to remove dead HSCs, sticking around for some time) and a decrease in dendritic cells (DC) (**Figure 34 for Reviewers**). At this time, we do not fully understand the reasons for the DC decrease but it is known that HSCs express some cytokines that may be relevant for DC recruitment. Please see also the opposite effect on DCs in *Rspo3* ^{Δ HSC} mice (**Figure 35 for Reviewers**), making changes in DC unlikely as a cause of the altered regeneration or injury seen in both HSC-depleted and *Rspo3* ^{Δ HSC} mice.

Immune cell characterization in HSC-depleted livers

Confirmation of efficient HSC depletion in this experiment

Lymphocytes

Myeloid cells

Figure 34 for Reviewers. Fine characterization of immune cells in livers from HSC-depleted mice. iDTR^{wt} and iDTR^{het} mice were euthanized one week after diphtheria toxin injection. Efficient HSC depletion was confirmed by qPCR for *Lrat*. Lymphoid and myeloid cell populations were isolated from the livers and quantified by FACS analysis.

- Similar to HSC-depleted mice, *Rspo3*^{ΔHSC} livers showed no changes in the majority of lymphoid and myeloid cells, except increased cDC1 and B cells (**Figure 35 for Reviewers**). The few changes we observed were distinct from HSC-depleted livers and, at least in part, expected (as there is no HSC killing in *Rspo3*^{ΔHSC} livers).

- As both HSC-depleted and *Rspo3*^{ΔHSC} mice showed similar alterations of zonation, toxic and metabolic liver injury and regeneration but none of the minor alterations in immune cells (most populations were unchanged) were consistent between these mice, **alterations in immune cells cannot explain the effects of HSC depletion or *Rspo3* knockout on homeostasis and on various liver injury and regeneration models.**

3B. The requirement for hepatocyte *Lgr4* is also not formally tested and as such the mechanism suggested by the authors is speculative.

Response to comment 3B: Previous studies by the Tchorz group (Planas-Paz, Nat Cell Biol. 2016;18(5):467-79) demonstrated a key role for *Lgr4* in regulating liver size, zonation and regeneration, using *Lgr4*^{ΔHep}, *Lgr5*^{ΔHep} and *Lgr4*+*Lgr5*^{ΔHep} mice (**Figure 36 for Reviewers**). The reduced liver size and regeneration were highly similar to our findings in *Rspo3*^{ΔHSC} mice and most effects were mediated by *Lgr4* (normal liver size, zonation or regeneration in *Lgr5*^{ΔHep} mice). The reduction of Wnt target genes was seen in all zones as *Lgr4* and *Lgr5* are activated by *Rspo3* from HSCs

and ECs – hence, the zonation looks similar as in our $Rspo3^{\Delta HSC+EC}$ mice. Likewise, the effects on liver size are a bit stronger as $Rspo3^{\Delta HSC}$ mice and the reduction is similar in extent as in $Rspo3^{\Delta HSC+EC}$ mice (**Figure 41 for Reviewers**).

- The Tchorz and Ruffner groups have demonstrated increased metabolic liver injury in $Lgr4+Lgr5^{\Delta Hep}$ mice (Saponara et al, Am J Pathol. 2023 Feb;193(2):161-181), leading high-fat diet-induced exacerbation of hepatic steatosis, injury and fibrosis (**Figure 37 for Reviewers**), similar to $Rspo3^{\Delta HSC}$ mice in our MASLD model.

- As pointed out earlier and to the other reviewers, the novelty of our study does not lie in the demonstration that yet another component of the $Lgr4/Lgr5/\beta$ -catenin pathway exerts similar effects but that the activity of this central pathway is critically regulated by HSCs and HSC-derived $Rspo3$ – which is highly unexpected and reveals a novel and fibrosis-independent role of this cell type in regulating liver size, zonation, metabolism, injury and regeneration via $Rspo3$. This signaling axis affects a wide range of liver disease including MASLD and ALD, as shown in our study.

4A. The dynamic changes in expression of $Rspo3$ are critical to the conclusions of the paper, however the authors have not shown data for expression of the R-spondin 3 protein and have also not formally proven that secretion of biologically active R-spondin 3 is dynamically controlled or are selective for HSC in the model. It is well established that changes at the transcript level are often not matched by a similar change at the protein level or that a soluble factor will become available in the extracellular space at a higher or lower level simply because the transcript or protein levels in the cell change. It is therefore critical to this manuscript that the authors demonstrate in as many ways as possible (IHC, IMC, Western blot, functional assays etc) that the dynamic changes in $Rspo3$ transcript correlate with changes in protein expression and biologically active R-spondin 3 in the extracellular compartment.

Response to comment 4A: We agree that protein expression is important. Yet, detection of $Rspo3$ protein is technically extremely challenging. After screening and optimizing a large number of Western blot antibodies and ELISAs, we could confirm $Rspo3$ protein expression by two independent methods. As suggested by the reviewer, we demonstrated (i) predominant expression of $Rspo3$ in HSCs and ECs, and (ii) demonstrated dynamic changes:

- ELISA demonstrates predominant expression in primary mouse HSCs and, to a slightly lesser degree, in primary mouse ECs but little expression in primary hepatocytes and KCs (**Figure 38B for Reviewers**). Similarly,

Western blot demonstrated expression of $Rspo3$ in primary HSCs and ECs but not in primary Kupffer cells or hepatocytes (**Figure 38B for Reviewers**). This pattern aligns completely with our own and multiple public sc/snRNA-seq data sets (see **Figure 39 for Reviewers**) as well as with our finding that combined deletion of $Rspo3$ in HSCs + ECs nearly eliminates hepatic $Rspo3$ expression, whereas deletion of $Rspo3$ in KCs or hepatocytes has no significant effects (**Figure 41 for Reviewers**).

- To analyze *Rspo3* mRNA in the most unbiased manner, we expanded sn/scRNA-seq analysis to include 7 datasets. The predominant expression in HSCs and ECs in these 7 murine and human datasets closely matches our ELISA and Western blots. (**Figure 39 for Reviewers**).

- As requested, the dynamic alterations of *Rspo3* mRNA were confirmed at the protein level. *Rspo3* protein levels were significantly reduced in liver extracts of CCl₄-treated livers compared to control livers (**Figure 40 for Reviewers**), as determined by ultrasensitive ELISA.
- We did not include IHC analysis as we experienced that IHC *Rspo3* leads to substantial non-specific signals (see comment below).
- To further convince the reviewer, we **deleted *Rspo3* in hepatocytes** (via AAV8-TBG-Cre) and ***Rspo3* in Kupffer cells** (via Clec4f-Cre). By both approaches, we did not observe (i) a reduction of hepatic *Rspo3* mRNA or (ii) alterations in Wnt target genes determined by qPCR by IHC (**Figure 41 for Reviewers**).
- In contrast, mice with deletion of *Rspo3* in HSCs plus ECs showed a near absence of hepatic *Rspo3* mRNA and Wnt target genes (**Figure 41 for Reviewers** – see also comment to Reviewer #1). These findings are consistent with sc/snRNA-seq from us and others, our ELISA, Western blot and functional studies. In conjunction, these data demonstrate HSCs and ECs as the key source of hepatic *Rspo3*.

1. Nearly complete elimination of *Rspo3* and Wnt target genes in *Rspo3*^{ΔHSC+EC} livers

2. Unchanged *Rspo3* and Wnt target genes in *Rspo3*^{ΔHep} livers (AAV8-TBG-Cre)

3. Unchanged *Rspo3* and Wnt target genes in *Rspo3*^{ΔKC} livers (Clec4f-Cre)

Figure 41 for Reviewers. Combined deletion of *Rspo3* in HSCs and ECs nearly eliminates hepatic *Rspo3* mRNA and Wnt target gene expression whereas deletion of *Rspo3* in hepatocytes or KCs exerts no effects. A-I. *Rspo3* was deleted in both HSCs and ECs (A-C, LratCre+ Lyve1Cre+ *Rspo3*^{fl/fl} = *Rspo3*^{ΔHSC+EC} mice), in hepatocytes (D-F, via AAV8-TBG-Cre = *Rspo3*^{ΔHep} mice) or in KCs (G-I, via Clec4fCre = *Rspo3*^{ΔKC} mice). *Rspo3* (A,C,F) and Wnt targets gene mRNA (B,D,H) and protein (C,E,H) were nearly eliminated by HSC+EC knockout of *Rspo3* but not affected by hepatocyte or KC knockout of *Rspo3*.

In summary, we have provided 5 additional approaches to confirm HSCs and ECs as key sources of hepatic *Rspo3*: **1.** ELISA (showing *Rspo3* secretion by HSCs and ECs and dynamic alterations in fibrotic livers); **2.** Western Blot; **3.** Analysis of 4 additional sc/snRNA-seq datasets (now 7 datasets); **4.** Hepatocyte- and KC-specific *Rspo3* deletion; **5.** Combined knock-out of *Rspo3* in HSC and ECs. All data are congruent demonstrating HSCs and ECs as key source of hepatic *Rspo3*.

4B. In respect to this Zhang et al (<https://doi.org/10.1371/journal.pone.0229445>) report that R-spondin 3 protein levels increase in the CCl₄ model and R-spondin 3 is expressed in HSC, hepatocytes and inflammatory cells, including macrophages.

Response to comment 4B: We completely agree with the reviewer that the Zhang et al expression data are hard to reconcile with all available public datasets, our own datasets, functional studies as well as HSC-, EC-, hepatocyte- and KC-specific knockout of Rspo3. We do not like to judge other people's work but need to touch upon some details to address this point and highlight incongruencies and the many complementary approaches we have employed:

- We believe that the expression of Rspo3 in hepatocytes and inflammatory cells, detected by IHC in the Zhang et al paper, represent either an artifact in areas of CCl₄-induced necrosis or Rspo3 bound to these cells after secretion from HSCs and ECs (but we consider the latter unlikely as - based on many RNA-seq datasets - there should a strong downregulation of Rspo3 in injured livers).
- The Rspo3 staining by Zhang et al is only seen in the areas injured by CCl₄ (**Figure 42A-B for Reviewers**). These necrotic areas are very commonly affected by non-specific staining – our labs have >20 years of experience with IHC in the CCl₄ model. Further, nearly every CD45+ or F4/80 cell expresses Rspo3 (**Figure 42C for Reviewers**) – we consider this very unlikely as Rspo3 is very lowly expressed and extremely difficult to detect.
- To the best of our knowledge, the Zhang et al paper in the literature relying on IHC - key papers in the Rspo3 field have avoided IHC and use FISH or Rspo3-GFP: Sigal M, et al. *Nature* 2017;548:451-455; Fischer AS, et al *R, J Clin Invest* 2022;132; Goto N, et al, *Cell Stem Cell* 2022;29:1246-1261 e6; Smillie CS, et al, *Cell* 2019;178:714-730 e22; Cox CB et al, *Sci Immunol* 2021;6; Harnack C et al, *Nat Commun* 2019;10:4368; Kazanskaya O et al, *Development* 2008;135:3655-64).
- The Zhang et al IHC data is not compatible with our data from (i) 7 sn/scRNA-seq datasets (including 4 published datasets from other groups); (ii) two methods of protein detection; (iii) HSC-, EC-hepatocyte- and KC-specific deletion and HSC/EC dual deletion, which all show that hepatocytes and inflammatory cells are not relevant sources of Rspo3, but that HSCs and ECs are. Furthermore, in our own studies using ko mice as controls, we were not able to identify Rspo3 antibodies that detect Rspo3 without non-specific staining.

4C. In human MASLD, R-spondin 3 increases with NASH grade, and again is expressed in HSC hepatocytes and immune cells when assessed by IHC

Response to comment 4C: The detection methods in human MASLD are based on the same IHC discussed in detail above, and show an incredible (nearly 20x) upregulation which contrasts all public dataset:

- Bulk RNA-seq data from 3 independent published cohorts (GSE49541, GSE135251, dbGaP phs001807.v1.p1) show in an unbiased manner a progressive decline of Rspo3 in F3-F4 vs F0-F1 patients, which aligns with our snRNA-seq data in MASLD and ALD patients (**Figure 43A-C for Reviewers**). The reduction in Rspo3 expression is most predominant in late disease stages. Likewise, snRNA-seq in two separate cohorts show a strong downregulation in MASLD and ALD (**Figure 43D for Reviewers**). Not only are the data from this large number of public datasets the opposite from the Zhang et al, but the discrepancy is enormous.

- Zhang et al show – in their own set of patients – an up to 20- fold increase in Rspo3 staining as well as $\approx 30\text{-}50\%$ of positive cells in fibrotic septa (**Figure 44 for Reviewers**). This is an extremely strong induction and we do not believe that the extremely large discrepancy to public datasets and our own snRNA-seq data (**Figure 43 for Reviewers**) can be explained by differences between RNA and protein levels.

- We have discussed in detail in the previous paragraph that all major papers on Rspo3 – for seemingly good reasons - use RNAscope or Rspo3-GFP mice for visualization in tissues. In summary, our data are based (i) 3 public bulk RNA-seq datasets in human MASLD/ALD, one snRNA-seq cohort of MASLD and one snRNA-seq cohort of ALD. These data align with downregulation of Rspo3 in HSCs in mice. Moreover, our data in 6 human and mouse sc/snRNA-seq datasets, ELISA, Western blot, HSC+EC Rspo3ko, hepatocyte Rspo3ko and Kupffer cell Rspo3ko all consistently show Rspo3 expression only in HSCs and ECs – contrasting the expression of Rspo3 in HSCs, hepatocytes and macrophages suggested by Zhang et al.

4D. They also show that R-spondin 3 promotes fibrosis, as neutralisation with an anti-Rspo3 antibody, OMP-131R10, attenuated CCl₄ and bleomycin induced fibrosis. This major discrepancy is not addressed.

Response to comment 4D: We would like to emphasize that – in contrast to data on Rspo3 expression - there is complete congruency between in our functional studies and those by Zhang et al on OMP-131R10

- In response to the reviewer's comment, we now performed additional experiments. Our manuscript already included data showing strongly reduced liver injury in Rspo3^{ΔHSC} and Rspo3^{ΔHSC-ind} mice after CCl₄ treatment. Our additional experiments now demonstrate a strong reduction of liver fibrosis in both constitutive HSC ko (Rspo3^{ΔHSC}) and inducible HSC knockout (Rspo3^{ΔHSC-ind}) after chronic CCl₄ treatment (**Figure 45 for Reviewers**).
- These data are similar to the Zhang et al, who observed decreased fibrosis after treatment with Rspo3-blocking antibody OMP-131R10 data.
- While fibrosis and injury were decreased in Rspo3^{ΔHSC}, we observed increased injury and fibrosis in MASLD and ALD in Rspo3 HSC mice, which reflects distinct functions of the β -catenin pathway, with reduced injury in response to Cyp P450-metabolized toxins and increased increased injury and fibrosis in MASLD and ALD - as Ctnnb1^{ΔHep} and Lgr4/5^{ΔHep} mice (discussed and referenced in detail in our manuscript).

REFeree #4

In this study, the authors used HSC-depleted mice as a model to investigate how HSCs contribute to liver regeneration and injury. They determined the critical role of HSCs in regulating the expression of liver zonation markers through WNT/ β -catenin signaling. In addition, they identified HSC derived R-spondin 3 as regulator of WNT signaling and liver zonation. The authors have previously developed HSC depletion methods and used these models to study the roles of HSCs in hepatocarcinogenesis (Nature 2022;610:356-365). In the current study, the authors further defined the functions of HSCs under physiological and pathological conditions and identified R-spondin 3 as an important mediator for HSC functions in controlling liver zonation, size and functions.

Response: We would like to thank the referee for these positive comments. As spelled out in more detail below, all referee's comments could be translated into experiments, which helped us to further improve the manuscript. **All key data shown in the Figures to Reviewers have been integrated into the manuscript.**

1. In the paper, the authors claim that HSCs control liver zonation; however, the evidence for this conclusion is a little bit weak. In the manuscript, the authors mainly focused on how HSC depletion downregulated the expression of several zone 3 genes but have not examined whether depletion of HSCs affects the expression of zone 2 and zone 1 genes. These data suggest that HSCs are required for the expression of some zone 3 genes, more data are required to make the conclusion that HSCs control liver zonation or that HSCs specifically control zone 1 gene expression.

Response to comment 1: We completely agree that deeper analysis is required – this was also suggested by the other referees. Our manuscript had integrated some analysis of Cyp2f2 expression and spatial transcriptomics, suggesting zone 1 expansion. We now expanded this significantly by analyzing additional markers and also quantifying proliferation and necrosis in a zone-specific manner. These data show clearly a contracted zone 3, and expanded zone 1, and functional consequences, i.e. zone-specific alterations of proliferation and cell death:

- Cyp2f2 and Hal IHC and zonal quantification in HSC-depleted and *Rspo3* ^{Δ HSC} mice (**Figure 1/46 for Reviewers**) revealed a significant expansion of the Cyp2f2+ and Hal+ area into the midzone, consistent with qPCR and RNA-seq and similar patterns in mice with inhibited β -catenin signaling (Ctnnb1 ^{Δ Hep} and Lgr4 ^{Δ Hep} mice).

- RNA-seq did not show changes in the midzone markers, but - similar to Trinh et al (Sci Signal 2023) - our manuscript shows a significant downregulation in *Ccnd1* mRNA, which is mostly midzonal.
- Consistent with the expanded zone 1, we observed an expansion of necrosis into the midzone by zone 1-predominant toxin allyl alcohol in HSC-depleted mice and *Rspo3* ^{Δ HSC} mice (**Figure 2/18/47 for Reviewers**).

- The strongest changes in proliferation and cell death as a consequence of HSC depletion or *Rspo3* knockout were seen in the midzone (see **Figures 51 and 52 for Reviewers** in response to point 2 from this referee).

2. Another major finding in this paper is an important function of HSC-derived R-spondin 3 in controlling liver zonation and functions. As stated by the authors, HSC derived R-spondin 3 is critical for the zonation marker (Cyp2e1, Cyp1a2) expression in pericentral region. scRNA seq data in Fig. 3a revealed a portion of ECs express R-spondin 3, but almost all HSCs express R-spondin 3. The question is whether expression of R-spondin 3 in HSCs has a zonation pattern, and how does the HSC R-spondin 3 affect hepatocytes in different zones?

Response to comment 2: The referee raises important points. We analyzed Rspo3 expression across zones and studied the role of HSC-derived Rspo3 (pericentrally to midzonally expressed) and EC-derived Rspo3 (most pericentral).

- Our spatial transcriptomics data shows a clear pericentral to periportal gradient of Rspo3 expression in HSCs with highly significant differences between the zones, as well as a strong expression

Figure 7-8/48 for Reviewers. Zonal expression of Rspo3 in HSCs and ECs. A-B. Spatial transcriptomics data demonstrated pericentral to periportal Rspo3 gradient in HSCs (A) and strong Rspo3 expression in pericentral ECs (B) in mice. C. Rspo3 gradient in HSCs were also reported in studies by Dobie et al (Cell Rep 2019) and Xu et al, (Nat Genet 2024).

- of Rspo3 in zone 3 ECs but not in other zones (**Figure 7-8/48 for Reviewers**). These findings are consistent with previous studies from the Henderson group (Dobie et al, Cell Rep. 2019 Nov 12;29(7):1832-1847) and the spatiotemporal liver Atlas published by the Esteban group (Xu et al, Nat Genet. 2024 May;56(5):953-969) (**Figure 7-8/48 for Reviewers**).

- We find that *Rspo3*^{ΔHSC} mice (i) have strongly-reduced Wnt target genes (*Cyp2e1*, *Cyp1a2*) in the majority of hepatocytes ranging from the pericentral zone to the midzone (**Figure 13/51 for Reviewers**) and (ii) an expansion of the periportal zone with an increase of Wnt-repressed genes such as *Hal* and *Cyp2f2*, extending into the midzone (**Figures 13/49 and 1/46 for Reviewers**).

2. ECs regulate only a small number of "WNT high" hepatocytes, located very pericentrally

Figure 13/49 for Reviewers. Distinct functions of HSC and HSC- and EC-derived Rspo3 in liver zonation and gene regulation. A-C. HSCs and HSC-derived Rspo3 but not EC-derived control Wnt upregulated genes in the pericentral to midzone (*Cyp2e1*, *Cyp1a2*) and Wnt-repressed genes (*Hal*) in periportal zone, comprising the majority of the liver's hepatocytes. D-F. EC-derived Rspo3 but not HSC or HSC-derived Rspo3 regulate Wnt target genes (*Glu1*, *Oat*) in a few layers of the most pericentral "Wnt high" hepatocytes.

- Consistent with the above-discussed strong effects of HSC and HSC-derived Rspo3 on the majority of hepatocytes in the liver, HSC-derived Rspo3 exerts a profound effect on liver size, liver regeneration after 70% PHx and TCPOBOP as well as on APAP-, CCl₄ and allyl alcohol-induced liver injury, MASLD and ALD. In contrast, there is no alteration of liver size, injury or regeneration in Rspo3^{ΔEC} mice (discussed in further detail in our response to comment 3, see also **Figure 14/50 for Reviewers**). The absent impact of EC-derived Rspo3 is consistent with the very limited expression of EC-derived Rspo3 the pericentral area, regulating only a few “Wnt high hepatocytes). Based on previous studies in mice with hepatocyte-specific GS knockout, the main function of EC-derived Rspo3 likely lies in the regulating of hepatic GS expression in order to control systemic ammonia level and protect the brain from toxic effects (known to contribute to hepatic encephalopathy).

1. HSCs regulate, via Rspo3, regulate liver size, metabolism, regeneration or MASLD

2. EC-derived Rspo3 does not regulate liver size, metabolism, regeneration or MASLD

Figure 14/50 for Reviewers. Distinct functions of HSC and HSC- and EC-derived Rspo3 in regulating liver size, injury, regeneration and metabolism. A-H. Mice with HSC-selective deletion of Rspo3 (*Rspo3*^{ΔHSC}) but not EC-selective deletion of Rspo3 (*Rspo3*^{ΔEC}) show decreased liver size (**A,E**), blunted regeneration after 70% PHx (**B, F**), decreased liver injury after APAP (**C,G**) and increased metabolic liver injury (**D,H**).

- To assess functional consequences of altered hepatocyte zonation, we performed zonal analysis of proliferation after 70% PHx or TCPOBOP in HSC-depleted and Rspo3^{ΔHSC} mice. We observed a large reduction of proliferation in the midzone (**Figure 51 for Reviewers**), which is consistent with our findings that the largest zonation changes in HSC-depleted and Rspo3^{ΔHSC} mice were in the midzone (**Figure 13/49 for Reviewers**).

Figure 51 for Reviewers. Zone-specific alterations of hepatocyte proliferation after 70% PHx or TCPOBOP treatment in HSC-depleted and Rspo3^{ΔHSC} livers. A-B. HSC-depleted mice (A) and Rspo3^{ΔHSC} mice (B) and their respective controls were subjected to 70% PHx or treatment with TCPOBOP, followed by Ki67 IHC and zone-specific quantification of proliferation.

- Consistent with the condensed Cyp2e1-positive zone 3, we also observed a significant decrease in pericentral to midzonal necrosis after CCl₄ and APAP in HSC-depleted and Rspo3^{AHSC} mice (**Figure 52 for Reviewers**).

3. In the paper, the authors provide data suggesting that R-spondin 3 controls liver zonation and repair via the activation of Wnt signaling. However, R-spondin 3 has been well characterized to be a modulator of Wnt signaling, and plays an important role in controlling tissue regeneration and repair (PMID: 31554819, PMID: 24532711), which reduces the enthusiasm of the novelty of the study.

Response to comment 3: The main novelty of our manuscript – arising from our initial studies in HSC-depleted mice and later transitioning to Rspo3 as the key HSC mediator – is the unexpected role of HSCs in regulating zonation, injury, metabolism and regeneration in the liver, i.e. key parameters regulating liver homeostasis, MASLD and ALD.

- Our revised manuscript now shows that HSC-derived and EC-derived Rspo3 play distinct roles: Rspo3-expressing HSCs regulate Wnt signals in the majority of hepatocytes, ranging from the pericentral to the midzone. These signals are essential for much of the liver’s zonation (condensed zone 3, expanded zone 1) and control injury, metabolism and regeneration. Rspo3-expressing ECs only regulate a few layers of “WNT high” hepatocytes around the central vein (**Figure 13/49 for Reviewers** in response to major comment 2). The most prominent feature of these hepatocytes is their high expression of glutathione synthase (GS). Rspo3 ko in ECs almost completely wipes out GS expression in the small number of “WNT high” hepatocytes but does not affect the other zones. Consistent with the small number of hepatocytes being regulated by EC-derived Rspo3, toxin-induced liver injury, regeneration and metabolic injury are not altered in Rspo3^{AEC} mice.
- The role for Rspo3 and its receptors as well as mesenchymal-epithelial crosstalk via this pathway are much better investigated in the intestinal niche as indicated by citations from the reviewer (PMID: 31554819: “R-spondin 3 promotes stem cell recovery and epithelial regeneration in the colon”. *Nat Commun.* 2019 Sep 25;10(1):4368), and only starting to be uncovered in the liver. Our findings shed light on how HSCs, via the Wnt pathway, regulate hepatocyte functions and important parameters such as liver size, zonation, metabolism, injury and regeneration (**Figure 14/50 for Reviewers**). As pointed out before, the novelty of our study does not lie in the regulation of the Wnt/beta-catenin pathway by Rspo3 but in the unexpected role of HSCs via this pathway. These findings will shift paradigms on the role of HSCs in liver homeostasis and disease.

4. Zone 2 has been reported to be the main driving force for liver regeneration after PHx. How about the effects of HSC depletion on the hepatocyte proliferation in each zone?

Response to comment 4: We again agree that this is important. In response, we have performed a zonal analysis of proliferation in the 70% PHx and TCPOBOP models. We additionally performed a zonal quantification of proliferation in toxic liver injury models.

- As suggested by the reviewer and shown in previous studies by Hao Zhu’s group, we find strong zonal differences in proliferation and the strongest effects of HSC depletion and Rspo3 deletion in the midzone. Please see our analysis of proliferation in response to point 3 (please see **Figure 51 for Reviewers** on the previous page in response to comment 3).
- Our zonal necrosis analysis also revealed the largest differences in zone 2 in the CCl₄, APAP and allyl alcohol models with decreased midzonal necrosis in HSC-depleted and Rspo3^{AHSC} mice after treatment with pericentral toxins CCl₄ and APAP (a consequence of the condensed Cyp2e1+ zones 2-3); increased midzonal necrosis after allyl alcohol (a consequence of the expanded zone 1). Please see our analysis of cell death in response to comments 1 and 3 (**Figure 2/18/47 and Figure 52 for Reviewers**).

5. In Fig. 1, the authors examined liver regeneration and toxic liver injury in HSC-depleted mice by using several models including APAP and CCL4. The authors proposed that HSCs produce soluble factors that protect against hepatocyte injury and promote liver regeneration. Later the authors showed that depletion of HSCs reduced the expression of Cyp2e1, a key enzyme involved in the hepatotoxicity of APAP and CCL4, which explained why these mice were resistant to APAP and CCL4 induced liver injury. This should be mentioned in results section.

Response to comment 5: We have emphasized this role and added references as suggested by the reviewer.

6. In Fig.3f and g, the expression pattern of Cyp1a2 is not consistent in *Rspo3* fl/fl control mice.

Response to comment 6: We agree that the staining was subpar and have improved our IHC.

- In the revised manuscript, the expression pattern of Cyp1a2 is consistent between the *Rspo3*^{fl/fl} control mice in our cohorts of *Rspo3*^{ΔHSC}, *Rspo3*^{ΔEC} and *Rspo3*^{ΔEC-ind} mice (**Figure 12/53 for Reviewers**). We now employ constitutive EC deletion (via *Lyve1-Cre* = *Rspo3*^{ΔEC}) and the staining for Cyp1a2 is very clear (now Extended Data Fig.8c in the manuscript). We also improved IHC for the inducible EC knockout (*Rspo3*^{ΔEC-ind} - now Extended Data Fig.8o in the manuscript). The IHC protocols differ slightly between the Schwabe and Augustin labs, hence the staining intensity for Cyp1a2 is not identical but the pattern and positive area are very similar.

We would sincerely like to thank the referees for their positive and helpful comments. We have integrated these final comments into the manuscript, as detail below:

REFEREE #1

This is an excellent revision and the authors addressed all my concerns. I am very much looking forward to seeing this story published soon.

Thank you the for these positive and supportive comments!

REFEREE #2

The authors should be commended for their thorough response to the initial review. Providing clarification and new data.

We would like to thank the reviewer for the positive comments and suggestions.

1. I would recommend including the TUNEL data in a supplementary to show that there is not a failure of the pericentral zone to become injured (as queried by the authors), but other than this I have no further questions.

- As suggested by the reviewer, we have included the TUNEL data into the supplements (Extended Data Figure 8g).

REFEREE #3

The authors are to be congratulated on their revised manuscript which is greatly improved and is undoubtedly a technical tour de force. Similar, the point-by-point response document is very impressive and provides a considerable body of additional work.

We are extremely grateful for these positive comments and helpful suggestions.

1. I have no further technical concerns to raise other than to ask whether the lower baseline liver-to-body weight ratio for mice lacking HSC expression of Rspo3 may have influenced some of the findings in the various liver injury models. This phenotype observation might also be included in the manuscript.

- As suggested by the reviewer, we have included a sentence on the relationship between small liver size and injury, but think that this is unlikely to be a causal factor as liver injury was increased in some models but decreased in others. We have added the following sentence: *“The smaller size of HSC-depleted livers is unlikely to be a contributing factor to altered injury responses, since liver injury was decreased in some and increased in other models.”*, we have included a sentence on the relationship between small

2. The authors may wish to also comment on the work of Zhou Y et al recently published in Metabolism (V154, May 2024) which suggests that Rspo3 in NASH perturbs liver zonation. How does this report impact on novelty of the authors study?

- As suggested, we referenced Zhou et al (Metabolism 2024 – now Ref.55). We also added the sentences *“Accordingly, liver zonation is perturbed in CLD, including MASLD^{54,55}”* and *“Previous studies, using global knockout or over-expression, have found Rspo3 to be an important regulator of liver zonation and hepatocyte proliferation, but did not dissect its cell-specific functions and role in the HSC-hepatocyte crosstalk^{24,41,55}.”* We do not think there is an impact on novelty as the study does not show a role for HSC and HSC-derived Rspo3 – the main finding of the study is the perturbation of zonation in MASH as indicated by the title. Previous studies, e.g. by Tchorz et al, had already described roles for b-catenin, Lgr4 and Rspo3 in liver zonation, but this was not known to be regulated by HSCs.

3. The relevance of HSC-derived Rspo3 in liver homeostasis is only experimentally tested in mice. The authors should at least discuss this limitation since we are provided with no evidence that Rspo3 protein is differentially expressed by HSC in the damaged human liver or that it has the same physiological role in human liver as described here in mice.

- We completely agree and have added the following sentence to the discussion: *“However, further studies are needed to confirm altered RSPO3 protein levels in patients and their associations with outcomes.”*

REFEREE #4

The authors have adequately addressed the concerns raised during the review. The manuscript is now significantly improved. I have no further comments.

Thank you!